# Transformers with RL or SFT Provably Learn Sparse Boolean Functions, But Differently

**Bochen Lyu** [* 1 2]  **Yiyang Jia** [* 3]  **Xiaohao Cai** [1]  **Zhanxing Zhu** [1]

## Abstract

Transformers can acquire Chain-of-Thought (CoT) capabilities to solve reasoning tasks via fine-tuning. Reinforcement learning (RL) and supervised fine-tuning (SFT) are two primary approaches to this end. In this work, we examine RL with verifiable process rewards and SFT for learning $k$-sparse Boolean functions with a one-layer transformer through intermediate reasoning steps akin to CoT. In particular, we consider Boolean functions that can be recursively decomposed into fixed 2-sparse Boolean functions. We first analyze the learning dynamics of RL fine-tuning with verifiable process rewards and SFT in a unified way, allowing us to identify sufficient conditions under which the transformer provably learns these functions. We then verify that these conditions hold for three basic examples, including $k$-PARITY, $k$-AND, and $k$-OR, thus demonstrating their learnability via both RL and SFT. Notably, we reveal that RL and SFT exhibit distinct learning behaviors: RL learns the whole CoT chain simultaneously, whereas SFT learns the CoT step-by-step. Overall, our findings provide insights on the mechanisms underlying RL and SFT and how they differ in triggering the CoT capabilities of transformers, and suggest that the comparison between RL and SFT may need to consider the reward design and the use of teacher forcing.

## 1. Introduction

Large language models (LLMs), with the transformer architecture being their core building block, are remarkably successful across a wide range of tasks, in particular reasoning. LLMs excel in solving complex reasoning tasks by iteratively generating intermediate steps (Wei et al., 2022)—an intriguing approach known as Chain-of-Thought (CoT). Fine-tuning has been shown to be a powerful method for improving CoT generation in LLMs, which in turn improves the multi-step reasoning performance of LLMs significantly (Zelikman et al., 2022; Lightman et al., 2024).

A widely adopted approach for fine-tuning to generate CoT is supervised fine-tuning (SFT), where the transformers are trained to minimize a loss over pairs of inputs and labeled outputs. Another increasingly prevalent fine-tuning approach is reinforcement learning (RL) (DeepSeek-AI et al., 2025; Ouyang et al., 2022; Bai et al., 2022). Instead of minimizing a loss over labeled CoT data, RL guides transformers to generate CoT to solve complex reasoning tasks by maximizing a reward function via policy gradient methods (Shao et al., 2024; DeepSeek-AI et al., 2025; Mnih et al., 2016; Schulman et al., 2017), which has shown significant potential for improving the reasoning capabilities of LLMs.

The remarkable success of fine-tuned transformers has spurred significant interest in understanding their underlying mechanisms. A large number of works have investigated various aspects such as expressivity (Li et al., 2024; Merrill & Sabharwal, 2024) and estimation error (Hu et al., 2024). For SFT, Kim & Suzuki (2025) studied the emergence of CoT by formalizing the mechanism specifically in the bit subset parity problem. They showed that a one-layer transformer provably learns parity in one-update via SFT with teacher-forcing and intermediate supervision akin to CoT, providing promising theoretical insights into reasoning capabilities acquired by SFT. Nevertheless, the theoretical understanding of RL (with process reward) and of how SFT and RL differently enhance reasoning capabilities of transformers is inadequate or even absent. This lack of theoretical understanding stands in stark contrast to the success of RL fine-tuning and the growing body of empirical observations for the comparison between RL and SFT, such as SFT memorizes but RL generalizes (Chu et al., 2025).

In this work, we take the first step towards addressing this gap. We focus on the problem of learning a broad class of functions (Bhattamishra et al., 2023)—*k*-sparse Boolean functions (Def. 2.1) that can be recursively decomposed

---

[*]Equal contribution [1]School of Electronics and Computer Science, University of Southampton, United Kingdom [2]DataCanvas, Beijing, China [3]Independent Researcher. Correspondence to: Zhanxing Zhu <z.zhu@soton.ac.uk>.

*Proceedings of the $43^{rd}$ International Conference on Machine Learning*, Seoul, South Korea. PMLR 306, 2026. Copyright 2026 by the author(s).

into fixed 2-sparse Boolean functions, which is a specific subclass of $k$-sparse Boolean functions and includes the bit subset parity problem as an example. These functions are hard to be learned in an end-to-end manner but can be provably learned with intermediate supervision (Kim & Suzuki, 2025; Wies et al., 2023; Shamir, 2017; Hu et al., 2025) that is akin to CoT, and hence provide suitable case studies for the emergence of reasoning capabilities for transformers through fine-tuning.

In particular, we examine the learning dynamics of *RL fine-tuning with verifiable process reward (Shao et al., 2024)* optimized by vanilla policy gradient, which has not been covered by prior works despite its importance, and *SFT without teacher-forcing* and augmented data, which is not covered by Kim & Suzuki (2025). Furthermore, for theoretical tractability, we analyze a one-layer transformer, consisting of a positional encoding, a self-attention layer, and a feedforward layer. This transformer is iteratively applied to its own output to generate intermediate steps before arriving at the final answer. We use SFT to refer to SFT without teacher forcing, and the extension to SFT with teacher forcing is direct (see Q4 of Appendix B).

**Our contribution.**

- For $k$-sparse Boolean functions (Definition 2.1) that are recursively decomposable via a binary tree structure using a fixed 2-sparse Boolean function, we decompose the learning into multi-step reasoning tasks (Fig. 2a). This allows a unifying analysis on a class of problems.

- We present a unifying approach to analyze the corresponding learning dynamics of both RL with verifiable process reward and SFT for diverse $k$-sparse Boolean functions by identifying a novel *critical gradient component*. Our analysis reveals that the separation of this component serves as a sufficient condition for learnability:

  – For RL (optimized by policy gradient), the transformer learns the target function after *a single gradient update* under the separation condition (Theorem 3.1), acquiring the entire reasoning chain simultaneously. This indicates that RL with process reward can exhibit similar simultaneous learning as SFT with teacher forcing.

  – For SFT (in the absence of both teacher forcing and data augmentation), the learnability is guaranteed under a similar separation condition (Theorem 3.3), but the transformer naturally exhibits a distinct step-wise learning behavior: it learns the reasoning chain *step-by-step*, requiring one gradient update per step and thus demanding a total number of updates equal to the chain length.

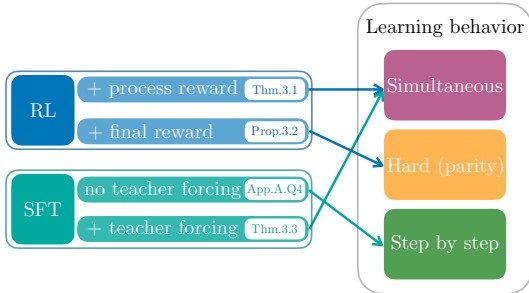

*Figure 1.* Summary of different learning behaviors of RL/SFT under different settings (Fig. 5) when learning the specific sparse Boolean functions with a one-layer transformer defined in Sec. 2.2.

- We apply this unified approach to three basic examples: $k$-PARITY, $k$-AND, and $k$-OR (Sec. 4). In each case, we verify that the separation condition of the critical gradient component is satisfied for both RL and SFT, validating the corresponding learnability of these specific functions.

According to our theoretical results, the empirical comparison between RL and SFT might not be a pure RL vs. SFT, because changing teacher forcing affects learning dynamics of SFT, and changing supervision reward (outcome- or process-based) affects learning dynamics of RL. Thus, the comparisons between RL and SFT should also consider these additional factors to avoid conflation.

**Comparison with related works.** The work most relevant to ours is Kim & Suzuki (2025) (more related works in App. C), which studied learning $k$-PARITY through SFT with teacher-forcing and SFT without teacher-forcing but with data augmentation and self-verification filtering. In contrast, we focus on comparing the learning dynamics of RL fine-tuning and SFT by studying a broad class of sparse Boolean functions, providing a unifying analysis. In addition, our model allows token generation via sampling, better aligning with practical autoregressive inference. Furthermore, we analyze RL fine-tuning and SFT without teacher-forcing and without any data augmentation or self-verification, which addresses the mismatch between training and inference. Together, these elements provide a new conceptual perspective.

Notably, we also make novel theoretical contributions: **(1)** We provide a general, unifying analysis by introducing the new critical gradient component, which captures how the gradient signal can distinguish between relevant and irrelevant positions; **(2)** While Kim & Suzuki (2025) showed the simultaneous learning of SFT with teacher-forcing, we reveal that RL with process reward can also learn the entire CoT chain simultaneously. As the mechanism of SFT with teacher forcing and that of RL are fundamentally distinct, our analysis rigorously gives a novel theoretical insight: two distinct mechanisms can lead to similar efficient

learning; and **(3)** Kim & Suzuki (2025) showed that SFT without teacher-forcing but with data augmentation and self-verification filtering exhibits a step-wise learning behavior. As a comparison, we remove these strategies and show that the step-wise learning can naturally arise from training transformers with CoT, i.e., it can be an intrinsic property of SFT.

**Organization.** We first present the problem setup in Sec. 2. We then establish sufficient conditions for the transformer to learn the general $k$-sparse Boolean functions via RL fine-tuning (Sec. 3.1) and SFT (Sec. 3.2), thereby giving a unified view of learnability. Finally, we examine these sufficient conditions specifically in $k$-PARITY (Sec. 4.1), $k$-AND, and $k$-OR (Sec. 4.2) in detail, proving that both RL and SFT enable the transformer to learn these functions. We add an FAQ in App. B, and present experiments in App. D.

## 2. Setup

### 2.1. Sparse Boolean Functions and the Decomposition

**Definition 2.1** ($k$-sparse Boolean functions). For a $d$-bit input sequence $\mathbf{x} \in \{+1, -1\}^d$ that follows the uniform distribution, let $B = \{i_1, i_2, \ldots, i_k\} \subseteq [d]$ with $2 \leq |B| = k \leq d$, then $\Phi_k : \{+1, -1\}^d \to \{\pm 1\}$ is a $k$-sparse Boolean function if it takes the input $\mathbf{x}$ and is determined by coordinates in $B$, i.e., $\Phi_k(\mathbf{x}) = \phi_k(x_{i_1}, x_{i_2}, \ldots, x_{i_k})$ for some fixed $\phi_k : \{-1, +1\}^k \to \{\pm 1\}$.

In this paper, we only consider $k$-sparse Boolean functions that can be recursively decomposed into fixed 2-sparse Boolean functions $\phi_2$. Specifically, we will study three such $k$-sparse Boolean functions:

- $k$-PARITY (i.e., the bit subset parity): $\Phi_k^{\text{parity}}(\mathbf{x})$ equals $+1$ if the number of $-1$ of $\mathbf{x}$ in $B$ is even and equals $-1$ otherwise, i.e., $\Phi_k^{\text{parity}}(\mathbf{x}) = \prod_{i \in B} x_i$.

- $k$-AND: $\Phi_k^{\text{and}}(\mathbf{x})$ equals $+1$ if $x_i = 1$ for all coordinates $i \in B$ and equals $-1$ otherwise, i.e., $\Phi_k^{\text{and}}(\mathbf{x}) = 2 \prod_{i \in B} \left(\frac{x_i + 1}{2}\right) - 1$.

- $k$-OR: $\Phi_k^{\text{or}}(\mathbf{x})$ equals $+1$ if there is at least one $x_i = 1$ for the coordinates $i \in B$ and equals $-1$ otherwise, i.e., $\Phi_k^{\text{or}}(\mathbf{x}) = 1 - 2 \prod_{i \in B} \left(\frac{1 - x_i}{2}\right)$.

**Recursive Decomposition.** $k$-PARITY is known to be hard to learn with gradient-based methods in an end-to-end manner, as it has been proved that the gradient contains negligible information of the objective function (Shalev-Shwartz et al., 2017). To enable efficient learning, Wies et al. (2023); Kim & Suzuki (2025) decomposed $k$-PARITY into sub-tasks that only perform 2-parity computations, allowing the model to learn in a sequential manner. Inspired by them, we consider $k$-sparse Boolean functions $\Phi_k(\cdot)$ that can be recursively decomposed into fixed 2-sparse Boolean

functions $\phi_2(\cdot, \cdot)$. In particular, we assume $k = 2^T$ for an integer $T$, and recursively decompose the calculation of $\Phi_k(\cdot)$ into $T$ intermediate steps. This forms a length-$T$ reasoning chain in which each step $t \in [T]$ performs a series of fixed 2-sparse Boolean function calculations. The recursive decomposition can be expressed by a complete binary tree with height $T$ and $2k - 1$ nodes (Fig. 2a): each non-leaf node has its value computed by a fixed 2-sparse Boolean function $\phi_2(\cdot, \cdot)$ over its child nodes, and the $t$-th level with $d_t = k/2^t$ nodes accounts for the $t$-th step of the reasoning chain; the lowest level has $k$ nodes $x_{i_j}$ for $i_j \in B$, representing the input data.

Formally, given an input $\mathbf{x} \in \{-1, +1\}^d$, a random subset $B \subseteq [d]$ with $|B| = k$, and a $k$-sparse Boolean function $\Phi_k(\mathbf{x})$, if we denote the sequence at the $t$-th level in Fig. 2a as $\mathbf{y}^{(t)} = (y_1^{(t)}, \ldots, y_{d_t}^{(t)}) \in \{-1, +1\}^{d_t}$ and the input data as $\mathbf{y}^{(0)} = \mathbf{x}$, then

$$\forall t \in [T],\ j \in [d_t]:\ y_j^{(t)} = \phi_2\left(y_{i_1^j}^{(t-1)}, y_{i_2^j}^{(t-1)}\right), \quad (1)$$

where $i_1^j$ and $i_2^j$ are two child nodes of the $j$-th node of level $t$ and are named as *relevant positions* as they determine $y_j^{(t)}$. The final answer is $\mathbf{y}_1^{(T)} = \phi_2(y_1^{(T-1)}, y_2^{(T-1)}) = \Phi_k(\mathbf{x})$.

Finally, given $\Phi_k(\mathbf{x})$, the corresponding subset $B$, and sufficient training samples, the goal of learning on the training samples is to successfully predict $\Phi_k(\mathbf{x})$ for any test $\mathbf{x}$.

**Extension to serial CoT decomposition.** While our main analysis in this paper uses the complete binary-tree decomposition, the analytical framework is not restricted to this balanced structure. It also applies to more common *serial CoT-style decompositions*, where the intermediate reasoning steps are generated one by one. In particular, with $B = \{i_1, \ldots, i_k\}$ for $k \geq 2$, consider the $k$-sparse Boolean functions that can be recursively solved by repeatedly applying the fixed 2-sparse Boolean function in a CoT style:

$$y_1 = \phi_2(x_{i_1}, x_{i_2}),\ y_j = \phi_2(y_{j-1}, x_{i_{j+1}}), \quad j \geq 2.$$

For example, for $k$-PARITY, $\phi_2(a, b) = ab$, so $y_j = \prod_{r=1}^{j+1} x_{i_r}$, namely the parity of the first $j + 1$ relevant bits. This forms a length-$(k - 1)$ reasoning chain in which each step $t \in [k - 1]$ performs a fixed 2-sparse Boolean function calculation (Fig. 3). Compared to the complete-binary tree decomposition, this serial decomposition does not require $k = 2^T$ for some integer $T$ and each token $y_j$ is a reasoning step, where the relevant positions for generating $y_j$ are its last step $y_{j-1}$ and $x_{i_{j+1}}$ from the input $\mathbf{x}$, since all earlier information is summarized by $y_{j-1}$. In App. I, we will show that, for the one-layer transformer introduced in Sec. 2.2 but with a different "pretrained" mask, the same gradient separation analysis applies to this serial decomposition. Therefore, the same learning behaviors hold: RL with process rewards

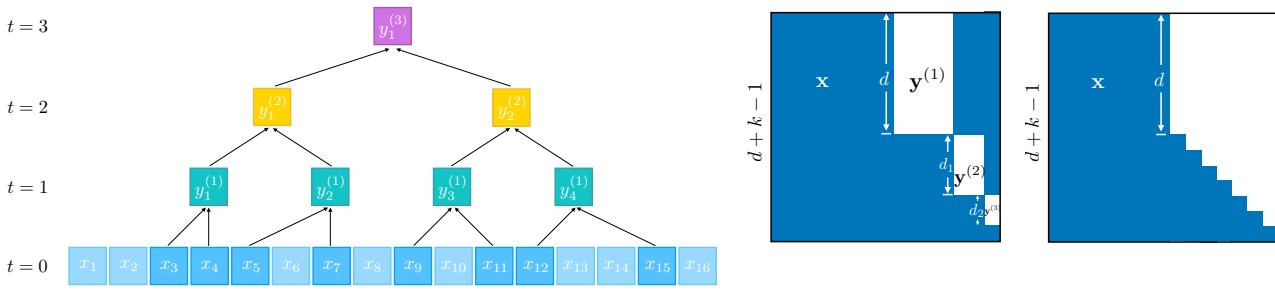

*(a)* Recursive decomposition of solving $\Phi_k(\mathbf{x})$

*(b)* Different masks for $\boldsymbol{W}$

*Figure 2.* **(a)** Recursive decomposition of learning a $k$-sparse Boolean function $\Phi_k(\mathbf{x})$ with a random set $B \subseteq [d]$ (shaded boxes in the lowest level) into solving sub-tasks by following a reasoning chain (bottom to top). Each level of the binary tree corresponds to a step of the reasoning chain, where each node in a level computes a 2-sparse Boolean function $\phi_2(\cdot, \cdot)$ over its two child nodes. **(b)** Self-attention weight $\boldsymbol{W}$ with different masks (blue entries are set as $-\infty$): **left:** the mask for $\mathbf{x}$ and causal mask; **right:** the mask for $\mathbf{x}$, causal mask, and "pretrained" mask.

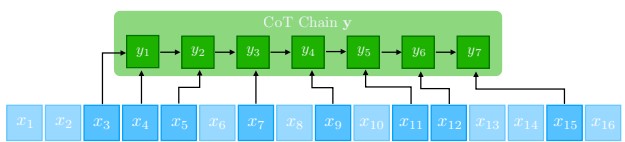

*Figure 3.* Serial CoT-style decomposition of learning a $k$-sparse Boolean function $\Phi_k(\mathbf{x})$ with a random set $B \subseteq [d]$ (shaded boxes in the lowest level) into solving sub-tasks by following a reasoning chain (green boxes). Each green box corresponds to a step of the reasoning chain that computes a 2-sparse Boolean function $\phi_2(\cdot, \cdot)$ over its relevant bits.

and SFT with teacher forcing learn the whole serial CoT chain simultaneously, while SFT without teacher forcing learns step by step under an appropriate filtering rule. Thus, the complete binary tree should be viewed as a tractable decomposition used for our analysis in the main paper, rather than as a fundamental restriction of the framework.

### 2.2. "Pretrained" Transformer and Chain-of-Thoughts

For theoretical tractability, we investigate a one-layer transformer, consisting of the positional encoding, a self-attention layer, and a feedforward layer. Following the recursive decomposition in Sec. 2.1, we will let the transformer iteratively generate the intermediate steps $\mathbf{y}^{(t)}$ with $d_t = k/2^t$ tokens for $t \in [T]$ (akin to CoT) before arriving at the final answer. For convenience, the total number of tokens of the whole reasoning sequence $(\mathbf{x}, \mathbf{y}^{(1)}, \ldots, \mathbf{y}^{(t)})$ is denoted by $N_t = \sum_{\tau=0}^{t} d_\tau$ with $d_0 = d$ and $N_{-1} = 0$.

We introduce the transformer below, from its input to the forward pass, and then explain the Chain-of-Thought.

**Input sequence.** Given an input $\mathbf{x}$ and the recursive decomposition of the $k$-sparse Boolean function $\Phi_k(\mathbf{x})$ (Fig. 2a), we add tokens $\mathbf{y} = (\mathbf{y}^{(1)}, \ldots, \mathbf{y}^{(T)})$ to $\mathbf{x}$ which are all initialized as $\mathbf{0}$ and will be iteratively updated by the transformer to serve as intermediate steps of the reasoning chain. This gives us $\mathbf{z} = (\mathbf{x}, \mathbf{y}) \in \{-1, +1, 0\}^{d+k-1}$.

**Positional encoding.** We concatenate the positional encoding $\mathbf{e}_j \in \mathbb{R}^{d+k-1}$ (the $j$-th standard basis vector of $\mathbb{R}^{d+k-1}$) to $z_j$ to form the complete encoding of the input sequence $\boldsymbol{Z} \in \mathbb{R}^{(d+k)\times(d+k-1)}$ (with $\mathbf{x}$ and $\mathbf{y}$ explicitly written as components)

$$\boldsymbol{Z} = \begin{pmatrix} x_1 & \cdots & x_d & y_1^{(1)} & y_2^{(1)} & \cdots & y_1^{(T)} \\ \mathbf{e}_1 & \cdots & \mathbf{e}_d & \mathbf{e}_{d+1} & \mathbf{e}_{d+2} & \cdots & \mathbf{e}_{d+k-1} \end{pmatrix}.$$

**Self-Attention.** A single-head self-attention layer (without residual connection) then updates $\boldsymbol{Z}$ to

$$f^{\mathrm{SA}}(\boldsymbol{Z}) = \boldsymbol{W}_V \boldsymbol{Z}\, \mathrm{softmax}\left[(\boldsymbol{W}_K \boldsymbol{Z})^\top (\boldsymbol{W}_Q \boldsymbol{Z})\right] \quad (2)$$

where $\boldsymbol{W}_V \in \mathbb{R}^{1\times(d+k)}$ is the value matrix, $\boldsymbol{W}_K, \boldsymbol{W}_Q \in \mathbb{R}^{d'\times(d+k)}$ are the key and query matrices, respectively, and softmax is applied column-wise. In this paper, we simplify the self-attention by merging the key matrix and query matrix as a single matrix $\boldsymbol{W}_{KQ} := \boldsymbol{W}_K^\top \boldsymbol{W}_Q \in \mathbb{R}^{(d+k)\times(d+k)}$ to focus only on the positional encoding while setting $\boldsymbol{W}_V$ to preserve only the $(\mathbf{x}, \mathbf{y})$ component, namely $\boldsymbol{W}_{KQ} = \begin{pmatrix} 0 & \mathbf{0}_{1\times(d+k-1)} \\ \mathbf{0}_{(d+k-1)\times 1} & \boldsymbol{W} \end{pmatrix}$, $\boldsymbol{W}_V = \begin{pmatrix} 1 & \mathbf{0}_{1\times(d+k-1)} \end{pmatrix}$. This has been a common choice in recent works due to its theoretical tractability (e.g., Kim & Suzuki, 2025; von Oswald et al., 2023; Zhang et al., 2023; Lyu et al., 2025; Huang et al., 2023). Then the final form of the output of the self-attention is $[f^{\mathrm{SA}}(\boldsymbol{Z})]_j = \sum_i z_i \sigma_j^i$ where we define the attention score as $\sigma_j^i := \exp(W_{i,j})/\sum_m \exp(W_{m,j})$.

**Feedforward layer.** The feedforward layer includes an activation function $\psi : [-1, 1] \to [0, 1]$ that applies to $f^{\mathrm{SA}}(\boldsymbol{Z})$ element-wise. The form of $\psi$ depends on that of the 2-sparse Boolean function $\phi_2$. This is because $\psi([f^{\mathrm{SA}}(\boldsymbol{Z})]_j)$ is related to how the token at the position $j$ depends on its prior tokens, and such dependence is determined by $\phi_2$.

**"Pretrained" transformer.** Taken together, the transformer $f(\cdot; \boldsymbol{W})$ takes an input sequence $\mathbf{z} = (\mathbf{x}, \mathbf{y})$ and outputs

*Figure 4.* The pretrained transformer iteratively uses its output to solve $\Phi_k(\mathbf{x})$ in a CoT manner.

$f(\mathbf{z}; \mathbf{W}) = \psi(f^{\text{SA}}(\mathbf{Z}))$, where the causal mask $W_{i,j} = -\infty$ for $i \geq j$ is implicitly added and we only consider $j \geq d + 1$ ($W_{i,j} = -\infty$ for any $j \leq d$), because only the reasoning chain $\mathbf{y}$ in the input $\mathbf{z}$ will be iteratively generated by the transformer. Furthermore, since we will consider fine-tuning via RL, the output $f(\mathbf{z}; \mathbf{W})$ should correspond to the next action for the current state, which is related to an intermediate reasoning step in Fig. 2a. Thus, it is natural for each action of RL to account for a reasoning step $\mathbf{y}^{(t)}$ for the reasoning chain $\mathbf{y}$. In light of that $\mathbf{y}^{(t)}$ only depends on $\mathbf{y}^{(t-1)}$ for any $t \in [T]$ (Fig. 2a) and tokens in the same sequence $\mathbf{y}^{(t)}$ are independent of each other, aside from the causal masks, we impose a "pretrained mask" on $\mathbf{W}$ that explicitly exploits such dependence to control the error propagation of the reasoning chain as follows (Fig. 2b): given $t \in [T], \forall j \in [N_{t-1} + 1, N_t]$,

$$W_{i,j} = -\infty \text{ when } \begin{cases} i > d, & \text{if } t = 1, \\ i > N_{t-1} \text{ or } i \leq N_{t-2}, & \text{otherwise.} \end{cases}$$

We heuristically view the transformer $f(\cdot; \mathbf{W})$ with this mask as our "pretrained transformer", which is a result of masking rather than direct pretraining. Despite this distinction, the mask serves as a structural prior, which, in essence, abstractly captures the idea that pretraining provides priors for fine-tuning.

**Forward pass of the transformer.** $f(\cdot; \mathbf{W})$ takes an input sequence $\mathbf{z} = (\mathbf{x}, \ldots, \mathbf{y}^{(t-1)}, \mathbf{y}^{(t)}, \ldots, \mathbf{y}^{(T)})$ and gives

$$[f(\mathbf{z}; \mathbf{W})]_{N_{t-1} + l^{(t)}} = \psi\left(\xi_{l^{(t)}}\right),$$
$$\xi_{l^{(t)}} := \sum_{i=1}^{d_{t-1}} y_i^{(t-1)} \sigma_{N_{t-1} + l^{(t)}}^{N_{t-2} + i}, \tag{3}$$

to update the $l^{(t)} \in [d_t]$ token of the reasoning sequence $\mathbf{y}^{(t)}$, where the attention score $\sigma_j^i$ determines the positions that the current token attends to. Thus, $\xi_{l^{(t)}}$ indicates how tokens of $\mathbf{y}^{(t-1)}$ contribute to the calculation of $y_{l^{(t)}}^{(t)}$.

**Chain-of-Thoughts.** We now describe how the "pretrained" transformer iteratively exploits its own output to generate intermediate steps to solve the $k$-sparse Boolean function, akin to CoT (Fig. 4). Specifically, given the input sequence $\mathbf{z} = (\mathbf{x}, \mathbf{y}^{(1)}, \ldots, \mathbf{y}^{(t)}, \ldots, \mathbf{y}^{(T)})$ with all intermediate reasoning steps $\mathbf{y}^{(t)}$ initialized as $\mathbf{0}$, starting from $t = 1$ to $T$, the transformer $f(\cdot; \mathbf{W})$ iteratively generates the token $\hat{y}_{l^{(t)}}^{(t)}$ of $\hat{\mathbf{y}}^{(t)}$ from $l^{(t)} = 1$ to $l^{(t)} = d_t$

according to Eq. (3), e.g., by sampling from $f(\hat{\mathbf{z}}; \mathbf{W})$ if it is a distribution, and applies the generated token to update $\hat{\mathbf{z}} = (\mathbf{x}, \ldots, \hat{\mathbf{y}}^{(t-1)}, \hat{y}_1^{(t)}, \ldots, \hat{y}_{l^{(t)}-1}^{(t)}, y_{l^{(t)}}^{(t)}, \ldots, \mathbf{y}^{(T)})$. This is repeated until all steps of the reasoning sequence $\hat{\mathbf{z}} = (\mathbf{x}, \ldots, \hat{\mathbf{y}}^{(t-1)}, \hat{\mathbf{y}}^{(t)}, \ldots, \hat{\mathbf{y}}^{(T)})$ are updated, and $\hat{\mathbf{y}}^{(T)}$ will be the final answer.

## 3. Sufficient Conditions for Provable Learning

We now study the learning dynamics of fine-tuning the transformer to learn the $k$-sparse Boolean functions via RL (Sec. 3.1) and via SFT (Sec. 3.2). To this end, we will develop a unifying approach that is capable of examining diverse fine-tuning approaches (RL or SFT) and $k$-sparse Boolean functions at the same time. Our approach hinges on the identification of a new *critical gradient component*, which drives the most substantial learning dynamics during training. For convenience, we denote $\mathbf{y}^{(:t)} := (\mathbf{y}^{(1)}, \ldots, \mathbf{y}^{(t)})$ and $\mathbf{y}^{(t:)} := (\mathbf{y}^{(t)}, \ldots, \mathbf{y}^{(T)})$.

### 3.1. RL Fine-tuning of the "Pretrained" Transformer

**Problem setup.** We view each input $\mathbf{z} = (\mathbf{x}, \mathbf{y})$ as a state and each output of the transformer as a distribution of the token for the action $\mathbf{y}^{(t)}$ (a reasoning step of the CoT chain) under the state $\mathbf{z}$, i.e., $p_{\mathbf{W}}(y_j^{(t)} = 1 | \mathbf{z}) = [f(\mathbf{z}; \mathbf{W})]_{N_{t-1}+j}$. The formulation of the transformer Eq. (3) implies that the distribution over an entire CoT reasoning chain $\mathbf{y}$ conditioned on the initial state $\mathbf{x}$ is

$$p_{\mathbf{W}}(\mathbf{y} | \mathbf{x}) = \prod_{t=1}^{T} p_{\mathbf{W}}(\mathbf{y}^{(t)} | \mathbf{y}^{(t-1)}). \tag{4}$$

Here the sequence $\mathbf{y}^{(t)}$ can be heuristically viewed as an action and a state. With this formulation, starting from the initial state $\mathbf{z} = (\mathbf{x}, \mathbf{y})$ with $\mathbf{y}$ initialized as 0, the transformer solves $\Phi_k(\mathbf{x})$ by generating a reasoning sequence (a trajectory) where each sampled action $\hat{\mathbf{y}}^{(t)}$ updates the state from $\hat{\mathbf{z}} = (\mathbf{x}, \hat{\mathbf{y}}^{(:t-1)}, \mathbf{y}^{(t:)})$ to $(\mathbf{x}, \hat{\mathbf{y}}^{(:t)}, \mathbf{y}^{(t+1:)})$ in a CoT manner (Fig. 4).

Inspired by the process reward in Shao et al. (2024), we provide reward to each reasoning step and analyze the following objective of RL that maximizes the *expected reward*:

**Expected reward:** $\max_{\mathbf{W}} \mathcal{R}(\mathbf{W}) := \max_{\mathbf{W}} \mathbb{E}_{\mathbf{x}}\left[R(\mathbf{x}; \mathbf{W})\right],$

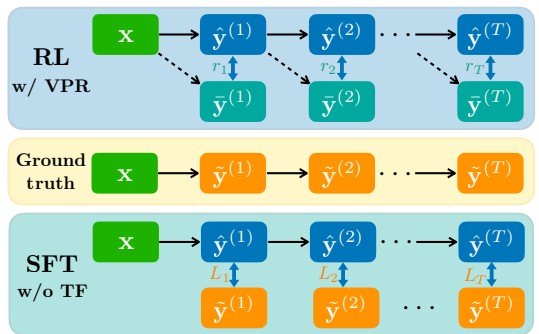

*Figure 5.* **Different learning objectives for RL with verifiable process rewards (RL w/ VPR, top) and SFT without teacher forcing (SFT w/o TF, bottom).** In RL w/ VPR, the model samples autoregressive trajectories $(\hat{\mathbf{y}}^{(1)}, \ldots, \hat{\mathbf{y}}^{(T)})$, and the step-$t$ process reward is computed using $\bar{\mathbf{y}}^{(t)}$, the correct label of $\mathbf{y}^{(t)}$ conditioned on the previously generated step $\hat{\mathbf{y}}^{(t-1)}$; hence the name verifiable process reward. In SFT w/o TF, the model also autoregressively generates $\hat{\mathbf{y}}^{(1)}, \ldots, \hat{\mathbf{y}}^{(T)}$, but each generated step $\hat{\mathbf{y}}^{(t)}$ is directly compared with the fixed ground-truth label $\tilde{\mathbf{y}}^{(t)}$. Hence, RL w/ VPR can learn all steps simultaneously from process rewards along sampled trajectories, while SFT w/o TF learns step by step because later supervision is useful only after earlier generations are correct.

with

$$R(\mathbf{x}; \boldsymbol{W}) := \mathbb{E}_{\mathbf{y} \sim p_{\boldsymbol{W}}(\cdot|\mathbf{x})}[r(\mathbf{x}, \mathbf{y})],$$

$$r(\mathbf{x}, \mathbf{y}) := \sum_{t=1}^{T} r_t \left(\mathbf{y}^{(t)}, \mathbf{y}^{(t-1)}\right),$$

$$(5)$$

$$r_t \left(\mathbf{y}^{(t)}, \mathbf{y}^{(t-1)}\right) := \frac{1}{k-1} \sum_{j=1}^{d_t} y_j^{(t)} \bar{y}_j^{(t)},$$

where $r : \{-1, +1\}^{d+k-1} \to [-1, 1]$ is the reward function, $\bar{y}_j^{(t)}$ is the correct label of $y_j^{(t)}$ given $\mathbf{y}^{(:t-1)}$, and $r_t$ is the $t$-th step process reward (Fig. 5, top panel).

Given $\Phi_k(\cdot)$ and its recursive decomposition as in Def. 2.1, if we let $\boldsymbol{W}^\star = \arg\max_{\boldsymbol{W}} \mathcal{R}(\boldsymbol{W})$, i.e., $\mathcal{R}(\boldsymbol{W}^\star) = 1$, then the transformer $f(\cdot; \boldsymbol{W}^\star)$ exhibits a unique self-attention score that only attends to the relevant positions and ignores all irrelevant ones, i.e., $\forall t \in [T], l^{(t)} \in [d_t]$ :

$$[\text{softmax}(\boldsymbol{W}^\star)]_{N_{t-1}+l^{(t)}}^{N_{t-2}+p} = \begin{cases} \frac{1}{2}, & \text{if } p \in \{i_1^{l^{(t)}}, i_2^{l^{(t)}}\}, \\ 0, & \text{otherwise.} \end{cases}$$

We now consider RL optimized by policy gradient to approximate the optimal parameters $\boldsymbol{W}^\star$. The policy gradient is computed by (Lem. E.1) $\nabla_{\boldsymbol{W}} R(\mathbf{x}; \boldsymbol{W}) = \mathbb{E}_{\mathbf{y} \sim p_{\boldsymbol{W}}(\cdot|\mathbf{x})}[\sum_{t=1}^{T} \nabla_{\boldsymbol{W}} \ln p_{\boldsymbol{W}}(\mathbf{y}^{(t)}|\mathbf{y}^{(t-1)}) r_{t:}]$ with $r_{t:} := \sum_{\tau=t}^{T} r_\tau (\mathbf{y}^{(\tau)}, \mathbf{y}^{(\tau-1)})$. Notably, when optimizing $p_{\boldsymbol{W}}(\mathbf{y}^{(t)}|\mathbf{y}^{(t-1)})$, the $t$-th step reward $r_t$ is considerably more important than other $r_{t'}$ with $t \neq t'$ (e.g., the model can achieve high $r_t$ even when it completely fails at later steps). This is because the correctness of $y_j^{(t)}$ generated

in the current action $t$ only depends on its two child nodes $y_{i_1^j}^{(t-1)}$ and $y_{i_2^j}^{(t-1)}$ (Fig. 2a). Thus we consider optimizing $\max_{\boldsymbol{W}} \mathcal{R}(\boldsymbol{W})$ with the following gradient

$$\nabla_{\boldsymbol{W}} R(\mathbf{x}; \boldsymbol{W}) = \mathbb{E}_{\mathbf{y} \sim p_{\boldsymbol{W}}(\cdot|\mathbf{x})} \left[ \sum_{t=1}^{T} \nabla_{\boldsymbol{W}} \ln p_{\boldsymbol{W}}\left(\mathbf{y}^{(t)}|\mathbf{y}^{(t-1)}\right) \right.$$
$$\left. \times r_t\left(\mathbf{y}^{(t)}, \mathbf{y}^{(t-1)}\right) \right]. \quad (6)$$

**Intuitive analysis.** During policy gradient training, $f(\cdot; \boldsymbol{W})$ generates a large number of trajectories (reasoning sequences) and receives a reward for each action. Thus, for the $t$-th action $\mathbf{y}^{(t)}$, as long as the policy gradient Eq. (6) contains sufficient information to tell the relevant positions (the child nodes of a token of $\mathbf{y}^{(t)}$) from the irrelevant ones (other nodes), $f(\cdot; \boldsymbol{W})$ can be optimized to obtain higher $t$-th step reward $r_t$. Interestingly, such separation of the policy gradient can be shown to be equivalent to the separation of the *critical gradient component*[1] (Lem. E.2) defined by

**Critical gradient component:**

$$\gamma_{l^{(t)}}^j(\mathbf{y}^{(t-1)}) := \frac{2}{k-1} \psi'(\xi_{l^{(t)}}) \phi_2\left(y_{i_1^{l^{(t)}}}^{(t-1)}, y_{i_2^{l^{(t)}}}^{(t-1)}\right) y_j^{(t-1)}$$
$$(7)$$

which is determined by the activation function $\psi$ and the 2-sparse Boolean function $\phi_2$ given $t \in [T]$ and $l^{(t)} \in [d_t]$. In addition, the $t$-th step reward $r_t$ does not depend on rewards of other steps according to the form of the output of the transformer Eq. (3). As a result, rewards for different steps can be optimized simultaneously under the separation of $\gamma_{l^{(t)}}^j$ for all tokens, which guarantees the learnability.

Below we present a more precise characterization for optimizing $\max_{\boldsymbol{W}} \mathcal{R}(\boldsymbol{W})$ with the sign of the policy gradient Eq. (6), as it will be positive at relevant positions and negative at irrelevant ones under the separation condition.

**Theorem 3.1** (Learnability via RL fine-tuning). *Given integers $d \geq k \geq 2$, consider a $k$-sparse Boolean function $\Phi_k(\cdot)$ with any subset $B \subseteq [d]$ as in Def. 2.1. Let $\boldsymbol{W}(0) = \mathbf{1}$ be the initialization, $\boldsymbol{W}^\star = \arg\max_{\boldsymbol{W}} \mathcal{R}(\boldsymbol{W})$ be the optimal parameters that solve $\max_{\boldsymbol{W}} \mathcal{R}(\boldsymbol{W})$, and $\eta = \Omega(\ln(d/\epsilon))$ for $\epsilon > 0$ be the learning rate.*

*If the separation of the critical gradient component $\gamma_{l^{(t)}}^p$ is satisfied $\forall t \in [T], l^{(t)} \in [d_t]$ and $\forall p \in \{i_1^{l^{(t)}}, i_2^{l^{(t)}}\}$, $p' \in [d_{t-1}] \backslash \{i_1^{l^{(t)}}, i_2^{l^{(t)}}\}$ ($p$ is a child node of $y_{l^{(t)}}^{(t)}$ but $p'$ is not):*

$$\mathbb{E}_{\mathbf{x}, \mathbf{y}^{(:t-1)} \sim p_{\boldsymbol{W}}(\cdot|\mathbf{x})}\left[\gamma_{l^{(t)}}^p(\mathbf{y}^{(t-1)}) - \gamma_{l^{(t)}}^{p'}(\mathbf{y}^{(t-1)})\right] > 0$$
$$(8)$$

*then fine-tuning the transformer $f(\cdot; \boldsymbol{W})$ via RL optimized by the sign of the policy gradient Eq. (6) after one up-*

---

[1]This name is because the policy gradient can be expressed by the critical gradient component, see Lem. E.2.

date $\boldsymbol{W}(1) = \boldsymbol{W}(0) + \eta \, \mathrm{sign}\left(\nabla_{\boldsymbol{W}} \mathcal{R}(\mathbf{x}; \boldsymbol{W})\right)$ *achieves* $\|\mathrm{softmax}(\boldsymbol{W}(1)) - \mathrm{softmax}(\boldsymbol{W}^{\star})\|_1 \leq \epsilon$.

To the best of our knowledge, Thm. 3.1 studies the learnability of $k$-sparse Boolean functions for transformers with CoT through RL fine-tuning for the first time. The learnability is guaranteed by the ***separation of the critical gradient component*** Eq. (8). Thm. 3.1 reveals the benefit of RL—learning the whole CoT chain in one-update, highlighting that RL can achieve similar efficient learning as SFT with teacher forcing (Kim & Suzuki, 2025).

**Process reward vs. final reward.** Thm. 3.1 is for RL with process reward, as the recursive decomposition (Fig. 2a) inherently demands verification of sequential steps as in math problems (Shao et al., 2024). As a comparison, another approach is RL with final reward $r^{\mathrm{F}} = \mathbf{y}^{(T)} \Phi_k(\mathbf{x})$ that is provided only for the final answer. This approach admits a different objective $\max_{\boldsymbol{W}} \mathcal{R}^{\mathrm{F}}(\boldsymbol{W}) := \max_{\boldsymbol{W}} \mathbb{E}_{\mathbf{x}, \mathbf{y} \sim p_{\boldsymbol{W}}(\cdot | \mathbf{x})}[r^{\mathrm{F}}(\mathbf{x}, \mathbf{y})]$. Below we show its hardness in contrast to the learnability of RL with process reward inspired by Shalev-Shwartz et al. (2017).

**Proposition 3.2** (Hardness of RL with final reward). *Let $\mathscr{H}$ be a class of functions $h : \{-1, +1\}^d \rightarrow \{-1, +1\}$ such that $\mathbb{E}_{\mathbf{x}}[h(\mathbf{x})h'(\mathbf{x})] = 0$ for any two distinct $h, h' \in \mathscr{H}$. Then for the model $p_{\boldsymbol{W}}(\cdot | \mathbf{x})$ with bounded gradient of the final output $\mathbb{E}_{\mathbf{x}}[\|\nabla_{\boldsymbol{W}} \mathbb{E}_{\mathbf{y} \sim p_{\boldsymbol{W}}(\cdot | \mathbf{x})}[\mathbf{y}^{(T)}]\|^2] \leq M$ and the objective of RL with final reward $\max_{\boldsymbol{W}} \mathcal{R}_h^{\mathrm{F}}(\boldsymbol{W}) = \mathbb{E}_{\mathbf{x}, \mathbf{y} \sim p_{\boldsymbol{W}}(\cdot | \mathbf{x})}[r_h^{\mathrm{F}}(\mathbf{x}, \mathbf{y})]$ where $r_h^{\mathrm{F}}(\mathbf{x}, \mathbf{y}) = \mathbf{y}^{(T)} h(\mathbf{x})$, the variance of the policy gradient $\nabla_{\boldsymbol{W}} \mathcal{R}_h^{\mathrm{F}}(\boldsymbol{W})$ is bounded as*

$$\mathrm{Var}(\mathscr{H}; \boldsymbol{W}) := \mathbb{E}_{h \in \mathscr{H}} \left[ \left\| \nabla_{\boldsymbol{W}} \mathcal{R}_h^{\mathrm{F}}(\boldsymbol{W}) \right.\right.$$
$$\left.\left. - \mathbb{E}_{h' \in \mathscr{H}}[\nabla_{\boldsymbol{W}} \mathcal{R}_{h'}^{\mathrm{F}}(\boldsymbol{W})] \right\|^2 \right] \leq \frac{M}{|\mathscr{H}|}. \quad (9)$$

We consider sparse parity. Let $\mathscr{H}$ be the collection of all possible sparse parity functions with $|\mathscr{H}| = 2^d$, which satisfies the condition of Prop. 3.2 (Shalev-Shwartz et al., 2017), and our target function is uniformly chosen from $\mathscr{H}$, then $\mathrm{Var}(\mathscr{H}; \boldsymbol{W})$ is exponentially small in $d$, indicating that the signal of target function contained in the policy gradient is drowned out by noise. This makes learning from such gradients difficult.

### 3.2. SFT of the Pretrained Transformer

**Problem setup.** SFT is a straightforward approach that aims to minimize a loss over pairs of labeled sequences and generated outputs. Formally, for each input $\mathbf{z} = (\mathbf{x}, \mathbf{0})$ with the CoT chain $\mathbf{y}$ initialized as $\mathbf{0}$, we let $\tilde{\mathbf{z}} = (\mathbf{x}, \tilde{\mathbf{y}}) = (\mathbf{x}, \tilde{\mathbf{y}}^{(1)}, \ldots, \tilde{\mathbf{y}}^{(T)})$ be the labeled sequence where $\tilde{\mathbf{y}}$ is the ground-truth CoT chain for solving $\Phi_k(\mathbf{x})$ (Fig. 2a). Following the CoT generation discussed in Fig. 4, given the input sequence $\mathbf{z} = (\mathbf{x}, \mathbf{0})$, the transformer solves $\Phi_k(\mathbf{x})$ by

iteratively applying its own output to generate

$$\hat{y}_{l^{(t)}}^{(t)} = \mathrm{sign}\left(\hat{q}_{l^{(t)}}^{(t)}\right), \ \hat{q}_{l^{(t)}}^{(t)} := 2[f(\hat{\mathbf{z}}; \boldsymbol{W})]_{N_{t-1} + l^{(t)}} - 1 \quad (10)$$

and update

$$\hat{\mathbf{z}} = (\mathbf{x}, \hat{\mathbf{y}}^{(:t-1)}, \hat{y}_1^{(t)}, \ldots, \hat{y}_{l^{(t)}-1}^{(t)}, y_{l^{(t)}}^{(t)}, \ldots, \mathbf{y}^{(t+1:)})$$

from $l^{(t)} = 1$ to $d_t$ and from $t = 0$ to $T$, until the whole CoT chain $\hat{\mathbf{y}}$ is generated such that $\hat{\mathbf{z}} = (\mathbf{x}, \hat{\mathbf{y}})$. The labeled sequence and the generated output pair is now $(\tilde{\mathbf{z}}, \hat{\mathbf{z}})$.

SFT now aims to match $\tilde{\mathbf{z}}$ and $\hat{\mathbf{z}}$ by minimizing a loss over them. To this end, we use the hinge-loss $\ell(\hat{a}, a) := \max(0, 1 - \hat{a}a)$ for $a \in \{-1, +1\}$ and $\hat{a} \in \mathbb{R}$, but the SFT result is not specific to hinge loss, as the step-wise learning dynamics intuitively comes from the autoregressive nature, i.e., later CoT steps only become reliably learnable once earlier steps are correct (App. F.2).

The population loss $\mathcal{L}(\boldsymbol{W})$ (Fig. 5, bottom panel) of our problem will be ($\hat{q}_j^{(t)}$ can be viewed as a score)

$$\textbf{Population loss:} \ \mathcal{L}(\boldsymbol{W}) := \mathbb{E}_{\mathbf{x}} \left[ \sum_{t=1}^T L_t(\mathbf{x}, \boldsymbol{W}) \right],$$
$$L_t(\mathbf{x}, \boldsymbol{W}) := \frac{1}{k-1} \sum_{j=1}^{d_t} \ell\left(\hat{q}_j^{(t)}, \tilde{y}_j^{(t)}\right); \quad (11)$$

and the objective of SFT is $\min_{\boldsymbol{W}} \mathcal{L}(\boldsymbol{W})$ by, e.g., gradient descent. We highlight that our approach in this section does not employ the teacher-forcing in prior works (Wies et al., 2023; Kim & Suzuki, 2025). In particular, our input sequence for the generation of CoT tokens Eq. (10) is $\hat{\mathbf{z}}$ whose intermediate tokens are all generated by the transformer in a natural autoregressive manner, while SFT with teacher-forcing employs the ground-truth sequence $\tilde{\mathbf{z}}$ as input sequence to generate tokens of the CoT. As a result, SFT with teacher forcing has a mismatch between training and inference. Therefore, we consider SFT without teacher forcing to address such mismatch.

**Intuitive analysis.** SFT admits a fundamentally distinct training objective compared to RL: the transformer generates a whole CoT chain in a natural auto-regressive way and compares it with the ground-truth label to minimize the loss. Thus, the ground-truth label of later CoT steps can be used to minimize the loss only if the prior steps are generated correctly.

This suggests learning dynamics driven by induction: **(1)** suppose that at the first gradient update SFT only optimizes $\min_{\boldsymbol{W}} \mathbb{E}_{\mathbf{x}}[L_1(\mathbf{x}, \boldsymbol{W})]$ to learn the first step of the CoT chain $\mathbf{y}^{(1)}$, then the previous step of $\mathbf{y}^{(1)}$ is $\mathbf{x}$, which is already correct because $\mathbf{x}$ itself is the ground-truth, thus

the ground-truth label $\tilde{\mathbf{y}}^{(1)}$ can be faithfully used for optimizing $\min_{\mathbf{W}} \mathbb{E}_{\mathbf{x}}[L_1(\mathbf{x}, \mathbf{W})]$; **(2)** similar to the case for RL, as long as the gradient $\nabla_{\mathbf{W}} \mathbb{E}_{\mathbf{x}}[L_1(\mathbf{x}, \mathbf{W})]$ can sufficiently separate the relevant positions from irrelevant ones, the population loss of the first step $\mathbb{E}_{\mathbf{x}}[L_1(\mathbf{x}, \mathbf{W})]$ can be minimized such that the generated $\hat{\mathbf{y}}^{(1)}$ can approximate the ground-truth $\tilde{\mathbf{y}}^{(1)}$ after one-gradient update; **(3)** now at the second gradient update, if SFT only optimizes $\min_{\mathbf{W}} \mathbb{E}_{\mathbf{x}}[L_2(\mathbf{x}, \mathbf{W})]$ to learn $\mathbf{y}^{(2)}$, then its previous step $\hat{\mathbf{y}}^{(1)}$ can be correctly generated by the transformer, and thus the ground-truth label $\tilde{\mathbf{y}}^{(2)}$ can be used as in the learning of $\mathbf{y}^{(1)}$; and **(4)** repeating this argument suggests that one gradient update solves one step $\mathbf{y}^{(t)}$. As a result, $T$ gradient updates solve the whole CoT chain.

Notably, such step-wise learning can emerge naturally in SFT of transformer—it is an intrinsic property that does not rely on data augmentation with the corresponding filter in Kim & Suzuki (2025) or curriculum learning. Below we present a more exact characterization, which also depends on the *critical gradient component* Eq. (7) yet in a slightly different way, illustrating the unifying nature of our approach. To be consistent with that for RL, we use sign gradient descent, which can be viewed as a special version of Adam (Kingma & Ba, 2017).

**Theorem 3.3** (Learnability via SFT). *Given integers $d \geq k \geq 2$, consider a $k$-sparse Boolean function $\Phi_k(\cdot)$ with any subset $B \in [d]$ as in Def. 2.1. Let $\mathbf{W}(0) = \mathbf{1}$ be the initialization, $\mathbf{W}^\star = \arg\min_{\mathbf{W}} \mathcal{L}(\mathbf{W})$ be the optimal parameters that solve $\min_{\mathbf{W}} \mathcal{L}(\mathbf{W})$, and $\eta = \Omega\left(\ln(d/\epsilon)\right)$ for $\epsilon > 0$ be the learning rate.*

*For the transformer $f(\cdot; \mathbf{W})$ fine-tuned via SFT by running sign gradient descent $\mathbf{W}(s + 1) = \mathbf{W}(s) - \eta\,\text{sign}\left(\nabla_{\mathbf{W}} \mathcal{L}(\mathbf{W}(s))\right)$, if the separation of the critical gradient component is satisfied for any $l^{(t)} \in [d_t]$ and any $t \in [T]$ in the sense that $\forall p \in \{i_1^{l^{(t)}}, i_2^{l^{(t)}}\}, p' \in [d_{t-1}] \backslash \{i_1^{l^{(t)}}, i_2^{l^{(t)}}\}$ :*

$$\mathbb{E}_{\mathbf{x}}\left[\gamma_{l^{(t)}}^p(\tilde{\mathbf{y}}^{(t-1)}) - \gamma_{l^{(t)}}^{p'}(\tilde{\mathbf{y}}^{(t-1)})\right] > 0, \qquad (12)$$

*where $\tilde{\mathbf{y}}^{(t-1)}$ is the ground-truth label of $\mathbf{y}^{(t-1)}$ given an input $\mathbf{x}$, then running sign gradient descent for $T$ iterations achieves $\|\text{softmax}(\mathbf{W}(T)) - \text{softmax}(\mathbf{W}^\star)\|_1 \leq \epsilon$.*

### 3.3. Summary of the Learning Dynamics

Thm. 3.1 and 3.3 show that both RL and SFT enable the pretrained transformer to acquire CoT generation ability to learn $k$-sparse Boolean functions under the separation of $\gamma_{l^{(t)}}^p$ between the case when $p$ is a child node of $l^{(t)}$ and that when $p$ is not. Given $t \in [T]$ and $l^{(t)} \in [d_t]$, when generating the $l^{(t)}$-th token of the reasoning sequence $\mathbf{y}^{(t)}$, it is crucial for the self-attention to correctly focus on the relevant positions $i_1^{l^{(t)}}$ and $i_2^{l^{(t)}}$ (the child nodes of $l^{(t)}$ (Fig. 2a)).

This guarantees that the sub-task Eq. (1) can be faithfully solved by the transformer. Our established conditions ensure that only attention scores of these relevant positions will be increased while that of all irrelevant positions will be decreased during the sign gradient training. Hence, the learnability of the transformer is obtained.

**Why different learning behaviors?** While the condition for fine-tuning via RL and that via SFT are similar, the transformer exhibits distinct learning behaviors for these two approaches due to their different training objectives (Fig. 5). RL allows the transformer to learn the whole CoT chain *simultaneously*, as it can learn each reasoning step $\mathbf{y}^{(t)}$ regardless of the learning of its prior steps given the reward, even though these prior steps are imperfectly learned, and thus one-update is sufficient. As a comparison, the transformer learns the reasoning chain *step-by-step* via SFT and demands $T$-updates for the whole chain. Intuitively, on one hand, the ground-truth labels of all steps of the CoT are fixed and determined by the input $\mathbf{x}$; on the other hand, the generation of later reasoning steps by transformers depends on the prior generated steps during SFT training. If the prior steps are not correctly generated by the transformer, then it cannot approximate ground-truth labels of later steps, since these labels are obtained from the correct prior steps which the transformer cannot generate. As a result, SFT must ensure the learning of prior steps before proceeding to later steps, leading to a step-wise learning behavior. In a similar essence, SFT with teacher forcing will learn simultaneously (Q4 of App. B).

In addition, the true label of the $t$-th step process reward of RL, $r_t$, depends only on the $(t-1)$-step, whereas in SFT all the intermediate labels are directly computed from the input $\mathbf{x}$. In the case when the true label in the $t$-th step process reward is also computed from $\mathbf{x}$ (like SFT), the learning dynamics of RL can become step-wise like SFT due to the autoregressive nature of transformers (see App. H, where we provide an exact solution to such learning dynamics).

## 4. PARITY, AND, and OR Can Satisfy the Separation Condition

Thm. 3.1 and Thm. 3.3 establish a general, unifying condition of the learnability for diverse $k$-sparse Boolean functions, i.e., the separation of the critical gradient component $\gamma_{l^{(t)}}^p$ Eq. (7). In this section, we examine whether the separation condition holds for three basic examples, including $k$-PARITY, $k$-AND and $k$-OR, allowing us to obtain the learnability results for these specific functions.

The critical gradient component depends on the forms of the activation $\psi(\cdot)$ and the 2-sparse Boolean function $\phi_2(\cdot, \cdot)$, which vary for different $k$-sparse Boolean functions. In Tab. 1, we summarize $\psi(\cdot)$ and $\phi(\cdot, \cdot)$ for $k$-PARITY, $k$-

*Table 1.* Formulations of $\psi(\cdot)$ and $\phi_2(\cdot, \cdot)$ combinations for different $k$-sparse Boolean functions.

| Function | $\psi(z)$ | $\phi_2(z_1, z_2)$ |
|:---:|:---:|:---:|
| $k$-PARITY | $z^2$ | $z_1 z_2$ |
| $k$-AND | $\max(z, 0)$ | $\frac{z_1 z_2 + z_1 + z_2 - 1}{2}$ |
| $k$-OR | $\min(z, 0) + 1$ | $\frac{-z_1 z_2 + z_1 + z_2 + 1}{2}$ |

AND and $k$-OR, respectively. In particular, the activation function $\psi : [-1, 1] \rightarrow [0, 1]$ ensures that the output of the transformer Eq. (3) can be seen as the probability of generating $y = 1$ for RL and a score of the token for SFT. $\psi(\cdot)$ should guarantee that $\psi((z_1 + z_2)/2)$ is large if $\phi_2(z_1, z_2) = 1$ such that the transformer has sufficient expressibility for the target $k$-sparse Boolean functions.

We will verify that the conditions in Thm. 3.1 and Thm. 3.3 are satisfied by the critical gradient component Eq. (7) induced by the $\psi(\cdot)$ and $\phi_2(\cdot, \cdot)$ combinations in Tab. 1. This reveals the learnability of the specific sparse Boolean functions in Tab. 1 via RL or SFT. We present numerical experiments in App. D to support the theoretical claims.

## 4.1. PARITY

For $k$-PARITY, $\Phi_k^{\text{parity}}(\mathbf{x}) = \prod_{i \in B} x_i$ for $B \subseteq [d]$, which returns $+1$ if the number of $-1$ of $\mathbf{x} \in \{-1, +1\}^d$ in $B$ is even and $-1$ otherwise. This gives us the 2-PARITY function $\phi_2(z_1, z_2) = z_1 z_2$ for the recursive decomposition (Sec. 2.1). We use $\psi(z) = z^2$ for $z \in [-1, 1]$.

Below we provide a more comprehensive characterization for the learning dynamics of RL fine-tuning for *arbitrary iterations* of the policy gradient as well as a discussion of the separation of critical gradient component $\gamma_{l^{(t)}}^p$.

**Theorem 4.1** ($k$-PARITY learning dynamics of RL fine-tuning). *Under the setting of Thm. 3.1, for $\Phi_k^{\text{parity}}(\mathbf{x})$, let $\psi(z) = z^2$ and $\phi_2(z_1, z_2) = z_1 z_2$. For a learning rate $\eta > 0$, if we run RL optimized by sign policy gradient for $S \in \mathbb{Z}^+$ iterations, then:*

*(1)* $\forall t \in [T]$, $l^{(t)} \in [d_t]$, *the separation of the critical gradient component $\gamma_{l^{(t)}}^p$ Eq. (8) is satisfied at each policy gradient iteration $s \in [S]$.*

*(2) The attention score (Eq. (3)) at the $s$-th policy gradient iteration has the form of*

$$\sigma_{N_{t-1}+l^{(t)}}^{N_{t-2}+p}(s) = \begin{cases} \frac{1}{2} \frac{1}{1 + \frac{d_{t-1} - 2}{2} \exp(-2\eta s)}, & p \in \{i_1^{l^{(t)}}, i_2^{l^{(t)}}\}, \\ \frac{1}{d_{t-1} - 2 + 2\exp(2\eta s)}, & \text{otherwise.} \end{cases}$$

Thm. 4.1 reveals that the separation condition for the critical gradient component $\gamma_{l^{(t)}}^p$ Eq. (8) is satisfied, and thus one-step update of RL can enable the transformer to learn the $k$-PARITY. Furthermore, Thm. 4.1 characterizes the

learning dynamics of the attention score for *arbitrary iterations*, which highlights that either a large learning rate $\eta$ or sufficient iteration number $S$ can lead the self-attention to focus on the relevant positions and ignore irrelevant ones:

$$\sigma_{N_{t-1}+l^{(t)}}^{N_{t-2}+p}(S) \overset{\text{large } S \text{ or } \eta}{\rightarrow} \begin{cases} \frac{1}{2}, & \text{if } p \in \{i_1^{l^{(t)}}, i_2^{l^{(t)}}\}, \\ 0, & \text{otherwise.} \end{cases}$$

In the following, we confirm that $k$-PARITY can also be learned by transformers via SFT in a step-wise learning manner, in contrast to the simultaneous learning of RL.

**Claim 4.1** (Transformers with CoT can learn $k$-PARITY via SFT). *Under the setting of Thm. 3.3, for $k$-PARITY $\Phi_k^{\text{parity}}(\mathbf{x})$, let $\psi(z) = z^2$ and $\phi_2(z_1, z_2) = z_1 z_2$, then the separation of critical gradient component $\gamma_{l^{(t)}}^p$ Eq. (12) is satisfied, and thus the learnability of Thm. 3.3 for SFT is guaranteed.*

### 4.2. AND and OR

We study two more $k$-sparse Boolean functions, $k$-AND with $\Phi_k^{\text{and}}(\mathbf{x}) = 2 \prod_{i \in B} \frac{x_i + 1}{2} - 1$ and $k$-OR with $\Phi_k^{\text{or}}(\mathbf{x}) = 1 - 2 \prod_{i \in B} \frac{1 - x_i}{2}$, and the functions $\psi(\cdot)$ and $\phi_2(\cdot, \cdot)$ are listed in Tab. 1. We show that both the condition in Thm. 3.1 for RL and that in Thm. 3.3 for SFT are satisfied, confirming the learnability of $k$-AND and $k$-OR for transformers via either RL or SFT.

**Claim 4.2** (Transformers with CoT can learn $k$-AND and $k$-OR via both RL and SFT). *Under the setting of Thm. 3.1 for RL (Thm. 3.3 for SFT), for $k$-AND $\Phi_k^{\text{and}}(\mathbf{x})$ and $k$-OR $\Phi_k^{\text{or}}(\mathbf{x})$, with their respective $\psi(\cdot)$ and $\phi_2(\cdot, \cdot)$ given in Tab. 1, the separation of $\gamma_{l^{(t)}}^p$ Eq. (8) for RL (Eq. (12) for SFT) is satisfied; thus the learnability of Thm. 3.1 for RL (Thm. 3.3 for SFT) is obtained for $\Phi_k^{\text{and}}(\cdot)$ and $\Phi_k^{\text{or}}(\cdot)$.*

## 5. Discussion

In this paper, we have investigated the learning dynamics of fine-tuning transformers with CoT via either RL or SFT for learning $k$-sparse Boolean functions, including $k$-PARITY, $k$-AND, and $k$-OR. We have established sufficient conditions for the provable learning of both RL and SFT—the separation of the critical gradient component. Furthermore, our results reveal that, while both RL and SFT are capable of learning these $k$-sparse Boolean functions, they exhibit distinct learning behaviors: RL learns the whole CoT chain simultaneously but SFT must solve the prior steps of CoT chain before learning the later steps, leading to a step-wise learning phenomenon. Our findings take the first step towards understanding the mechanism of fine-tuning transformers with CoT via RL and provide additional theoretical insights on that via SFT by removing teacher-forcing as well as the additional data augmentation, resulting in a tractable comparison between them. We disucss limitations and future directions in App. A.

# Acknowledgements

B.L. is funded by a studentship provided by the School of Electronics and Computer Science, University of Southampton. The authors acknowledge DataCanvas AlayaNeW for providing computational resources. The authors thank the anonymous reviewers for their feedback.

# Impact Statement

This paper presents work whose goal is to advance the field of machine learning. There are many potential societal consequences of our work, none of which we feel must be specifically highlighted here.

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

# Appendix

The appendix is organized as follows:

For convenience, we summarize our notations in Table 2.

## A. Limitation and Future Directions

As we focus on the learning dynamics in a theoretically tractable setting, our analysis relies on simplifying assumptions, including a one-layer transformer, special structured attention masks, and special Boolean functions that can be decomposed into fixed 2-sparse Boolean functions. The class of such Boolean functions is clearly only a subset of all Boolean functions. A natural future direction is the extension to more realistic architectures, such as multi-layer and/or multi-head transformers.

Another important direction is broader empirical validation beyond the special Boolean functions considered in this paper. Experiments on more realistic reasoning tasks would help clarify how the proposed mechanism relates to empirical observations in general post-training.

Finally, our theory mainly provides sufficient conditions for learnability; failure should occur when the separation condition fails, so that the training signal no longer consistently favors relevant positions over irrelevant ones, such as RL with only final reward (Prop. 3.2). A more systematic analysis of finite-sample effects, noisy optimization, and imperfect rewards is an important future direction.

## B. FAQ

1. **Q:** What is the main problem setup of this paper?

    **A:** This paper studies learning $k$-sparse Boolean functions that can be recursively decomposed as fixed 2-sparse Boolean functions (Sec. 2.1) with a one-layer transformer (Sec. 2.2). Such Boolean functions are only a subset of all possible Boolean functions. Furthermore, the contributions are mostly theoretical, i.e., a theoretical characterization of learning dynamics in a tractable setting.

*Table 2.* **Notations**

| | |
|---|---|
| $\mathbf{x}$ | input sequence |
| $\mathbf{y}$ | intermediate reasoning sequence |
| $\mathbf{y}^{(t)}$ | $t$-th step reasoning sequence |
| $y_j^{(t)}$ | $j$-th token of $\mathbf{y}^{(t)}$ |
| $\mathbf{y}^{(:t)}$ | $(\mathbf{y}^{(1)}, \ldots, \mathbf{y}^{(t)})$ |
| $\mathbf{y}^{(t:)}$ | $(\mathbf{y}^{(t)}, \ldots, \mathbf{y}^{(T)})$ |
| $\hat{y}_j^{(t)}, \hat{\mathbf{y}}$ | generated reasoning sequence |
| $\mathbf{z}$ | $(\mathbf{x}, \mathbf{y})$ |
| $\mathbf{Z}$ | $\mathbf{z}$ with positional encoding |
| $\Phi_k$ | $k$-sparse Boolean function |
| $\phi_2$ | 2-sparse Boolean function |
| $\psi$ | activation function |
| $[T]$ | integers between 1 and T |
| $d_t$ | number of nodes in the $t$-th level |
| $N_t$ | total number of tokens in $(\mathbf{x}, \mathbf{y}^{(:t)})$ |
| $\sigma_j^i$ | attention score |
| $\mathcal{R}$ | expected reward of RL |
| $R(\mathbf{x}; \boldsymbol{W})$ | expected reward given $\mathbf{x}$ |
| $r_t$ | $t$-th step reward |
| $r_{t:}$ | $\sum_{\tau=t}^{T} r_\tau(\mathbf{y}^{(\tau)}, \mathbf{y}^{(\tau-1)})$ |
| $\boldsymbol{W}^\star$ | optimal parameters (for RL or SFT) |
| $\gamma_{l(t)}^j$ | critical gradient component |
| $\mathcal{L}$ | population loss of SFT |
| $L_t$ | $t$-th step loss |
| $\eta$ | learning rate |
| $\mathcal{Y}^{(:t)}$ | trajectory space of $\mathbf{y}^{(:t)}$ |

2. **Q:** Is the analytical framework restricted to the complete-binary tree decomposition in Fig. 2a?

   **A:** No. The analytical framework, i.e., the separation of the critical gradient component, can be extended beyond the complete binary-tree topology to a more common serial CoT setting (App. I). In this serial CoT setting, the problem has the structure (Fig. 3)

   $$y_1 = \phi_2(x_{i_1}, x_{i_2}), \ y_j = \phi_2(y_{j-1}, x_{i_{j+1}}), \ j = 2, \ldots, k-1,$$

   where $\mathbf{x}$ is the input, $\mathbf{y}$ is a serial CoT, and we do not require $k = 2^T$ for some integer $T$. For the transformer, we can replace the current tree mask with a more flexible and weaker serial mask that, when generating $y_j$, allows the model to attend to all input bits of $\mathbf{x}$ but only to the most recent CoT step $y_{j-1}$. This matches the serial recursion and no longer strictly enforces the exact binary tree in Fig. 2a. In this setting, the same proof strategy can be adapted, and the same qualitative learning dynamics can be recovered. Please refer to App. I for more details.

3. **Q:** Does this paper study both RL with final (outcome-based) reward and process reward?

   **A:** This paper mainly focuses on RL with process reward and provides the corresponding learnability result (Thm. 3.1). For RL with final reward, this paper only provides a negative result for learning $k$-PARITY (Prop. 3.2).

4. **Q:** Does this paper study both SFT with teacher forcing and without teacher forcing?

   **A:** This paper mainly focuses on SFT without teacher forcing to address the mismatch between the training and inference (Thm. 3.3), i.e., at training the model only has correct prior steps, while at inference it must condition on its own generated steps.

   But the results can be easily generalized to SFT with teacher forcing: under teacher forcing, each CoT step is trained with the correct previous steps; therefore, the same separation condition in Thm. 3.3 is satisfied for all CoT steps

simultaneously, so all steps can receive useful gradient in the same update and thus SFT with teacher forcing learns all steps simultaneously as RL with process reward.

5. **Q:** Beyond the comparison of the learning dynamics of RL with process reward and that of SFT (with or without) teacher forcing, what are the additional insights for RL vs. SFT?

   **A:** We suggest that the comparison might not be a pure RL-vs.-SFT: changing teacher forcing affects learning dynamics of SFT, and changing supervision reward affects learning dynamics of RL. Thus, the empirical comparisons between RL and SFT might conflate optimization method with these additional factors (e.g., reward design and teacher forcing), rather than isolating RL vs. SFT alone. Therefore, one should control for whether supervision is outcome- or process-based and whether training uses perfect prefixes or model-generated prefixes when comparing RL and SFT.

6. **Q:** What are the connections to more realistic and general cases?

   **A:** Our core intuition is qualitatively consistent with prior empirical observations on process supervision. Specifically, our theory shows that process reward can provide informative learning signal across the reasoning chain, and thus RL does not need to wait until earlier steps are fully learned before improving later ones. Empirically, Uesato et al. (2022) studied process- and outcome-based rewards for mathematical reasoning and showed that low reasoning-trajectory error requires process-based feedback. Lightman et al. (2023) explained this advantage through precise credit assignment: process supervision specifies the exact location of any errors that occur. These observations are qualitatively consistent with the same underlying intuition, i.e., process reward provides a step-level learning signal over multiple parts of the chain.

7. **Q:** What are the main stylized assumptions?

   **A:** We use a one-layer transformer where: (1) we merge the key and query matrices, $\boldsymbol{W}_K$ and $\boldsymbol{W}_Q$, into a single matrix $\boldsymbol{W}_{KQ}$, which is a common choice in this line of works; (2) the merged matrix $\boldsymbol{W}_{KQ}$ has a special formulation (Page 4, below Eq. (2)) that focuses on the position of the sequence, so that the self-attention is mainly selecting the most important positions from the sequence when generating a token; (3) the feedforward layer only has a non-linear activation function, which is determined by the target $k$-sparse Boolean function $\Phi(\cdot)$ (Tab. 1); and (4) we apply a "pretrained" mask for the transformer which serves as a structural prior and, in essence, abstractly captures the idea that pretraining provides priors for fine-tuning (Fig. 2b).

8. **Q:** What is the key part of the theory?

   **A:** the key part of our theory is not the binary-tree structure, but the fact that the gradient signal can distinguish relevant nodes from irrelevant ones, which can be captured through the separation of the critical gradient component—the most informative part of gradient signal. For example, in the extension to a more common serial CoT setting, the topology changes but the relevant nodes of each step are still distinguishable from irrelevant ones through the gradient signal, and thus the learnability results can also be established. In this sense, the separation condition is the formal way of expressing that the informative part of the gradient is stronger on the relevant nodes than on the irrelevant ones.

## C. Related Works

**Transformers with CoT.** Recently, the success of CoT reasoning in transformer models has attracted a lot of attention. Along this direction, a series of existing works have investigated the improvement of transformer expressiveness by providing CoT (Merrill & Sabharwal, 2024; Chen et al., 2024a; Li et al., 2024), while some other works aimed to reveal the inherent limitations (Barceló et al., 2025; Amiri et al., 2025). One crucial aspect of analyzing the transformers with CoT is optimization dynamics (Huang et al., 2025; Kim & Suzuki, 2025), which typically focused on the single-head transformer. A recent work (Yang et al., 2025) generalized the analysis to multi-head transformers and showed that a one-layer transformer can learn symbolic multi-step reasoning aided by sufficient intermediate reasoning steps of CoT.

**Learning Boolean functions with transformers.** (Sparse) Boolean functions have been shown to be fundamentally hard for transformers to learn in an end-to-end manner. This can be attributed to the "simplicity bias" that encourages transformers to prefer low-degree functions (Vasudeva et al., 2025; Hahn & Rofin, 2024). However, when providing a "scratchpad" (CoT) as intermediate supervision, the problem is decomposed into easier sub-tasks and a one-layer transformer can learn the sparse PARITY in one gradient update via teacher forcing (Kim & Suzuki, 2025). For other sparse Boolean functions AND and OR, Hu et al. (2025) showed a similar result: a single-head softmax-attention cannot solve these sparse

Boolean functions without any additional supervision, while it is capable of learning them via teacher-forcing. Our work also focuses on the learnability of transformers for sparse Boolean functions. Compared to prior works, we take the first step to investigate the underlying mechanism of fine-tuning via RL, which has not been covered by prior works, and we relax several constraints of SFT such as teacher-forcing, allowing us to reveal a distinction between the learning behavior of RL and that of SFT. Wang et al. (2025) studied the learnability of $k$-fold compositional functions by transformers, focusing on how data difficulty affects training. They demonstrated that gradient-based learning (in an SFT manner) with curriculum or data mixture can enable efficient learning. The curriculum learning in Wang et al. (2025) is also stepwise, similar to our conclusion that SFT learns stepwise. The difference is that our stepwise learning behavior can emerge naturally in SFT from transformers, an intrinsic property that does not rely on external curriculum learning.

**Learning dynamics of transformers.** Learning dynamics is a fundamental aspect for deep learning. With the increasing importance of transformers, many existing works have investigated the learning dynamics of transformers across a wide range of tasks, especially the special in-context learning (Yang et al., 2024; Chen et al., 2024b; Huang et al., 2023; Zhang et al., 2023) and the corresponding neural scaling laws (Lyu et al., 2025). In this work, we also analyze the learning dynamics of transformers, while our focus is fine-tuning via RL or SFT rather than in-context learning, which stands at the core of our characterization of the corresponding learnability.

## D. Numerical Experiments

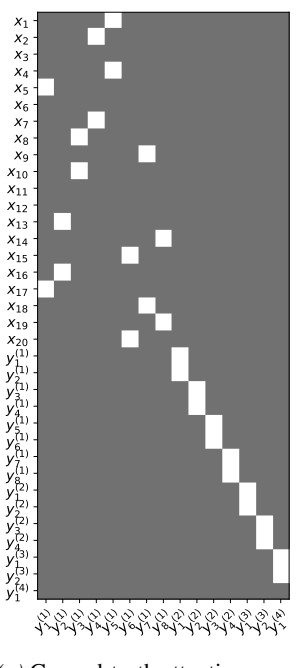

*(a)* Ground-truth attention score

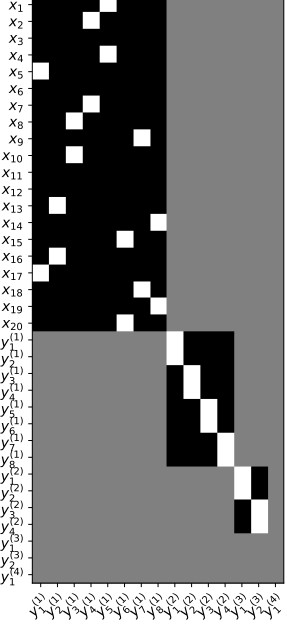

*(b)* Sign of policy gradient

*Figure 6.* **(a)** The ground truth $(\sigma^\star)_{N_{t-1}+l^{(t)}}^{N_t-2+p}$. Each white box is 0.5 and each gray box is 0. **(b)** $\mathrm{sign}(\nabla_{\boldsymbol{W}}\mathcal{R}(\boldsymbol{W}))$ at $\boldsymbol{W}(0) = \mathbf{1}$. Each white box has value $+1$ and each black box has value $-1$. Gray boxes have value 0 coming from causal mask and pretrained mask.

We conduct straightforward numerical experiments to empirically verify our theoretical claims. We specifically consider $k$-PARITY. To learn it, we use a one-layer transformer $f(\cdot; \boldsymbol{W})$ (with $\boldsymbol{W} = \mathbf{1}$ at initialization) as specified in Sec. 2.2. The input data $\mathbf{x} \in \{-1, +1\}^d$ and we construct a random subset $B \subseteq [d]$ with $|B| = k$ for the $k$-PARITY. We let $d = 20$ and $k = 16$; hence the CoT chain will have 4 steps, namely $\mathbf{y} = (\mathbf{y}^{(1)}, \mathbf{y}^{(2)}, \mathbf{y}^{(3)}, \mathbf{y}^{(4)})$. We uniformly sample $50,000$ samples of $\mathbf{x}$ to be our training dataset, where we also build the corresponding ground-truth reasoning sequence $\tilde{\mathbf{y}}$ for SFT as discussed in Sec. 3.2. In Fig. 6a, we plot the ground-truth attention score that can build $\tilde{\mathbf{y}}$, where each white box in a column $y_j^{(t)}$ denotes one child node of $y_j^{(t)}$ such that the attention score has value 0.5 at each white box and 0 at gray boxes. We omit the attention scores for $W_{i,j}$ with $j \leq d$ as we will not generate $\mathbf{x} \in \{-1, +1\}^d$.

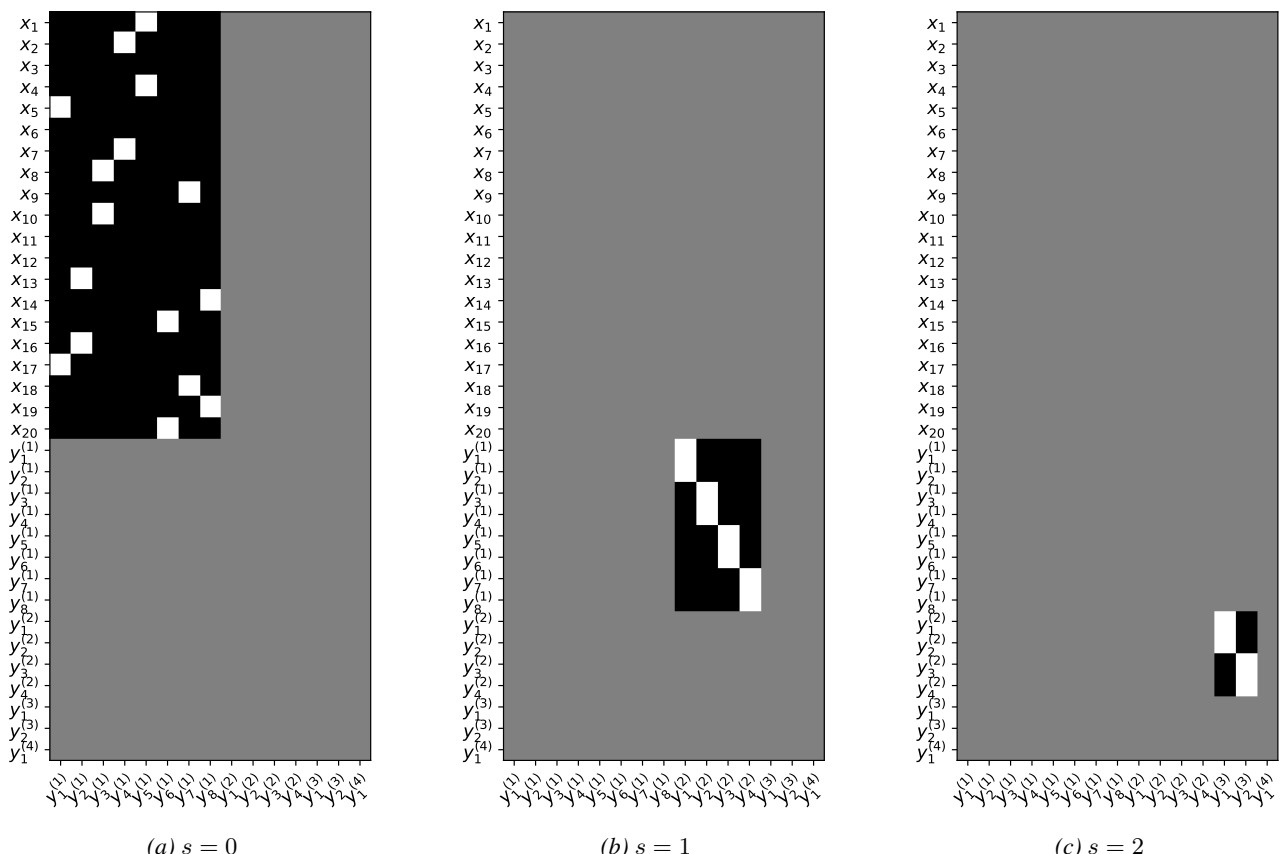

*(a) s = 0*             *(b) s = 1*             *(c) s = 2*

*Figure 7.* $\text{sign}(-\nabla_{\boldsymbol{W}} \mathcal{L}(\boldsymbol{W}))$ computed by $\boldsymbol{W}(s)$ for different updating step $s$. Each white box has value $+1$, each black box has value $-1$, and grey boxes are 0.

**RL fine-tuning.** We fine-tune $f(\cdot; \boldsymbol{W})$ via RL optimized by the sign of the policy gradient following Sec. 3.1. We plot $\text{sign}(\nabla_{\boldsymbol{W}}\mathcal{R}(\boldsymbol{W}))$ at $\boldsymbol{W}(0)$ in Fig. 6b. Compared to Fig. 6a, we can clearly see that $\nabla_{\boldsymbol{W}}\mathcal{R}(\boldsymbol{W})$ has positive values in all relevant positions and negative values in irrelevant positions, i.e., a distinct separation that enables the transformer to learn $k$-PARITY after one update.

**SFT.** In contrast to the one-update learning of RL, SFT exhibits a step-wise learning behavior, as shown in Fig. 7. We use $s$ to count the iteration number. In particular, the gradient $\nabla_{\boldsymbol{W}}\mathcal{L}(\boldsymbol{W})$ only has nonzero values for the first reasoning step $\mathbf{y}^{(1)}$ at $s = 0$, where it has positive values at relevant positions of each token of $\mathbf{y}^{(1)}$ and negative values at irrelevant positions. Hence SFT can and only can learn the first CoT step at the first update of sign gradient descent. Similarly, the transformer can and only can learn $\mathbf{y}^{(2)}$ at $s = 1$, etc.

## E. Proofs of Section 3.1

### E.1. Formulation of the Policy Gradient

**Additional notation.** For the whole CoT reasoning sequence $\mathbf{y}$, we denote its trajectory space by $\mathcal{Y}^{(:T)}$, and the trajectory space of $\mathbf{y}^{(:t)}$ by $\mathcal{Y}^{(:t)}$.

**Lemma E.1** (Formulation of the policy gradient). *The gradient for the expected reward*

$$\mathcal{R}(\boldsymbol{W}) := \underset{\mathbf{x}}{\mathbb{E}}\left[R(\mathbf{x}, \boldsymbol{W})\right] := \underset{\mathbf{x}}{\mathbb{E}}\left[\underset{\mathbf{y} \sim p_{\boldsymbol{W}}(\cdot|\mathbf{x})}{\mathbb{E}}\left[\sum_{t=1}^{T} r_t\left(\mathbf{y}^{(t)}, \mathbf{y}^{(t-1)}\right)\right]\right] \tag{13}$$

*is computed by*

$$\nabla_{\boldsymbol{W}} R(\mathbf{x}; \boldsymbol{W}) = \underset{\mathbf{y} \sim p_{\boldsymbol{W}}(\cdot|\mathbf{x})}{\mathbb{E}}\left[\sum_{t=1}^{T} \nabla_{\boldsymbol{W}} \ln p_{\boldsymbol{W}}\left(\mathbf{y}^{(t)}|\mathbf{y}^{(t-1)}\right) \sum_{\tau=t}^{T} r_\tau\left(\mathbf{y}^{(\tau)}, \mathbf{y}^{(\tau-1)}\right)\right]. \tag{14}$$

*Proof.* According to the definition of $R(\mathbf{x}; \boldsymbol{W})$, we can write its gradient as

$$\nabla_{\boldsymbol{W}} R(\mathbf{x}; \boldsymbol{W})$$
$$= \sum_{\mathbf{y} \in \mathcal{Y}^{(:T)}} \nabla_{\boldsymbol{W}} p_{\boldsymbol{W}}(\mathbf{y}|\mathbf{x}) \sum_{\tau=1}^{T} r_\tau\left(\mathbf{y}^{(\tau)}, \mathbf{y}^{(\tau-1)}\right) \tag{15}$$
$$= \sum_{t=1}^{T} \sum_{\mathbf{y} \in \mathcal{Y}^{(:T)}} p_{\boldsymbol{W}}(\mathbf{y}|\mathbf{x}) \nabla_{\boldsymbol{W}} \ln p_{\boldsymbol{W}}\left(\mathbf{y}^{(t)}|\mathbf{y}^{(t-1)}\right) \left(r_{:t-1}(\mathbf{y}^{(:t-1)}) + r_{t:}(\mathbf{y}^{(t-1:)})\right),$$

where we apply

$$\nabla_{\boldsymbol{W}} p_{\boldsymbol{W}}(\mathbf{y}|\mathbf{x}) = p_{\boldsymbol{W}}(\mathbf{y}|\mathbf{x}) \nabla_{\boldsymbol{W}} \ln p_{\boldsymbol{W}}(\mathbf{y}|\mathbf{x})$$

and $p_{\boldsymbol{W}}(\mathbf{y}|\mathbf{x}) = \prod_{t=1}^{T} p_{\boldsymbol{W}}\left(\mathbf{y}^{(t)}|\mathbf{y}^{(t-1)}\right)$ with $\mathbf{y}^{(0)} = \mathbf{x}$ in the second equality, and we define

$$r_{:t}(\mathbf{y}^{(:t)}) := \sum_{\tau=1}^{t} r_\tau\left(\mathbf{y}^{(\tau)}, \mathbf{y}^{(\tau-1)}\right),$$
$$r_{t:}(\mathbf{y}^{(t-1:)}) := \sum_{\tau=t}^{T} r_\tau\left(\mathbf{y}^{(\tau)}, \mathbf{y}^{(\tau-1)}\right). \tag{16}$$

For any $t \in [T]$, we can derive

$$\sum_{\mathbf{y} \in \mathcal{Y}^{(:T)}} p_{\boldsymbol{W}}(\mathbf{y}|\mathbf{x}) \nabla_{\boldsymbol{W}} \ln p_{\boldsymbol{W}}\left(\mathbf{y}^{(t)}|\mathbf{y}^{(t-1)}\right) r_{:t-1}(\mathbf{y}^{(:t-1)})$$

$$= \sum_{\mathbf{y}^{(:t)} \in \mathcal{Y}^{(:t)}} p_{\boldsymbol{W}}\left(\mathbf{y}^{(:t)}|\mathbf{x}\right) \nabla_{\boldsymbol{W}} \ln p_{\boldsymbol{W}}\left(\mathbf{y}^{(t)}|\mathbf{y}^{(t-1)}\right) r_{:t-1}(\mathbf{y}^{(:t-1)}) \left[ \sum_{\mathbf{y}^{(t+1:)} \in \mathcal{Y}^{(:t+1)}} p_{\boldsymbol{W}}\left(\mathbf{y}^{(t+1:)}|\mathbf{y}^{(:t)}\right) \right]$$

$$= \sum_{\mathbf{y}^{(:t-1)} \in \mathcal{Y}^{(:t-1)}} p_{\boldsymbol{W}}\left(\mathbf{y}^{(:t-1)}|\mathbf{x}\right) r_{:t-1}(\mathbf{y}^{(:t-1)}) \sum_{\mathbf{y}^{(t)} \in \mathcal{Y}^{(t)}} p_{\boldsymbol{W}}\left(\mathbf{y}^{(t)}|\mathbf{y}^{(t-1)}\right) \nabla_{\boldsymbol{W}} \ln p_{\boldsymbol{W}}\left(\mathbf{y}^{(t)}|\mathbf{y}^{(t-1)}\right) \tag{17}$$

$$= \sum_{\mathbf{y}^{(:t-1)} \in \mathcal{Y}^{(:t-1)}} p_{\boldsymbol{W}}\left(\mathbf{y}^{(:t-1)}|\mathbf{x}\right) r_{:t-1}(\mathbf{y}^{(:t-1)}) \nabla_{\boldsymbol{W}} \left[ \sum_{\mathbf{y}^{(t)} \in \mathcal{Y}^{(t)}} p_{\boldsymbol{W}}\left(\mathbf{y}^{(t)}|\mathbf{y}^{(t-1)}\right) \right] = 0,$$

where we apply $\sum_{\mathbf{y}^{(t+1:)}} p_{\boldsymbol{W}}\left(\mathbf{y}^{(t+1:)}|\mathbf{y}^{(:t)}\right) = 1$ in the second equality. Thus, Eq. (15) only has the term $r_{t:}$ and the claim of the lemma is established. $\square$

### E.2. Separation of the Critical Gradient Component

For convenience, we recall that the critical gradient component is defined by

$$\textbf{Critical gradient component: } \gamma_{l^{(t)}}^{j}(\mathbf{y}^{(t-1)}) := \frac{2}{k-1} \psi'\left(\xi_{l^{(t)}}\right) \phi_2\left(y_{i_1^{l^{(t)}}}^{(t-1)}, y_{i_2^{l^{(t)}}}^{(t-1)}\right) y_j^{(t-1)}. \tag{18}$$

Below we discuss the relation between the policy gradient Eq. (6), and the critical gradient component and the equivalence between their separations.

**Lemma E.2.** $\forall t \in [T], l^{(t)} \in [d_t]$, let $i_1^{l^{(t)}}$ and $i_2^{l^{(t)}}$ be two child nodes of the $l^{(t)}$-th node of $\mathbf{y}^{(t)}$, then the policy gradient $\mathbb{E}_{\mathbf{x}}[\nabla_{\boldsymbol{W}} R(\mathbf{x}; \boldsymbol{W})]$ Eq. (6) can be expressed by the critical gradient component as $(p \in [d_{t-1}])$

$$\partial_{W_{N_{t-2}+p, N_{t-1}+l^{(t)}}} R(\mathbf{x}; \boldsymbol{W})$$

$$= \mathbb{E}_{\mathbf{y}^{(:t-1)} \sim p_{\boldsymbol{W}}(\cdot|\mathbf{x})} \left[ \gamma_{l^{(t)}}^{p}(\mathbf{y}^{(t-1)}) - \sum_{i=1}^{d_{t-1}} \sigma_{N_{t-1}+l^{(t)}}^{N_{t-2}+i} \gamma_{l^{(t)}}^{i}(\mathbf{y}^{(t-1)}) \right] \sigma_{N_{t-1}+l^{(t)}}^{N_{t-2}+p}. \tag{19}$$

Furthermore, if $\boldsymbol{W} = c\mathbf{1}$ for an arbitrary constant $c$, the separation of the policy gradient $\forall p \in \{i_1^{l^{(t)}}, i_2^{l^{(t)}}\}$, $p' \in [d_{t-1}] \backslash \{i_1^{l^{(t)}}, i_2^{l^{(t)}}\}$:

$$\mathbb{E}_{\mathbf{x}}\left[ \partial_{W_{N_{t-2}+p, N_{t-1}+l^{(t)}}} R(\mathbf{x}; \boldsymbol{W}) - \partial_{W_{N_{t-2}+p', N_{t-1}+l^{(t)}}} R(\mathbf{x}; \boldsymbol{W}) \right] > 0 \tag{20}$$

is equivalent to the separation of the critical gradient component $\forall p \in \{i_1^{l^{(t)}}, i_2^{l^{(t)}}\}$, $p' \in [d_{t-1}] \backslash \{i_1^{l^{(t)}}, i_2^{l^{(t)}}\}$ :

$$\mathbb{E}_{\mathbf{x}, \mathbf{y}^{(:t-1)} \sim p_{\boldsymbol{W}}(\cdot|\mathbf{x})}\left[ \gamma_{l^{(t)}}^{p}(\mathbf{y}^{(t-1)}) - \gamma_{l^{(t)}}^{p'}(\mathbf{y}^{(t-1)}) \right] > 0. \tag{21}$$

*Proof.* To prove this claim, we start from analyzing the detailed formulation of the policy gradient with all components written explicitly. Specifically, given $t \in [T]$, only the components with $p \in [d_{t-1}]$ and $l^{(t)} \in [d_t]$,

$$W_{N_{t-2}+p, N_{t-1}+l^{(t)}} \neq -\infty \tag{22}$$

given the pretrained mask discussed in Sec. 2.2. As a result, we only need to analyze the policy gradient for these components.

According to

$$\partial_{W_{N_{t-2}+p,N_{t-1}+l^{(t)}}} R(\mathbf{x};\boldsymbol{W})$$

$$\overset{(i)}{=} \sum_{\mathbf{y}\in\mathcal{Y}^{(:t)}} p_{\boldsymbol{W}}(\mathbf{y}^{(:t-1)}|\mathbf{x}) p_{\boldsymbol{W}}(\mathbf{y}^{(t+1:)}|\mathbf{y}^{(t)}) \partial_{W_{N_{t-2}+p,N_{t-1}+l^{(t)}}} p_{\boldsymbol{W}}(\mathbf{y}^{(t)}|\mathbf{y}^{(t-1)}) r_t(\mathbf{y}^{(t)},\mathbf{y}^{(t-1)})$$

$$= \sum_{\mathbf{y}^{(:t-1)}\in\mathcal{Y}^{(:t-1)}} p_{\boldsymbol{W}}(\mathbf{y}^{(:t-1)}|\mathbf{x}) \sum_{\mathbf{y}^{(t:)}\in\mathcal{Y}^{(t:)}} \partial_{W_{N_{t-2}+p,N_{t-1}+l^{(t)}}} p_{\boldsymbol{W}}(\mathbf{y}^{(t)}|\mathbf{y}^{(t-1)}) r_t(\mathbf{y}^{(t)},\mathbf{y}^{(t-1)}) p_{\boldsymbol{W}}(\mathbf{y}^{(t+1:)}|\mathbf{y}^{(t)})$$

$$= \sum_{\mathbf{y}^{(:t-1)}\in\mathcal{Y}^{(:t-1)}} p_{\boldsymbol{W}}(\mathbf{y}^{(:t-1)}|\mathbf{x})$$

$$\times \left[ \sum_{\mathbf{y}^{(t)}\in\mathcal{Y}^{(t)}} \partial_{W_{N_{t-2}+p,N_{t-1}+l^{(t)}}} p_{\boldsymbol{W}}(\mathbf{y}^{(t)}|\mathbf{y}^{(t-1)}) r_t(\mathbf{y}^{(t)},\mathbf{y}^{(t-1)}) \sum_{\mathbf{y}^{(t+1:)}\in\mathcal{Y}^{(t+1:)}} p_{\boldsymbol{W}}(\mathbf{y}^{(t+1:)}|\mathbf{y}^{(t)}) \right]$$

$$\overset{(ii)}{=} \mathbb{E}_{\mathbf{y}^{(:t-1)}\sim p_{\boldsymbol{W}}(\cdot|\mathbf{x})} \left[ \sum_{\mathbf{y}^{(t)}\in\mathcal{Y}^{(t)}} \partial_{W_{N_{t-2}+p,N_{t-1}+l^{(t)}}} p_{\boldsymbol{W}}(\mathbf{y}^{(t)}|\mathbf{y}^{(t-1)}) r_t(\mathbf{y}^{(t)},\mathbf{y}^{(t-1)}) \right], \tag{23}$$

where $(i)$ is a result of the definition of $\nabla_{\boldsymbol{W}} R(\mathbf{x};\boldsymbol{W})$, the fact that each $\mathbf{y}^{(t)}$ only depends on $\mathbf{y}^{(t-1)}$ due to the pretrained mask

$$p_{\boldsymbol{W}}(\mathbf{y}|\mathbf{x}) = p_{\boldsymbol{W}}(\mathbf{y}^{(:t-1)}|\mathbf{x}) p_{\boldsymbol{W}}(\mathbf{y}^{(t:)}|\mathbf{x},\mathbf{y}^{(:t-1)}) = p_{\boldsymbol{W}}(\mathbf{y}^{(:t-1)}|\mathbf{x}) p_{\boldsymbol{W}}(\mathbf{y}^{(t:)}|\mathbf{y}^{(t-1)}),$$

and that $W_{N_{t-2}+i,N_{t-1}+l^{(t)}}$ only determines $p_{\boldsymbol{W}}(\mathbf{y}^{(t)}|\mathbf{y}^{(t-1)})$; $(ii)$ is because the law of total probability. Now, as the formulation of the transformer implies Eq. (3) that

$$p_{\boldsymbol{W}}(y_{l^{(t)}}^{(t)} = 1|\mathbf{y}^{(t-1)}) = [f(\mathbf{x},\mathbf{y};\boldsymbol{W})]_{N_{t-1}+l^{(t)}} = \psi\left(\xi_{l^{(t)}}\right) \tag{24}$$

with $\xi_{l^{(t)}} := \sum_{i=1}^{d_{t-1}} y_i^{(t-1)} \sigma_{N_{t-1}+l^{(t)}}^{N_{t-2}+i}$, we need the gradient of the attention score $\sigma_i^j$, which is computed as

$$\partial_{W_{i,j}} \sigma_n^m = \delta_{jn}(\delta_{im} - \sigma_j^i)\sigma_n^m. \tag{25}$$

This gives us (with some straightforward algebra)

$$\partial_{W_{N_{t-2}+p,N_{t-1}+l^{(t)}}} p_{\boldsymbol{W}}(y_{l^{(t)}}^{(t)} = 1|\mathbf{y}^{(t-1)})$$

$$= \psi'(\xi_{l^{(t)}}) \sum_{i=1}^{d_{t-1}} y_i^{(t-1)} \partial_{W_{N_{t-2}+p,N_{t-1}+l^{(t)}}} \sigma_{N_{t-1}+l^{(t)}}^{N_{t-2}+i}$$

$$= \psi'(\xi_{l^{(t)}}) \sum_{i=1}^{d_{t-1}} y_i^{(t-1)} \left( \delta_{ip} - \sigma_{N_{t-1}+l^{(t)}}^{N_{t-2}+p} \right) \sigma_{N_{t-1}+l^{(t)}}^{N_{t-2}+i} \tag{26}$$

$$= \psi'(\xi_{l^{(t)}}) \left( y_p^{(t-1)} - \sum_{i=1}^{d_{t-1}} y_i^{(t-1)} \sigma_{N_{t-1}+l^{(t)}}^{N_{t-2}+i} \right) \sigma_{N_{t-1}+l^{(t)}}^{N_{t-2}+p}.$$

On the other hand, by using the fact that $p_{\boldsymbol{W}}(y_{l^{(t)}}^{(t)} = -1|\mathbf{y}^{(t-1)}) = 1 - p_{\boldsymbol{W}}(y_{l^{(t)}}^{(t)} = 1|\mathbf{y}^{(t-1)})$, we are able to conclude that

$$\partial_{W_{N_{t-2}+p,N_{t-1}+l^{(t)}}} p_{\boldsymbol{W}}(y_{l^{(t)}}^{(t)} = -1|\mathbf{y}^{(t-1)})$$

$$= -\psi'(\xi_{l^{(t)}}) \left( y_p^{(t-1)} - \sum_{i=1}^{d_{t-1}} y_i^{(t-1)} \sigma_{N_{t-1}+l^{(t)}}^{N_{t-2}+i} \right) \sigma_{N_{t-1}+l^{(t)}}^{N_{t-2}+p}. \tag{27}$$

Thus, we can summarize the gradient as

$$\partial_{W_{N_{t-2}+p,N_{t-1}+l^{(t)}}} p_{\boldsymbol{W}}(y_{l^{(t)}}^{(t)}|\mathbf{y}^{(t-1)})$$

$$= \psi'(\xi_{l^{(t)}}) \left( y_p^{(t-1)} - \sum_{i=1}^{d_{t-1}} y_i^{(t-1)} \sigma_{N_{t-1}+l^{(t)}}^{N_{t-2}+i} \right) \sigma_{N_{t-1}+l^{(t)}}^{N_{t-2}+p} y_{l^{(t)}}^{(t)}. \tag{28}$$

**Expressing the policy gradient with the critical gradient component.** We now express the policy gradient using the critical gradient component to prove the first claim of the lemma. Applying Eq. (28), we are able to rewrite Eq. (23) as (recall that all tokens $y_j^{(t)}$ for $j \in [d_{t-1}]$ of the sequence $\mathbf{y}^{(t)}$ are independent of each other hence $p_{\mathbf{W}}(\mathbf{y}^{(t)}|\mathbf{y}^{(t-1)}) = \prod_{l=1}^{d_t} p_{\mathbf{W}}(y_l^{(t)}|\mathbf{y}^{(t-1)})$):

$$
\sum_{\mathbf{y}^{(t)} \in \mathcal{Y}^{(t)}} \partial_{W_{N_{t-2}+p, N_{t-1}+l^{(t)}}} p_{\mathbf{W}}(\mathbf{y}^{(t)}|\mathbf{y}^{(t-1)}) r_t(\mathbf{y}^{(t)}, \mathbf{y}^{(t-1)})
$$

$$
= [\psi'(\xi_{l^{(t)}})] \left( y_p^{(t-1)} - \sum_{i=1}^{d_{t-1}} y_i^{(t-1)} \sigma_{N_{t-1}+l^{(t)}}^{N_{t-2}+i} \right) \sigma_{N_{t-1}+l^{(t)}}^{N_{t-2}+p} \sum_{\mathbf{y}^{(t)} \in \mathcal{Y}^{(t)}} \frac{p_{\mathbf{W}}(\mathbf{y}^{(t)}|\mathbf{y}^{(t-1)})}{p_{\mathbf{W}}(y_{l^{(t)}}^{(t)}|\mathbf{y}^{(t-1)})} r_t(\mathbf{y}^{(t)}, \mathbf{y}^{(t-1)}) y_{l^{(t)}}^{(t)}
$$

$$
\overset{(i)}{=} \frac{2}{k-1} [\psi'(\xi_{l^{(t)}})] \left( y_p^{(t-1)} - \sum_{i=1}^{d_{t-1}} y_i^{(t-1)} \sigma_{N_{t-1}+l^{(t)}}^{N_{t-2}+i} \right) \sigma_{N_{t-1}+l^{(t)}}^{N_{t-2}+p} \bar{y}_{l^{(t)}}^{(t)}
$$

$$
\overset{(ii)}{=} \frac{2}{k-1} \left[ \psi'(\xi_{l^{(t)}}) \phi \left( y_{i_1^{l^{(t)}}}^{(t-1)}, y_{i_2^{l^{(t)}}}^{(t-1)} \right) \right] \left( y_p^{(t-1)} - \sum_{i=1}^{d_{t-1}} y_i^{(t-1)} \sigma_{N_{t-1}+l^{(t)}}^{N_{t-2}+i} \right) \sigma_{N_{t-1}+l^{(t)}}^{N_{t-2}+p}
$$

$$
\overset{(iii)}{=} \left( \gamma_{l^{(t)}}^p(\mathbf{y}^{(t-1)}) - \sum_{i=1}^{d_{t-1}} \gamma_{l^{(t)}}^i(\mathbf{y}^{(t-1)}) \sigma_{N_{t-1}+l^{(t)}}^{N_{t-2}+i} \right) \sigma_{N_{t-1}+l^{(t)}}^{N_{t-2}+p}, \tag{29}
$$

where, denoting $\mathbf{y}_{l'}^{(t)} = (y_1^{(t)}, \ldots, y_{l-1}^{(t)}, y_{l+1}^{(t)}, \ldots, y_{d_t}^{(t)})$ as $\mathbf{y}^{(t)}$ without the $l$-th token, the equality $(i)$ is because

$$
\sum_{\mathbf{y}^{(t)} \in \mathcal{Y}^{(t)}} \frac{p_{\mathbf{W}}(\mathbf{y}^{(t)}|\mathbf{y}^{(t-1)})}{p_{\mathbf{W}}(y_{l^{(t)}}^{(t)}|\mathbf{y}^{(t-1)})} r_t(\mathbf{y}^{(t)}, \mathbf{y}^{(t-1)}) y_{l^{(t)}}^{(t)}
$$

$$
= \frac{1}{k-1} \sum_{\mathbf{y}^{(t)} \in \mathcal{Y}^{(t)}} p_{\mathbf{W}}(\mathbf{y}_{l'^{(t)}}^{(t)}|\mathbf{y}^{(t-1)}) \left( \bar{y}_{l^{(t)}}^{(t)} + \sum_{l \neq l^{(t)}} y_l^{(t)} \bar{y}_l^{(t)} y_{l^{(t)}}^{(t)} \right)
$$

$$
= \frac{2}{k-1} \bar{y}_{l^{(t)}}^{(t)} + \frac{1}{k-1} \left( \sum_{y_{l^{(t)}}^{(t)} \in \{-1, +1\}} y_{l^{(t)}}^{(t)} \right) \sum_{\mathbf{y}_{l'^{(t)}}^{(t)}} p_{\mathbf{W}}(\mathbf{y}_{l'^{(t)}}^{(t)}|\mathbf{y}^{(t-1)}) \sum_{l \neq l^{(t)}} y_l^{(t)} \bar{y}_l^{(t)}
$$

$$
= \frac{2}{k-1} \bar{y}_{l^{(t)}}^{(t)}; \tag{30}
$$

the equality $(ii)$ applies the definition of $\bar{y}$; the equality $(iii)$ applies the definition of the critical gradient component

$$
\gamma_{l^{(t)}}^p(\mathbf{y}^{(t-1)}) := \frac{2}{k-1} \psi'(\xi_{l^{(t)}}) \phi_2 \left( y_{i_1^{l^{(t)}}}^{(t-1)}, y_{i_2^{l^{(t)}}}^{(t-1)} \right) y_p^{(t-1)}. \tag{31}
$$

Now, inserting Eq. (29) back to Eq. (23) proves the first claim of the lemma Eq. (19).

**Establishing the equivalence between the separations.** Now we establish the equivalence between the separation of the policy gradient and that of the critical gradient component. Under the setting of the lemma we can assume $\mathbf{W} = \mathbf{1}$ w.l.g such that $\sigma_{N_{t-1}+i}^{N_{t-2}+j} = 1/d_{t-1}$, then Eq. (29) implies that

$$
\partial_{W_{N_{t-2}+p, N_{t-1}+l^{(t)}}} R(\mathbf{x}; \mathbf{W}) - \partial_{W_{N_{t-2}+p', N_{t-1}+l^{(t)}}} R(\mathbf{x}; \mathbf{W})
$$

$$
= \frac{1}{d_{t-1}} \mathbb{E}_{\mathbf{y}^{(:t-1)} \sim p_{\mathbf{W}}(\cdot|\mathbf{x})} \left[ \gamma_{l^{(t)}}^p(\mathbf{y}^{(t-1)}) - \gamma_{l^{(t)}}^{p'}(\mathbf{y}^{(t-1)}) \right]. \tag{32}
$$

Hence, the separation of the critical gradient component is equivalent to the separation of the policy gradient as stated in the lemma. $\qquad \square$

### E.3. Proof of Theorem 3.1

For convenience, we restate Thm. 3.1 below.

**Theorem E.3** (Restated Thm. 3.1). *Given integers $d \geq k \geq 2$, consider a $k$-sparse Boolean function $\Phi_k(\cdot)$ with any subset $B \subseteq [d]$ as in Def. 2.1. Let $\boldsymbol{W}(0) = \boldsymbol{1}$ be the initialization and let $\boldsymbol{W}^\star = \arg\max_{\boldsymbol{W}} \mathcal{R}(\boldsymbol{W})$ be the optimal parameter that solves $\max_{\boldsymbol{W}} \mathcal{R}(\boldsymbol{W})$. Set learning rate $\eta = \Omega\left(\ln(d/\epsilon)\right)$ for any $\epsilon > 0$. If the separation of the critical gradient component $\gamma_{l^{(t)}}^p$ is satisfied for $\forall t \in [T], l^{(t)} \in [d_t]$ and $\forall p \in \{i_1^{l^{(t)}}, i_2^{l^{(t)}}\}$, $p' \in [d_{t-1}]\backslash\{i_1^{l^{(t)}}, i_2^{l^{(t)}}\}$ :*

$$\mathbb{E}_{\mathbf{x}, \mathbf{y}^{(:t-1)} \sim p_{\boldsymbol{W}}(\cdot|\mathbf{x})}\left[\gamma_{l^{(t)}}^p(\mathbf{y}^{(t-1)}) - \gamma_{l^{(t)}}^{p'}(\mathbf{y}^{(t-1)})\right] > 0 \tag{33}$$

*($p$ is a child node of $y_{l^{(t)}}^{(t)}$ while $p'$ is not), then fine-tuning the transformer $f(\cdot; \boldsymbol{W})$ via RL optimized by the sign of the policy gradient Eq. (6) after one update*

$$\boldsymbol{W}(1) = \boldsymbol{W}(0) + \eta \operatorname{sign}\left(\nabla_{\boldsymbol{W}} \mathcal{R}(\mathbf{x}; \boldsymbol{W})\right)$$

*achieves*

$$\|\operatorname{softmax}(\boldsymbol{W}(1)) - \operatorname{softmax}(\boldsymbol{W}^\star)\|_1 \leq \epsilon.$$

*Proof.* We first present a sketch.

**Proof sketch.** The proof follows a two step procedure: (i) we show that the separation of the critical gradient component implies that the policy gradient has positive values at relevant positions and negative values for all irrelevant positions; (ii) then we show that one update of the policy gradient is sufficient for the transformer to have low error. We now present the proof.

**Step (i): separation of $\gamma_{l^{(t)}}^p$ implies positive gradient at relevant positions.** First of all, Eq. (29) allows us to derive a simple relation

$$\sum_{p=1}^{d_{t-1}} \partial_{W_{N_{t-2}+p, N_{t-1}+l^{(t)}}} R(\mathbf{x}; \boldsymbol{W}) = 0 \tag{34}$$

by conducting some simple algebra. Furthermore, at initialization $\boldsymbol{W} = c\boldsymbol{1}$ and w.l.g we assume $c = 1$. If the separation of the critical gradient $\forall t \in [T], l^{(t)} \in [d_t]$ and $\forall p \in \{i_1^{l^{(t)}}, i_2^{l^{(t)}}\}$, $p' \in [d_{t-1}]\backslash\{i_1^{l^{(t)}}, i_2^{l^{(t)}}\}$ :

$$\mathbb{E}_{\mathbf{x}, \mathbf{y}^{(:t-1)} \sim p_{\boldsymbol{W}}(\cdot|\mathbf{x})}\left[\gamma_{l^{(t)}}^p(\mathbf{y}^{(t-1)}) - \gamma_{l^{(t)}}^{p'}(\mathbf{y}^{(t-1)})\right] > 0 \tag{35}$$

is satisfied, then Lem. E.2 suggests that

$$\mathbb{E}_{\mathbf{x}}\left[\partial_{W_{N_{t-2}+p, N_{t-1}+l^{(t)}}} R(\mathbf{x}; \boldsymbol{W}) - \partial_{W_{N_{t-2}+p', N_{t-1}+l^{(t)}}} R(\mathbf{x}; \boldsymbol{W})\right] > 0. \tag{36}$$

According to the symmetry of $i_1^{l^{(t)}}$ and $i_2^{l^{(t)}}$ and that of all other positions (i.e., $\gamma_{l^{(t)}}^{i_1^{l^{(t)}}} = \gamma_{l^{(t)}}^{i_2^{l^{(t)}}}$ and $\forall p \in [d_{t-1}]\backslash\{i_1^{l^{(t)}}, i_2^{l^{(t)}}\}$ : $\gamma_{l^{(t)}}^p = c_2$ for some constant $c_2$), we are able to write

$$\mathbb{E}_{\mathbf{x}}\left[\partial_{W_{N_{t-2}+p, N_{t-1}+l^{(t)}}} R(\mathbf{x}; \boldsymbol{W})\right] = \begin{cases} G_1, & p \in \{i_1^{l^{(t)}}, i_2^{l^{(t)}}\}, \\ G_2, & p \in [d_{t-1}]\backslash\{i_1^{l^{(t)}}, i_2^{l^{(t)}}\}, \end{cases} \tag{37}$$

using Eq. (29) where $G_1 > G_2$ according to Eq. (36). Now use Eq. (34), we obtain

$$2G_1 + (d_{t-1} - 2)G_2 = 0 \implies G_1 = -\frac{d_{t-1} - 2}{2}G_2. \tag{38}$$

Thus, we must have $G_1 > 0$ and $G_2 < 0$ since $d_{t-1} - 2 > 0$ while $G_1 > G_2$, i.e., the policy gradient has positive values at the relevant positions $p \in \{i_1^{l^{(t)}}, i_2^{l^{(t)}}\}$ (the child nodes of $l^{(t)}$-th token of $\mathbf{y}^{(t)}$) and negative values at irrelevant positions (all other nodes).

**Step (ii): one policy gradient update is sufficient.** We now directly use the sign of the policy gradient to update the model with learning rate $\eta > 0$:

$$W_{N_{t-2}+p, N_{t-1}+l^{(t)}}(1) = W_{N_{t-2}+p, N_{t-1}+l^{(t)}}(0) + \eta \, \text{sign} \left( \mathbb{E}_{\mathbf{x}} \left[ \partial_{W_{N_{t-2}+p, N_{t-1}+l^{(t)}}} R(\mathbf{x}; \boldsymbol{W}) \right] \right).$$

As the gradient has positive values at relevant positions and negative values at irrelevant positions, we conclude that after one-update of the policy gradient

$$W_{N_{t-2}+p, N_{t-1}+l^{(t)}}(1) = \begin{cases} 1 + \eta, & p \in \{i_1^{l^{(t)}}, i_2^{l^{(t)}}\}, \\ 1 - \eta, & p \in [d_{t-1}]\backslash\{i_1^{l^{(t)}}, i_2^{l^{(t)}}\}. \end{cases} \tag{39}$$

After the softmax we obtain

$$\forall t \in [T], l^{(t)} \in [d_t]: \; \sigma_{N_{t-1}+l^{(t)}}^{N_{t-2}+p}(1) = \begin{cases} \frac{1}{2} \frac{1}{1 + \frac{d_{t-1}-2}{2} e^{-2\eta}}, & p \in \{i_1^{l^{(t)}}, i_2^{l^{(t)}}\}, \\ \frac{1}{d_{t-1}-2+2e^{2\eta}}, & p \in [d_{t-1}]\backslash\{i_1^{l^{(t)}}, i_2^{l^{(t)}}\}. \end{cases} \tag{40}$$

On the other hand, the model parameter $\boldsymbol{W}^\star$ that solves $\max_{\boldsymbol{W}} \mathcal{R}(\boldsymbol{W})$ has the formulation of

$$\forall t \in [T], l^{(t)} \in [d_t]: \; (\sigma^\star)_{N_{t-1}+l^{(t)}}^{N_{t-2}+p} = \begin{cases} \frac{1}{2}, & p \in \{i_1^{l^{(t)}}, i_2^{l^{(t)}}\}, \\ 0, & p \in [d_{t-1}]\backslash\{i_1^{l^{(t)}}, i_2^{l^{(t)}}\}, \end{cases} \tag{41}$$

such that the model $f(\cdot; \boldsymbol{W}^\star)$ only attends to the relevant positions when generating $y_{l^{(t)}}^{(t)}$ as long as the expressivity of the transformer $f(\cdot; \boldsymbol{W}^\star)$ is guaranteed (i.e., it can solve the $k$-sparse Boolean functions $\Phi_k(\cdot)$ perfectly). Therefore, we can calculate

$$\forall t \in [T], l^{(t)} \in [d_t]: \sum_{p=1}^{d_{t-1}} \left| \sigma_{N_{t-1}+l^{(t)}}^{N_{t-2}+p}(1) - (\sigma^\star)_{N_{t-1}+l^{(t)}}^{N_{t-2}+p} \right| = \frac{1}{\frac{1}{2} + \frac{e^{2\eta}}{d_{t-1}-2}}, \tag{42}$$

which gives us

$$\|\text{softmax}(\boldsymbol{W}(1)) - \text{softmax}(\boldsymbol{W}^\star)\|_1 = \max_t \frac{1}{\frac{1}{2} + \frac{e^{2\eta}}{d_{t-1}-2}} = \frac{1}{\frac{1}{2} + \frac{e^{2\eta}}{d-2}}. \tag{43}$$

To make $\frac{1}{\frac{1}{2} + \frac{e^{2\eta}}{d-2}} \leq \epsilon$ for some given $\epsilon > 0$, we need to ensure

$$\eta \geq \frac{1}{2} \ln \left( \frac{(d-2)(2-\epsilon)}{2\epsilon} \right) \implies \eta = \Omega \left( \ln \frac{d}{\epsilon} \right). \tag{44}$$

This proves the claim. $\qquad\square$

### E.4. Hardness of RL with Final Reward

In this section, we prove Prop. 3.2 to show that the policy gradient contains negligible information of the objective so that it cannot tell relevant positions from irrelevant ones when using final reward.

*Proof.* We are interested in evaluating the variance

$$\text{Var}(\mathcal{H}; \boldsymbol{W}) := \mathbb{E}_{h \in \mathcal{H}} \left[ \left\| \nabla_{\boldsymbol{W}} \mathcal{R}_h^{\text{F}}(\boldsymbol{W}) - \mathbb{E}_{h' \in \mathcal{H}} [\nabla_{\boldsymbol{W}} \mathcal{R}_{h'}^{\text{F}}(\boldsymbol{W})] \right\|^2 \right]. \tag{45}$$

To derive an upper bound, it is sufficient to show that there exists some constant vector $\boldsymbol{a}$ and function $G(\mathcal{H})$ such that

$$\mathbb{E}_{h \in \mathcal{H}} \left[ \left\| \nabla_{\boldsymbol{W}} \mathcal{R}_h^{\text{F}}(\boldsymbol{W}) - \boldsymbol{a} \right\|^2 \right] \leq G(\mathcal{H}). \tag{46}$$

Below we construct such a vector $\boldsymbol{a}$. For this purpose, we first evaluate the formulation of the policy gradient (we use $\nabla$ to denote $\nabla_{\boldsymbol{W}}$)

$$
\begin{aligned}
&\nabla \mathcal{R}_h^{\mathrm{F}}(\boldsymbol{W}) \\
&= \mathbb{E}_{\mathbf{x}} \left[ \sum_{\mathbf{y} \in \mathcal{Y}^{(:T)}} \nabla p_{\boldsymbol{W}}(\mathbf{y}|\mathbf{x}) \mathbf{y}^{(T)} h(\mathbf{x}) \right] \\
&= \mathbb{E}_{\mathbf{x}} \left[ \nabla_{\boldsymbol{W}} \mathbb{E}_{\mathbf{y} \sim p_{\boldsymbol{W}}(\cdot|\mathbf{x})} \left[ \mathbf{y}^{(T)} \right] h(\mathbf{x}) \right] := \mathbb{E}_{\mathbf{x}} \left[ \boldsymbol{v}(\mathbf{x}) h(\mathbf{x}) \right],
\end{aligned}
\tag{47}
$$

where we define

$$
\boldsymbol{v}(\mathbf{x}) := \nabla_{\boldsymbol{W}} \mathbb{E}_{\mathbf{y} \sim p_{\boldsymbol{W}}(\cdot|\mathbf{x})} \left[ \mathbf{y}^{(T)} \right].
$$

Then, according to the assumption of bounded gradient in Prop. 3.2, we have $\|\boldsymbol{v}(\mathbf{x})\|^2 \leq M$. Now let $\boldsymbol{a} = \mathbf{0}$, then we obtain

$$
\mathbb{E}_{h \in \mathcal{H}} \left[ \left\| \nabla_{\boldsymbol{W}} \mathcal{R}_h^{\mathrm{F}}(\boldsymbol{W}) - \boldsymbol{a} \right\|^2 \right] = \mathbb{E}_{h \in \mathcal{H}} \left[ \left\| \mathbb{E}_{\mathbf{x}} \left[ h(\mathbf{x}) \boldsymbol{v}(\mathbf{x}) \right] \right\|^2 \right].
\tag{48}
$$

As long as we can bound the R.H.S of Eq. (48), we can prove our claim. To this end, we let $\langle h, v \rangle_{L_2} = \mathbb{E}_{\mathbf{x}}[h(\mathbf{x})v(\mathbf{x})]$ denote inner product in the $L_2$ space of square-integrable functions w.r.t the relevant distribution. Then the bound can be established as follows:

$$
\begin{aligned}
\mathbb{E}_{h \in \mathcal{H}} \left[ \left\| \mathbb{E}_{\mathbf{x}} \left[ h(\mathbf{x}) \boldsymbol{v}(\mathbf{x}) \right] \right\|^2 \right] &= \mathbb{E}_{h \in \mathcal{H}} \left[ \sum_\mu \left( \mathbb{E}_{\mathbf{x}} \left[ h(\mathbf{x}) v_\mu(\mathbf{x}) \right] \right)^2 \right] \\
&= \sum_\mu \sum_i \frac{1}{|\mathcal{H}|} \left( \mathbb{E}_{\mathbf{x}} \left[ h_i(\mathbf{x}) v_\mu(\mathbf{x}) \right] \right)^2 \\
&\stackrel{(i)}{=} \sum_\mu \sum_i \frac{1}{|\mathcal{H}|} \langle h_i, v_\mu \rangle_{L_2}^2 \\
&\stackrel{(ii)}{\leq} \sum_\mu \frac{1}{|\mathcal{H}|} \langle v_\mu, v_\mu \rangle_{L_2}^2 \\
&= \frac{1}{|\mathcal{H}|} \mathbb{E}_{\mathbf{x}} \left[ \|\boldsymbol{v}(\mathbf{x})\|^2 \right] \leq \frac{2M}{|\mathcal{H}|},
\end{aligned}
\tag{49}
$$

where $(i)$ uses the definition of $\langle h, v \rangle$; $(ii)$ follows from $\mathbb{E}_{\mathbf{x}}[h(\mathbf{x})h'(\mathbf{x})] = 0$ for distinct $h, h' \in \mathcal{H}$ and $\langle h_i, h_i \rangle_{L_2}^2 \leq 1$. $\qquad \square$

## F. Proofs of Section 3.2

We first present several helpful lemmas before proving Thm. 3.3, which is deferred to App. F.1.

**Lemma F.1** (Equivalence between the separation of gradient and that of critical gradient component)**.** *For the population loss Eq. (11), given $t \in [T]$ and $l^{(t)} \in [d_t]$, if $\forall p \in [d_{t-1}]$ the parameter has the form $W_{N_{t-2}+p, N_{t-1}+l^{(t)}} = c$ for some constant $c$, then*

$$
\begin{aligned}
&- \partial_{W_{N_{t-2}+p, N_{t-1}+l^{(t)}}} L_t(\mathbf{x}; \boldsymbol{W}) - \left( -\partial_{W_{N_{t-2}+p', N_{t-1}+l^{(t)}}} L_t(\mathbf{x}; \boldsymbol{W}) \right) \\
&= \frac{1}{d_{t-1}} \left[ \gamma_{l^{(t)}}^p(\hat{\mathbf{y}}^{(t-1)}) - \gamma_{l^{(t)}}^{p'}(\hat{\mathbf{y}}^{(t-1)}) \right].
\end{aligned}
\tag{50}
$$

*Proof.* To establish such equivalence, we need to derive the formulation of the gradient (recall that we use the hinge loss)

$$
\begin{aligned}
\nabla_{\boldsymbol{W}} \mathcal{L}(\boldsymbol{W}) &:= \mathbb{E}_{\mathbf{x}} \left[ \nabla_{\boldsymbol{W}} \sum_{t=1}^T L_t(\mathbf{x}; \boldsymbol{W}) \right] \\
&= \frac{1}{k-1} \sum_{t=1}^T \sum_{l^{(t)}=1}^{d_t} \mathbb{E}_{\mathbf{x}} \left[ \nabla_{\boldsymbol{W}} \ell \left( \hat{q}_{l^{(t)}}^{(t)}, \tilde{y}_{l^{(t)}}^{(t)} \right) \right] \\
&= -\frac{1}{k-1} \sum_{t=1}^T \sum_{l^{(t)}=1}^{d_t} \mathbb{E}_{\mathbf{x}} \left[ \tilde{y}_{l^{(t)}}^{(t)} \nabla_{\boldsymbol{W}} \hat{q}_{l^{(t)}}^{(t)} \right],
\end{aligned}
\tag{51}
$$

where we use the fact that the activation function is assumed as $\psi : [-1, 1] \to [0, 1]$ to remove $\max$ of the hinge loss in the last equality. In addition, we note that $q^{(t)}_{l^{(t)}}$ only depends on $W_{i,j}$ with $j = N_{t-1} + l^{(t)}$ such that

$$\partial_{W_{i,j}} q^{(t)}_{l^{(t)}} = 0 \ \text{if} \ j \neq N_{t-1} + l^{(t)}, \tag{52}$$

hence we will not consider these components. We now find the gradient of the score. Specifically, given $t \in [T], l^{(t)} \in [d_t]$, recall that only the components with $p \in [d_{t-1}]$,

$$W_{N_{t-2}+p, N_{t-1}+l^{(t)}} \neq -\infty \tag{53}$$

according to the pretrained mask discussed in Sec. 2.2, then (similar to Eq. (26))

$$
\begin{aligned}
&\frac{\tilde{y}^{(t)}_{l^{(t)}}}{k-1} \partial_{W_{N_{t-2}+p, N_{t-1}+l^{(t)}}} \hat{q}^{(t)}_{l^{(t)}} \\
&= \frac{2\tilde{y}^{(t)}_{l^{(t)}}}{k-1} \psi'\left(\xi_{l^{(t)}}\right) \left( \hat{y}^{(t-1)}_p - \sum_{i=1}^{d_{t-1}} \hat{y}^{(t-1)}_i \sigma^{N_{t-2}+i}_{N_{t-1}+l^{(t)}} \right) \sigma^{N_{t-2}+p}_{N_{t-1}+l^{(t)}} \\
&= \left( \gamma^p_{l^{(t)}}(\hat{\mathbf{y}}^{(t-1)}) - \sum_{i=1}^{d_{t-1}} \gamma^i_{l^{(t)}}(\hat{\mathbf{y}}^{(t-1)}) \sigma^{N_{t-2}+i}_{N_{t-1}+l^{(t)}} \right) \sigma^{N_{t-2}+p}_{N_{t-1}+l^{(t)}}.
\end{aligned}
\tag{54}
$$

Therefore, we can now establish the relation between the separation of the gradient of the population loss and that of the critical gradient component. Specifically, given $t \in [T], l^{(t)} \in [d_t]$ for any $p \in [d_{t-1}]$, if $W_{N_{t-2}+p, N_{t-1}+l^{(t)}} = c$ then all corresponding attention scores have the same value; thus

$$
\begin{aligned}
&-\partial_{W_{N_{t-2}+p, N_{t-1}+l^{(t)}}} L_t(\mathbf{x}; \boldsymbol{W}) - \left( -\partial_{W_{N_{t-2}+p', N_{t-1}+l^{(t)}}} L_t(\mathbf{x}; \boldsymbol{W}) \right) \\
&= \frac{1}{d_{t-1}} \left[ \gamma^p_{l^{(t)}}(\hat{\mathbf{y}}^{(t-1)}) - \gamma^{p'}_{l^{(t)}}(\hat{\mathbf{y}}^{(t-1)}) \right].
\end{aligned}
\tag{55}
$$

$\square$

**Lemma F.2** (Separation of critical gradient component $\gamma^p_{l^{(t)}}$ ensures positive gradient update at relevant positions and negative gradient update at irrelevant ones). *Given $t \in [T], \forall l^{(t)} \in [d_t]$ the gradient update $-\partial_{W_{N_{t-2}+p, N_{t-1}+l^{(t)}}} L_t(\mathbf{x}; \boldsymbol{W})$ has positive values at the relevant positions $p \in \{i^{l^{(t)}}_1, i^{l^{(t)}}_2\}$ (the child nodes of $l^{(t)}$-th token of $\mathbf{y}^{(t)}$) and negative values at irrelevant positions (all other nodes) if the following conditions hold:*

*(i) the model parameter $W_{N_{t-2}+p, N_{t-1}+l^{(t)}} = c$ for any $p \in [d_{t-1}]$ and $l^{(t)} \in [d_t]$;*

*(ii) $\hat{\mathbf{y}}^{(t-1)}$ is correctly generated as $\tilde{\mathbf{y}}^{(t-1)}$, i.e., $\hat{\mathbf{y}}^{(t-1)} = \tilde{\mathbf{y}}^{(t-1)}$;*

*(iii) the separation of the critical gradient component $\gamma^p_{l^{(t)}}$ in Lem. F.1 is satisfied.*

*Proof.* We consider optimizing $L_t$ under the case where $\hat{\mathbf{y}}^{(t-1)} = \tilde{\mathbf{y}}^{(t-1)}$ (i.e., $\hat{\mathbf{y}}^{(t-1)}$ is correctly generated and is the same as the ground-truth label $\tilde{\mathbf{y}}^{(t-1)}$) and the parameter $W_{N_{t-2}+p, N_{t-1}+l^{(t)}} = c$ for any $p \in [d_{t-1}]$ and $l^{(t)} \in [d_t]$. As a result, if the separation of the critical gradient component is satisfied at the $t$-th step

$$
\begin{aligned}
\forall l^{(t)} \in [d_t], p \in \{i^{l^{(t)}}_1, i^{l^{(t)}}_2\}, p' \in [d_{t-1}] \backslash \{i^{l^{(t)}}_1, i^{l^{(t)}}_2\} : \\
\mathbb{E}_{\mathbf{x}} \left[ \gamma^p_{l^{(t)}}(\tilde{\mathbf{y}}^{(t-1)}) - \gamma^{p'}_{l^{(t)}}(\tilde{\mathbf{y}}^{(t-1)}) \right] > 0,
\end{aligned}
\tag{56}
$$

we can conclude that

$$\mathbb{E}_{\mathbf{x}} \left[ -\partial_{W_{N_{t-2}+p, N_{t-1}+l^{(t)}}} L_t(\mathbf{x}; \boldsymbol{W}) - \left( -\partial_{W_{N_{t-2}+p', N_{t-1}+l^{(t)}}} L_t(\mathbf{x}; \boldsymbol{W}) \right) \right] > 0 \tag{57}$$

by applying Lem. F.1 and $\hat{\mathbf{y}}^{(t-1)} = \tilde{\mathbf{y}}^{(t-1)}$. Now, similar to the proof of RL (App. E.3), we observe that

$$\sum_{p=1}^{d_{t-1}} \partial_{W_{N_{t-2}+p, N_{t-1}+l^{(t)}}} L_t(\mathbf{x}; \boldsymbol{W}) = 0, \tag{58}$$

which further implies that

$$
\mathbb{E}_{\mathbf{x}}\left[-\partial_{W_{N_{t-2}+p,N_{t-1}+l^{(t)}}} L_t(\mathbf{x};\boldsymbol{W})\right] = \begin{cases} G_1, & p \in \{i_1^{l^{(t)}}, i_2^{l^{(t)}}\}, \\ G_2 & p \in [d_{t-1}]\setminus\{i_1^{l^{(t)}}, i_2^{l^{(t)}}\}, \end{cases}
\tag{59}
$$

with $G_1 > G_2$. Here we use the symmetry of $i_1^{l^{(t)}}$ and $i_2^{l^{(t)}}$ and that of all other positions (i.e., $\gamma_{l^{(t)}}^{i_1^{l^{(t)}}} = \gamma_{l^{(t)}}^{i_2^{l^{(t)}}}$ and $\forall p \in [d_{t-1}]\setminus\{i_1^{l^{(t)}}, i_2^{l^{(t)}}\} : \gamma_{l^{(t)}}^p = c_2$ for some constant $c_2$). Now use Eq. (58), then we must have $G_1 > 0$ and $G_2 < 0$. Therefore, the lemma is proved. $\qquad\square$

### F.1. Proof of Thm. 3.3

We now prove Thm. 3.3. For convenience, we restate Thm. 3.3 below.

**Theorem F.3** (Restated Thm. 3.3). *Given integers $d \geq k \geq 2$, consider a $k$-sparse Boolean function $\Phi_k(\cdot)$ with any subset $B \in [d]$ as in Def. 2.1. Let $\boldsymbol{W}(0) = \mathbf{1}$ be the initialization and let*

$$
\boldsymbol{W}^\star = \arg\min_{\boldsymbol{W}} \mathcal{L}(\boldsymbol{W})
$$

*be the optimal parameter that solves $\min_{\boldsymbol{W}} \mathcal{L}(\boldsymbol{W})$. Set learning rate $\eta = \Omega\left(\ln(d/\epsilon)\right)$ for any $\epsilon > 0$. Let the transformer $f(\cdot; \boldsymbol{W})$ be fine-tuned via SFT by running sign gradient descent*

$$
\boldsymbol{W}(s+1) = \boldsymbol{W}(s) - \eta\,\mathrm{sign}\left(\nabla_{\boldsymbol{W}}\mathcal{L}(\boldsymbol{W}(s))\right).
\tag{60}
$$

*If the separation of the critical gradient component is satisfied for any $l^{(t)} \in [d_t]$ and any $t \in [T]$ in the sense that*

$$
\forall p \in \{i_1^{l^{(t)}}, i_2^{l^{(t)}}\}, p' \in [d_{t-1}]\setminus\{i_1^{l^{(t)}}, i_2^{l^{(t)}}\} : \ \mathbb{E}_{\mathbf{x}}\left[\gamma_{l^{(t)}}^p(\tilde{\mathbf{y}}^{(t-1)}) - \gamma_{l^{(t)}}^{p'}(\tilde{\mathbf{y}}^{(t-1)})\right] > 0,
\tag{61}
$$

*where $\tilde{\mathbf{y}}^{(t-1)}$ is the ground-truth label of $\mathbf{y}^{(t-1)}$ given an input $\mathbf{x}$, then running sign gradient descent for $T$ iterations achieves*

$$
\|\mathrm{softmax}(\boldsymbol{W}(T)) - \mathrm{softmax}(\boldsymbol{W}^\star)\|_1 \leq \epsilon.
\tag{62}
$$

*Proof.* Note that, different from RL, once the input sequence $\mathbf{x}$ is given, the ground-truth label $\tilde{\mathbf{y}}$ and the generated output sequence $\hat{\mathbf{y}}$ before each update of sign gradient descent are both determined. There will be two main steps of the proof, and we start with the first one.

**Step (i): Conditions of Lem. F.2 at the step $t$ guarantee the learnability of $\mathbf{y}^{(t)}$.** Given $t \in [T]$, with Lem. F.2, we are able to prove the learnability of $\mathbf{y}^{(t)}$, as shown below. Suppose that the model parameters are given by condition (i) and $\hat{\mathbf{y}}^{(t-1)} = \tilde{\mathbf{y}}^{(t-1)}$ as in condition (ii) of Lem. F.2. If the condition (iii) is also satisfied, i.e., the separation of $\gamma_{l^{(t)}}^p$ is satisfied at the step $t$, then we can conclude that the gradient update $-\partial_{W_{N_{t-2}+p,N_{t-2}+l^{(t)}}} L_t(\mathbf{x}; \boldsymbol{W})$ has positive values at the relevant positions $p \in \{i_1^{l^{(t)}}, i_2^{l^{(t)}}\}$ and negative values at irrelevant positions for any $l^{(t)} \in [d_t]$. As a result, one update of sign gradient descent gives us

$$
W_{N_{t-2}+p,N_{t-1}+l^{(t)}} = \begin{cases} 1+\eta, & p \in \{i_1^{l^{(t)}}, i_2^{l^{(t)}}\}, \\ 1-\eta, & p \in [d_t]\setminus\{i_1^{l^{(t)}}, i_2^{l^{(t)}}\}. \end{cases}
\tag{63}
$$

After the softmax we obtain

$$
\forall l^{(t)} \in [d_t]: \ \sigma_{N_{t-1}+l^{(t)}}^{N_{t-2}+p} = \begin{cases} \frac{1}{2}\frac{1}{1+\frac{d_{t-1}-2}{2}e^{-2\eta}} = \lambda_1^{(t)}, & p \in \{i_1^{l^{(t)}}, i_2^{l^{(t)}}\}, \\ \frac{1}{d_{t-1}-2+2e^{2\eta}} = \lambda_2^{(t)}, & p \in [d_{t-1}]\setminus\{i_1^{l^{(t)}}, i_2^{l^{(t)}}\}. \end{cases}
\tag{64}
$$

Compared to the optimal model parameter $(\sigma^\star)_{N_{t-1}+l^{(t)}}^{N_{t-2}+p}$ for $p \in [d_{t-1}]$ that solves $L_t$

$$
\forall l^{(t)} \in [d_t]: \ (\sigma^\star)_{N_{t-1}+l^{(t)}}^{N_{t-2}+p} = \begin{cases} \frac{1}{2}, & p \in \{i_1^{l^{(t)}}, i_2^{l^{(t)}}\}, \\ 0, & p \in [d_{t-1}]\setminus\{i_1^{l^{(t)}}, i_2^{l^{(t)}}\}, \end{cases}
\tag{65}
$$

we can evaluate the error as

$$\forall l^{(t)} \in [d_t] : \sum_{p=1}^{d_{t-1}} \left| \sigma_{N_{t-1}+l^{(t)}}^{N_{t-2}+p} - (\sigma^\star)_{N_{t-1}+l^{(t)}}^{N_{t-2}+p} \right| = \frac{1}{\frac{1}{2} + \frac{e^{2\eta}}{d_{t-1}-2}} \leq \epsilon, \tag{66}$$

if $\eta = \Omega(\ln(d/\epsilon))$. Furthermore, noting that

$$\sum_{i=1}^{d_{t-1}} y_i^{(t-1)} \sigma_{N_{t-1}+l^{(t)}}^{N_{t-2}+i}$$

$$= \frac{y_{i_1^{l^{(t)}}}^{(t-1)} + y_{i_1^{l^{(t)}}}^{(t-1)}}{2} \left[ 1 - \frac{d_{t-1}-2}{d_{t-1}-2+2e^{2\eta}} \right] + \frac{1}{d_{t-1}-2+2e^{2\eta}} \sum_{p \in [d_{t-1}]\setminus\{i_1^{l^{(t)}}, i_2^{l^{(t)}}\}} y_p^{(t-1)}$$

$$= \sum_{i=1}^{d_{t-1}} y_i^{(t-1)} (\sigma^\star)_{N_{t-1}+l^{(t)}}^{N_{t-2}+i}$$

$$+ \frac{1}{d_{t-1}-2+2e^{2\eta}} \left[ -(d_{t-1}-2) \frac{y_{i_1^{l^{(t)}}}^{(t-1)} + y_{i_1^{l^{(t)}}}^{(t-1)}}{2} + \sum_{p \in [d_{t-1}]\setminus\{i_1^{l^{(t)}}, i_2^{l^{(t)}}\}} y_p^{(t-1)} \right], \tag{67}$$

we are able to conclude

$$\sum_{i=1}^{d_{t-1}} y_i^{(t-1)} (\sigma^\star)_{N_{t-1}+l^{(t)}}^{N_{t-2}+i} - 2\epsilon \leq \sum_{i=1}^{d_{t-1}} y_i^{(t-1)} \sigma_{N_{t-1}+l^{(t)}}^{N_{t-2}+i}$$

$$\leq \sum_{i=1}^{d_{t-1}} y_i^{(t-1)} (\sigma^\star)_{N_{t-1}+l^{(t)}}^{N_{t-2}+i} + 2\epsilon. \tag{68}$$

Therefore, if $\psi(\cdot)$ guarantees the flexibility of the expressibility of the transformer in the sense that

$$\phi_2(z_1, z_2) = \text{sign}\left( 2\psi\left( \frac{z_1+z_2}{2} \right) - 1 \right) = \text{sign}\left( 2\psi\left( \frac{z_1+z_2}{2} + \lambda \right) - 1 \right), \tag{69}$$

where we take $\lambda = -2\epsilon$ when $z_1 + z_2$ equals 1 or 0 and take $2\epsilon$ when $z_1 + z_2 = -1$ for sufficiently small $\epsilon$, one update of the sign gradient descent ensures that

$$\text{sign}\left( 2\psi\left( \sum_{i=1}^{d_{t-1}} y_i^{(t-1)} (\sigma^\star)_{N_{t-1}+l^{(t)}}^{N_{t-2}+i} \right) - 1 \right) = \text{sign}\left( 2\psi\left( \sum_{i=1}^{d_{t-1}} y_i^{(t-1)} \sigma_{N_{t-1}+l^{(t)}}^{N_{t-2}+i} \right) - 1 \right), \tag{70}$$

and, as a result, all tokens $\hat{y}_{l^{(t)}}^{(t)}$ for $l^{(t)} \in [d_t]$ will be generated correctly.

Furthermore, denoting

$$\gamma_{l^{(t)}}^p(\tilde{\mathbf{y}}^{(t-1)}) = \begin{cases} I, & \text{if } p \in \{i_1^{(t)}, i_2^{(t)}\} \\ J, & \text{otherwise.} \end{cases} \tag{71}$$

after the update, then the expected negative gradient at the $t$-th step can be written as, according to Eq. (54),

$$\begin{cases} \lambda_1^{(t)} \left[ I - (2\lambda_1^{(t)} I + (d_{t-1}-2)\lambda_2^{(t)} J) \right], & \text{if } p \in \{i_1^{(t)}, i_2^{(t)}\} \\ \lambda_2^{(t)} \left[ J - (2\lambda_1^{(t)} I + (d_{t-1}-2)\lambda_2^{(t)} J) \right], & \text{otherwise.} \end{cases} \tag{72}$$

Using the constraint $2\lambda_1^{(t)} + (d_{t-1}-2)\lambda_2^{(t)} = 1$, this can be further simplified to (for $p \in \{i_1^{(t)}, i_2^{(t)}\}$) $\lambda_1^{(t)} \lambda_2^{(t)}(d_{t-1}-2)(I-J)$, and, similarly for $p \notin \{i_1^{(t)}, i_2^{(t)}\}$, $-2\lambda_1^{(t)} \lambda_2^{(t)}(I-J)$. Hence, both of them are proportional to $\lambda_1^{(t)} \lambda_2^{(t)}$ with bounded $I - J$. Using the constraint $2\lambda_1^{(t)} + (d_{t-1}-2)\lambda_2^{(t)} = 1$ again, we have

$$\lambda_1^{(t)} \lambda_2^{(t)} = \lambda_1^{(t)} \frac{1 - 2\lambda_1^{(t)}}{d_{t-1}-2}. \tag{73}$$

Note that after the update $\lambda_1^{(t)} = \frac{1}{2} - \delta \geq \frac{1}{2} - \epsilon$ for a sufficiently small constant $\delta$, then we are able to write

$$\lambda_1^{(t)}\lambda_2^{(t)} = \frac{\delta(1-2\delta)}{d_{t-1}-2} = \frac{\delta}{d_{t-1}-2} + O(\delta^2), \tag{74}$$

which is negligible for sufficiently small $\epsilon$ (hence sufficiently small $\delta$). We thus zero out the gradient $\nabla_{\boldsymbol{W}}\mathbb{E}[L_t]$ after the $t$-th step update, i.e., all tokens in $\mathbf{y}^{(t)}$ are correctly generated.

**Step (ii): $T$-updates of sign gradient descent can achieve small error.** We prove this claim in an induction manner. We use non-negative integers $s$ to count sign gradient descent, which is distinct from $t$ of the CoT chain. We let $\eta = \Omega(\ln(d/\epsilon))$.

1. First of all, at $s = 0$, the models parameters $\boldsymbol{W} = \mathbf{1}c$, hence, condition (i) of Lem. F.2 is satisfied. Furthermore, all $\hat{y}_{l^{(1)}}^{(1)}$ must be assigned to a same value, i.e.,

$$\text{either } \hat{y}_{l^{(1)}}^{(1)} = 1, \forall l^{(1)} \in [d_1] \text{ or } \hat{y}_{l^{(1)}}^{(1)} = -1, \forall l^{(1)} \in [d_1], \tag{75}$$

because $\sigma_{N_0+l^{(1)}}^{N_{-1}+p}(s = 0) = 1/d_0$ for any $l^{(1)} \in [d_1]$ such that all $\xi_{l^{(1)}}$ for different $l^{(1)}$ are equal according to the definition which further implies that the outputs of the transformer $\psi(\xi_{l^{(1)}})$ for different $l^{(1)}$ are equal. As a result, Eq. (54) tells us $\partial_{W_{N_0+p,N_1+l^{(2)}}} \hat{q}_{l^{(2)}}^{(2)} = 0$ for any $l^{(2)} \in [d_2]$ and $p \in [d_1]$. Similar argument can give us $\partial_{W_{N_{t-2}+p,N_{t-1}+l^{(t)}}} \hat{q}_{l^{(t)}}^{(t)} = 0$ for any $t \in [2,T]$ and any $l^{(t)} \in [d_t]$. Hence, all parameters $W_{i,j}$ with $j \in [N_1 + 1, N_T]$ will not be updated at $s = 0$, and we only need to consider optimizing $\mathbb{E}_{\mathbf{x}}[L_1(\mathbf{x}, \boldsymbol{W})]$.

   For the first step of the CoT chain $t = 1$, we have $\hat{\mathbf{y}}^{(t-1)} = \tilde{\mathbf{y}}^{(t-1)} = \mathbf{x}$, and thus condition (ii) of Lem. F.2 is satisfied. If the condition (iii) of Lem. F.2, the separation of the critical gradient component, is further satisfied, then we can apply our statement in **Step (i)** to conclude that one update of sign gradient descent gives us

$$\sum_{p=1}^{d_0} \left| \sigma_{N_0+l^{(1)}}^{N_{-1}+p} - (\sigma^\star)_{N_0+l^{(1)}}^{N_{-1}+p} \right| = \frac{1}{\frac{1}{2} + \frac{e^{2\eta}}{d-2}} \leq \epsilon, \tag{76}$$

   and all tokens $\hat{y}_{l^{(1)}}^{(1)}$ for $l^{(1)} \in [d_1]$ will be generated correctly for sufficiently small $\epsilon$ that guarantees Eq. (69).

2. Now at $s = 1$, we consider the second step of the CoT chain $t = 2$, where $\hat{\mathbf{y}}^{(t-1)}$ is generated correctly and thus the first step will not be optimized again due to negligible gradient. Therefore, following the argument for $s = 0$, we can easily conclude that only $\mathbb{E}_{\mathbf{x}}[L_2(\mathbf{x}; \boldsymbol{W})]$ will be optimized and, under the separation of the critical gradient component, we have that after one update of sign gradient descent

$$\forall l^{(2)} \in [d_2] : \sum_{p=1}^{d_1} \left| \sigma_{N_1+l^{(2)}}^{N_0+p} - (\sigma^\star)_{N_1+l^{(2)}}^{N_0+p} \right| = \frac{1}{\frac{1}{2} + \frac{e^{2\eta}}{d_1-2}} \leq \frac{1}{\frac{1}{2} + \frac{e^{2\eta}}{d-2}} \leq \epsilon, \tag{77}$$

   and $\hat{\mathbf{y}}^{(2)}$ will be generated correctly.

3. Repeating the above argument, we can conclude that, if the separation of the critical gradient component is satisfied as in Thm. 3.3 and the parameter is initialized as $\boldsymbol{W} = \mathbf{1}c$, each update of the sign gradient descent solves and only solves one step $\mathbf{y}^{(t)}$ and $T$ updates solve the whole CoT chain, with an error

$$\|\text{softmax}(\boldsymbol{W}(T)) - \text{softmax}(\boldsymbol{W}^\star)\|_1 = \max_t \frac{1}{\frac{1}{2} + \frac{e^{2\eta}}{d_t-2}} = \frac{1}{\frac{1}{2} + \frac{e^{2\eta}}{d-2}} \leq \epsilon. \tag{78}$$

$\square$

### F.2. Extension to General SFT Loss Functions

In Thm. 3.3, we have used the hinge loss, which simplifies the final expressions. But ther results are not restricted to the hinge loss. In this section. we discuss how the analysis extends to other margin loss to clarify which parts of the argument are intrinsic to the autoregressive SFT learning dynamics, i.e., later reasoning steps become learnable only after earlier generated steps are correct, and the choice of loss function only affects the magnitude and weighting of the gradient signal.

**General margin losses.** We now consider a more general binary margin loss of the form

$$\ell(\hat{q}, a) = \varphi(a\hat{q}), \qquad a \in \{\pm 1\},$$

where $\hat{q}$ is the model score and $a$ is the target label. We assume that $\varphi : \mathbb{R} \to \mathbb{R}_+$ is nonincreasing, so that larger signed margin $a\hat{q}$ corresponds to smaller loss. This class includes the hinge, logistic, and exponential losses:

$$\varphi_{\text{hinge}}(u) = \max\{0, 1 - u\}, \tag{79}$$

$$\varphi_{\text{log}}(u) = \log(1 + e^{-u}), \tag{80}$$

$$\varphi_{\text{exp}}(u) = e^{-u}. \tag{81}$$

For differentiable $\varphi$, define its derivative as

$$\lambda(u) := -\varphi'(u).$$

Since $\varphi$ is nonincreasing, $\lambda(u) \geq 0$ wherever the derivative exists. The gradient of the loss with respect to the model parameters is then

$$-\nabla_{\boldsymbol{W}} \ell(\hat{q}, a) = \lambda(a\hat{q}) \, a \, \nabla_{\boldsymbol{W}} \hat{q}. \tag{82}$$

Thus, relative to the proof of hinge loss (Eq. (51) of App. F.1), the only new factor is the nonnegative scalar $\lambda(a\hat{q})$, which depends on the current signed margin. Specifically, for logistic and exponential losses, respectively,

$$\lambda_{\text{log}}(u) = \frac{1}{1 + e^u}, \qquad \lambda_{\text{exp}}(u) = e^{-u}.$$

Both are strictly positive for all finite $u$, so they preserve the direction of the gradient of general margin losses Eq. (82) while only changing its magnitude.

**Weighted critical gradient component.** The main hinge-loss argument identifies a critical gradient component

$$\gamma_{l^{(t)}}^{p}\left(\hat{\mathbf{y}}^{(t-1)}\right)$$

that separates relevant previous-step positions from irrelevant ones, which associates with position $p$ when predicting coordinate $l^{(t)}$ at step $t$. Under hinge loss, the sign of the update is determined by the separation between

$$\gamma_{l^{(t)}}^{p}\left(\hat{\mathbf{y}}^{(t-1)}\right) \quad \text{and} \quad \gamma_{l^{(t)}}^{p'}\left(\hat{\mathbf{y}}^{(t-1)}\right),$$

where $p$ is a relevant parent position and $p'$ is an irrelevant position.

For a general margin loss, according to Eq. (82), the corresponding quantity now becomes a weighted version of the original critical gradient component $\gamma_{l^{(t)}}^{p}\left(\hat{\mathbf{y}}^{(t-1)}\right)$:

$$\textbf{Weighted Critical Gradient Component: } \Gamma_{l^{(t)}}^{p}\left(\hat{\mathbf{y}}^{(t-1)}\right) := \lambda\left(\tilde{y}_{l^{(t)}}^{(t)} \hat{q}_{l^{(t)}}^{(t)}\right) \gamma_{l^{(t)}}^{p}\left(\hat{\mathbf{y}}^{(t-1)}\right). \tag{83}$$

The scalar weight depends on the signed margin of the prediction being trained, but not on the particular previous position $p$ whose attention weight is being differentiated. Therefore, for a fixed prediction coordinate $l^{(t)}$, the loss changes the scale of the gradient comparison but does not directly alter the relevant vs. irrelevant structure.

**Generalized separation condition.** The weighted critical gradient component allows us to extend the conclusion in Thm. 3.3 to more general margin loss functions by replacing the original critical gradient component with its weighted version, i.e., the analog of the hinge loss calculation in Lem. F.1 now will be

$$-\partial_{W_{p,j}} L_t(\mathbf{x}; \boldsymbol{W}) - \left(-\partial_{W_{p',j}} L_t(\mathbf{x}; \boldsymbol{W})\right) = \frac{1}{d_{t-1}} \left[\Gamma_{l^{(t)}}^{p}\left(\hat{\mathbf{y}}^{(t-1)}\right) - \Gamma_{l^{(t)}}^{p'}\left(\hat{\mathbf{y}}^{(t-1)}\right)\right]. \tag{84}$$

Hence the original proof can be viewed as the special case $\Gamma_{l^{(t)}}^{p} = \gamma_{l^{(t)}}^{p}$. Then under a general margin loss, the corresponding separation condition is also a weighted separation condition

$$\mathbb{E}_{\mathbf{x}}\left[\Gamma_{l^{(t)}}^{p}\left(\tilde{\mathbf{y}}^{(t-1)}\right) - \Gamma_{l^{(t)}}^{p'}\left(\tilde{\mathbf{y}}^{(t-1)}\right)\right] = \mathbb{E}_{\mathbf{x}}\left[\lambda\left(\tilde{y}_{l^{(t)}}^{(t)} \hat{q}_{l^{(t)}}^{(t)}\right)\left(\gamma_{l^{(t)}}^{p}\left(\tilde{\mathbf{y}}^{(t-1)}\right) - \gamma_{l^{(t)}}^{p'}\left(\tilde{\mathbf{y}}^{(t-1)}\right)\right)\right] > 0.$$

under the same symmetry and equal-initialization conditions used in the hinge-loss analysis, where $p$ is a relevant position and $p'$ is an irrelevant one. If $\lambda$ is bounded below by $\mu > 0$ and the unweighted separation condition holds with a positive margin, then the weighted condition follows whenever the loss-dependent weight does not reverse or erase the underlying separation. Logistic and exponential losses satisfy this requirement under the bounded margin regime considered above because their derivatives are strictly negative and controlled.

We can summarize the above discussion in the following proposition.

**Proposition F.4** (Learnability via SFT with general monotone margin losses). *Consider the same SFT setting as in Thm. 3.3, but replace the hinge loss by a differentiable margin loss*

$$\ell(\hat{q}, a) = \varphi(a\hat{q}),$$

*where $\varphi$ is nonincreasing. Suppose that $\varphi'$ is bounded below, and the the weighted critical gradient separation condition holds:*

$$\mathbb{E}_{\mathbf{x}} \left[ \Gamma_{l(t)}^p \left( \tilde{\mathbf{y}}^{(t-1)} \right) - \Gamma_{l(t)}^{p'} \left( \tilde{\mathbf{y}}^{(t-1)} \right) \right] > 0 \tag{85}$$

*for every relevant $p$ and irrelevant $p'$ at each step $t$. Then the sign of the SFT population gradient favors the relevant parent positions over irrelevant positions in the same sense as in the hinge loss analysis. As a result, the same step-wise learning behavior for the hinge loss in Thm. 3.3 holds for $\varphi$.*

*Proof.* The proof follows the hinge loss argument (App. F.1) with one modification. For the hinge loss, the negative gradient of the loss with respect to the score is proportional to the label $a$. For the general margin loss,

$$-\nabla_{\mathbf{W}} \ell(\hat{q}, a) = \lambda(a\hat{q}) \, a \, \nabla_{\mathbf{W}} \hat{q},$$

where $\lambda(a\hat{q}) = -\varphi'(a\hat{q}) \geq 0$. Therefore, each critical gradient component is multiplied by the scalar margin weight $\lambda(a\hat{q})$, which is finite and bounded away from zero on the realized margin range. The same comparison between relevant and irrelevant positions then applies to the weighted quantities $\Gamma_{l(t)}^p$ and $\Gamma_{l(t)}^{p'}$. The weighted separation condition ensures that the population sign-gradient update increases the relative attention assigned to relevant positions. The autoregressive induction over $t$ is unchanged, since it depends only on the fact that the model-generated prefix becomes correct one step at a time. This completes the extension. $\square$

The logistic and exponential losses satisfy the conditions in Prop. F.4, and thus the step-wise learning dynamics in the hinge loss analysis is recovered. This extension shows that the hinge loss is not essential to the qualitative SFT phenomenon. Its main role is algebraic: in the active-margin regime, the hinge loss has constant derivative, so the gradient comparison is unweighted. For a general monotone margin loss, the same comparison is weighted by the derivative of the loss at the current margin.

The step-wise nature of SFT is instead caused by autoregressive training on the model-generated prefix. Later reasoning steps receive a useful gradient only after earlier generated steps become correct. This mechanism is independent of the precise form of the margin loss, as long as it is differentiable monotone margin losses whose derivatives are strictly negative and bounded on the relevant margin range.

# G. Proofs of Section 4

In this section, for given $k$-sparse Boolean functions, we verify that they satisfy the conditions established in Thm. 3.1 and that in Thm. 3.3, which reveals that transformers with CoT provably learn these functions via RL or SFT.

- App. G.1 discusses the design of the activation functions listed in Tab. 1;

- App. G.2 discusses $k$-PARITY;

- App. G.3.1 discusses $k$-AND as well as $k$-OR.

## G.1. Formulations of the Activation Functions

Recall that the activation function $\psi : [-1, 1] \rightarrow [0, 1]$ ensures that the output of the transformer Eq. (3) can be seen as the probability of generating $y = 1$ for RL as well as a score of the token for SFT. Therefore, we must make sure that $\psi((z_1 + z_2)/2)$ is large if $\phi_2(z_1, z_2) = 1$ such that the transformer has sufficient expressibility to capture the target $k$-sparse Boolean functions. Following this principle:

1. For $k$-PARITY, we have $\phi_2(z_1, z_2) = z_1 z_2$ in the sense that

$$\phi_2(z_1, z_2) = \begin{cases} 1, & z_1 = z_2, \\ -1, & z_1 \neq z_2; \end{cases} \tag{86}$$

   therefore, we need $\psi(\pm 1) \approx 1$ while $\psi(0) \approx 0$. A natural choice will be $\psi(x) = x^2$.

2. For $k$-AND, we have

$$\phi_2(z_1, z_2) = \begin{cases} 1, & z_1 = z_2 = 1, \\ -1, & \text{otherwise}; \end{cases} \tag{87}$$

   hence we need $\psi(1) \approx 1$ while $\psi(-1)$ and $\psi(0)$ are approximately 0. A simple function that satisfies this condition is $\psi(x) = \max(x, 0)$.

3. For $k$-OR, we have

$$\phi_2(z_1, z_2) = \begin{cases} -1, & z_1 = z_2 = -1, \\ 1, & \text{otherwise}; \end{cases} \tag{88}$$

   hence we need $\psi(-1) \approx 0$ while $\psi(1)$ and $\psi(0)$ are approximately 1. A simple function that satisfies this condition will be $\psi(x) = 1 + \min(x, 0)$.

## G.2. **Parity**

### G.2.1. FINE-TUNING VIA RL

For parity, we can give a more exact characterization of the learning dynamics as shown in Thm. 4.1. Recall that $\psi(z) = z^2$ and $\phi_2(z_1, z_2) = z_1 z_2$. We have

$$\psi'(\xi_{l^{(t)}}) = 2\xi_{l^{(t)}} = 2 \sum_j \sigma_{N_{t-1}+l^{(t)}}^{N_{t-2}+j} y_j^{(t-1)}, \tag{89}$$

$$\phi\left(y_{i_1^{l^{(t)}}}^{(t-1)}, y_{i_2^{l^{(t)}}}^{(t-1)}\right) = y_{i_1^{l^{(t)}}}^{(t-1)} y_{i_2^{l^{(t)}}}^{(t-1)}. \tag{90}$$

We first present a helpful lemma.

**Lemma G.1.** *Given $t \in [T]$, if the sum of the square of the attention score $\sigma_{N_{t-1}+l^{(t)}}^{N_{t-2}+q}$ over $q$ is the same for different $l^{(t)}$:*

$$\forall l^{(t)} \in [d_t] : \sum_{q=1}^{d_{t-1}} \left(\sigma_{N_{t-1}+l^{(t)}}^{N_{t-2}+q}\right)^2 = c_1^{(t)}, \tag{91}$$

*then we have that the expectations for different tokens $y_{l^{(t)}}^{(t)}$ in the same layer $t$ have a same value:*

$$\forall l^{(t)} \in [d_t] : \mathop{\mathbb{E}}_{\mathbf{x}, \mathbf{y}^{(:t)} \sim p_{\mathbf{W}}(\cdot | \mathbf{x})} \left[y_{l^{(t)}}^{(t)}\right] = c_2^{(t)}, \tag{92}$$

*with $|c_2^{(t)}| < 1$.*

We first discuss the proof of Thm. 4.1 before discussing the proof of the above lemma.

*Proof of Thm. 4.1.* The proof is based on the evaluation of the critical gradient component, which enables to verify that it satisfies the separation condition in Thm. 3.1. Below we calculate $\gamma_{l^{(t)}}^p$:

$$
\mathbb{E}_{\mathbf{y}^{(:t-1)}}\left[\gamma_{l^{(t)}}^p(\mathbf{y}^{(:t-1)})\right] := \mathbb{E}_{\mathbf{y}^{(:t-1)}}\left[\left(\sum_{j=1}^{d_{t-1}} \sigma_{N_{t-1}+l^{(t)}}^{N_{t-2}+j} y_j^{(t-1)}\right) y_{i_1^{l^{(t)}}}^{(t-1)} y_{i_2^{l^{(t)}}}^{(t-1)} y_p^{(t-1)}\right]. \tag{93}
$$

Consider optimizing the expected reward $\mathcal{R}(\boldsymbol{W})$ with the sign of the policy gradient. We use non-negative integers $s$ to count the iterations of the policy gradient. At the first iteration $s = 0$ we already know that

$$
s = 0, \; \forall t \in [T], l^{(t)} \in [d_t]: \; \sum_q (\sigma_{N_{t-1}+l^{(t)}}^{N_{t-2}+q})^2 = c_1^{(t)}(0) \tag{94}
$$

because $\boldsymbol{W}(0) = \mathbf{1}c$. Then Lem. G.1 gives us

$$
s = 0, \; \forall t \in [T], l^{(t)} \in [d_t]: \; \mathbb{E}_{\mathbf{x},\mathbf{y}^{(:t)}}\left[y_{l^{(t)}}^{(t)}\right] = c_2^{(t)}(0). \tag{95}
$$

Thus, according to Eq. (93), the critical gradient component can be calculated as

- if $p \in \{i_1^{l^{(t)}}, i_2^{l^{(t)}}\}$, say $p = i_2^{l^{(t)}}$, then

$$
\begin{aligned}
&\mathbb{E}_{\mathbf{x},\mathbf{y}^{(:t-1)}}\left[\gamma_{l^{(t)}}^p(\mathbf{y}^{(:t-1)})\right] \\
&= \sum_{j=1}^{d_{t-1}} \sigma_{N_{t-1}+l^{(t)}}^{N_{t-2}+j} \mathbb{E}_{\mathbf{x},\mathbf{y}^{(:t-1)}}\left[y_j^{(t-1)} y_{i_1^{l^{(t)}}}^{(t-1)}\right] \\
&\stackrel{(i)}{=} \sum_{j \neq i_1^{l^{(t)}}} \sigma_{N_{t-1}+l^{(t)}}^{N_{t-2}+j} \mathbb{E}_{\mathbf{x},\mathbf{y}^{(:t-1)}}\left[y_j^{(t-1)}\right] \mathbb{E}_{\mathbf{x},\mathbf{y}^{(:t-1)}}\left[y_{i_1^{l^{(t)}}}^{(t-1)}\right] + \sigma_{N_{t-1}+l^{(t)}}^{N_{t-2}+i_1^{l^{(t)}}} \\
&\stackrel{(ii)}{=} \left(c_2^{(t-1)}(0)\right)^2 + \sigma_{N_{t-1}+l^{(t)}}^{N_{t-2}+i_1^{l^{(t)}}}\left(1 - \left(c_2^{(t-1)}(0)\right)^2\right),
\end{aligned} \tag{96}
$$

where we use that $y_j^{(t)}$ and $y_{j'}^{(t)}$ are independent if $j \neq j'$ in $(i)$ and $\sum_j \sigma_{N_{t-1}+l^{(t)}}^{N_{t-2}+j} = 1$ in $(ii)$. Similarly, if $p = i_1^{l^{(t)}}$,

$$
\begin{aligned}
&\mathbb{E}_{\mathbf{x},\mathbf{y}^{(:t-1)}}\left[\gamma_{l^{(t)}(\mathbf{y}^{(:t-1)})}^p\right] \\
&= \sum_{j \neq i_2^{l^{(t)}}} \sigma_{N_{t-1}+l^{(t)}}^{N_{t-2}+j} \mathbb{E}_{\mathbf{x},\mathbf{y}^{(:t-1)}}\left[y_j^{(t-1)}\right] \mathbb{E}_{\mathbf{x},\mathbf{y}^{(:t-1)}}\left[y_{i_2^{l^{(t)}}}^{(t-1)}\right] + \sigma_{N_{t-1}+l^{(t)}}^{N_{t-2}+i_2^{l^{(t)}}} \\
&= \left(c_2^{(t-1)}(0)\right)^2 + \sigma_{N_{t-1}+l^{(t)}}^{N_{t-2}+i_2^{l^{(t)}}}\left(1 - \left(c_2^{(t-1)}(0)\right)^2\right).
\end{aligned} \tag{97}
$$

- if $p \notin \{i_1^{l^{(t)}}, i_2^{l^{(t)}}\}$, then (similar to the case of $p \in \{i_1^{l^{(t)}}, i_2^{l^{(t)}}\}$)

$$
\begin{aligned}
&\mathbb{E}_{\mathbf{x},\mathbf{y}^{(:t-1)}}\left[\gamma_{l^{(t)}}^p(\mathbf{y}^{(:t-1)})\right] \\
&= \sum_j \sigma_{N_{t-1}+l^{(t)}}^{N_{t-2}+j} \mathbb{E}_{\mathbf{x},\mathbf{y}^{(:t-1)}}\left[y_j^{(t-1)} y_p^{(t-1)} y_{i_1^{l^{(t)}}}^{(t-1)} y_{i_2^{l^{(t)}}}^{(t-1)}\right] \\
&= \sum_{j \neq i_1^{l^{(t)}}, i_2^{l^{(t)}}, p} \sigma_{N_{t-1}+l^{(t)}}^{N_{t-2}+j} \mathbb{E}_{\mathbf{x},\mathbf{y}^{(:t-1)}}\left[y_j^{(t-1)} y_p^{(t-1)} y_{i_1^{l^{(t)}}}^{(t-1)} y_{i_2^{l^{(t)}}}^{(t-1)}\right] + \sigma_{N_{t-1}+l^{(t)}}^{N_{t-2}+p} \mathbb{E}_{\mathbf{x},\mathbf{y}^{(:t-1)}}\left[y_{i_1^{l^{(t)}}}^{(t-1)} y_{i_2^{l^{(t)}}}^{(t-1)}\right] \\
&\quad + \sigma_{N_{t-1}+l^{(t)}}^{N_{t-2}+i_1^{l^{(t)}}} \mathbb{E}_{\mathbf{x},\mathbf{y}^{(:t-1)}}\left[y_p^{(t-1)} y_{i_1^{l^{(t)}}}^{(t-1)}\right] + \sigma_{N_{t-1}+l^{(t)}}^{N_{t-2}+i_2^{l^{(t)}}} \mathbb{E}_{\mathbf{x},\mathbf{y}^{(:t-1)}}\left[y_p^{(t-1)} y_{i_2^{l^{(t)}}}^{(t-1)}\right] \\
&= \left(c_2^{(t-1)}(0)\right)^2 \left[\left(c_2^{(t-1)}(0)\right)^2 \right. \\
&\quad \left. + \left(1 - \left(c_2^{(t-1)}(0)\right)^2\right)\left(\sigma_{N_{t-1}+l^{(t)}}^{N_{t-2}+p} + \sigma_{N_{t-1}+l^{(t)}}^{N_{t-2}+i_1^{l^{(t)}}} + \sigma_{N_{t-1}+l^{(t)}}^{N_{t-2}+i_2^{l^{(t)}}}\right)\right].
\end{aligned} \tag{98}
$$

Since we assume $\boldsymbol{W}(0) = \boldsymbol{1}c$, all $\sigma_{N_{t-1}+l^{(t)}}^{N_{t-2}+p}$ for different $p \in [d_{t-1}]$ are equal and we can easily conclude that for $p \in \{i_1^{l^{(t)}}, i_2^{l^{(t)}}\}, p' \notin \{i_1^{l^{(t)}}, i_2^{l^{(t)}}\}$,

$$
\begin{aligned}
& \partial_{W_{N_{t-2}+p,N_{t-1}+l^{(t)}}} R - \partial_{W_{N_{t-2}+p',N_{t-1}+l^{(t)}}} R \\
& = \frac{1 - \left(c_2^{(t-1)}(0)\right)^2}{d_{t-1}} \frac{(d_{t-1}-3)\left(c_2^{(t-1)}(0)\right)^2 + 1}{d_{t-1}} > 0.
\end{aligned}
\tag{99}
$$

This means that the separation of the critical gradient component and that of the gradient are satisfied. Therefore, as shown in Thm. 3.1, only the parameters in the position $\{i_1^{l^{(t)}}, i_2^{l^{(t)}}\}$ become larger and all other parameters become smaller after the first step of the sign policy gradient:

$$
s = 1, \forall t \in [T], l^{(t)} \in [d_t] : W_{N_{t-2}+p,N_{t-1}+l^{(t)}} = \begin{cases} 1 + \eta, & p \in \{i_1^{l^{(t)}}, i_2^{l^{(t)}}\}, \\ 1 - \eta, & \text{otherwise .} \end{cases}
\tag{100}
$$

After the softmax we obtain

$$
s = 1, \forall t \in [T], l^{(t)} \in [d_t] : \sigma_{N_{t-1}+l^{(t)}}^{N_{t-2}+p} = \begin{cases} \frac{1}{2} \frac{1}{1 + \frac{d_t - 2}{2}e^{-2\eta}} := \lambda_1^{l^{(t)}}, & p \in \{i_1^{l^{(t)}}, i_2^{l^{(t)}}\}, \\ \frac{1}{d_t - 2 + 2e^{2\eta}} := \lambda_2^{l^{(t)}}, & \text{otherwise ,} \end{cases}
\tag{101}
$$

and we conclude that

$$
s = 1, \forall t \in [T], l^{(t)} \in [d_t] : \sum_q (\sigma_{N_{t-1}+l^{(t)}}^{N_{t-2}+q})^2 = c_1^{(t)}(1)
\tag{102}
$$

by conducting simple algebra. This means that the condition Eq. (91) still holds at $s = 1$. Therefore, Lem. G.1 can be applied again, and, following the computation of Eq. (96) and Eq. (98), we are able to conclude that at $s = 1$:

- if $p \in \{i_1^{l^{(t)}}, i_2^{l^{(t)}}\}$, then

$$
\begin{aligned}
& \sigma_{N_{t-1}+l^{(t)}}^{N_{t-2}+p} \mathbb{E}_{\mathbf{x},\mathbf{y}^{(:t-1)}}\left[\gamma_{l^{(t)}}^p(\mathbf{y}^{(:t-1)})\right] \\
& = \lambda_1^{l^{(t)}}\left(c_2^{(t-1)}(1)\right)^2 + \left(\lambda_1^{l^{(t)}}\right)^2\left(1 - \left(c_2^{(t-1)}(1)\right)^2\right);
\end{aligned}
\tag{103}
$$

- if $p \notin \{i_1^{l^{(t)}}, i_2^{l^{(t)}}\}$, then

$$
\begin{aligned}
& \sigma_{N_{t-1}+l^{(t)}}^{N_{t-2}+p} \mathbb{E}_{\mathbf{x},\mathbf{y}^{(:t-1)}}\left[\gamma_{l^{(t)}}^p(\mathbf{y}^{(:t-1)})\right] \\
& = \lambda_2^{l^{(t)}}\left(c_2^{(t-1)}(1)\right)^2\left[\left(c_2^{(t-1)}(1)\right)^2 + \left(1 - \left(c_2^{(t-1)}(1)\right)^2\right)\left(\lambda_2^{l^{(t)}} + 2\lambda_1^{l^{(t)}}\right)\right].
\end{aligned}
\tag{104}
$$

According to the form of the policy gradient Eq. (29), we also need to evaluate

$$
\begin{aligned}
& \mathbb{E}_{\mathbf{x},\mathbf{y}^{(:t-1)}}\left[\sum_{i=1}^{d_{t-1}} \gamma_{l^{(t)}}^i(\mathbf{y}^{(t-1)})\right] \sigma_{N_{t-1}+l^{(t)}}^{N_{t-2}+i} \\
& = 2\lambda_1^{l^{(t)}}\left[\left(c_2^{(t-1)}(1)\right)^2 + \lambda_1^{l^{(t)}}\left(1 - \left(c_2^{(t-1)}(1)\right)^2\right)\right] \\
& \quad + (d_t - 2)\lambda_2^{l^{(t)}}\left(c_2^{(t-1)}(1)\right)^2\left[\left(c_2^{(t-1)}(1)\right)^2 + \left(1 - \left(c_2^{(t-1)}(1)\right)^2\right)\left(\lambda_2^{l^{(t)}} + 2\lambda_1^{l^{(t)}}\right)\right].
\end{aligned}
\tag{105}
$$

Combined these equations, noting that $2\lambda_1^{l^{(t)}} + (d_t - 2)\lambda_2^{l^{(t)}} = 1$, we are now able to find the expression of the gradient

when $p \in \{i_1^{l^{(t)}}, i_2^{l^{(t)}}\}$:

$$
\begin{aligned}
& \partial_{W_{N_{t-2}+p, N_{t-1}+l^{(t)}}} R \\
&= \lambda_1^{l^{(t)}} \left(1 - 2\lambda_1^{l^{(t)}}\right) \left\{ \left(c_2^{(t-1)}(1)\right)^2 + \lambda_1^{l^{(t)}} \left(1 - \left(c_2^{(t-1)}(1)\right)^2\right) \right. \\
&\quad \left. - \left(c_2^{(t-1)}(1)\right)^2 \left[\left(c_2^{(t-1)}(1)\right)^2 + \left(1 - \left(c_2^{(t-1)}(1)\right)^2\right)\left(\lambda_2^{l^{(t)}} + 2\lambda_1^{l^{(t)}}\right)\right] \right\} \\
&= \lambda_1^{l^{(t)}} \left(1 - 2\lambda_1^{l^{(t)}}\right) \left(1 - \left(c_2^{(t-1)}(1)\right)^2\right) \left[\left(c_2^{(t-1)}(1)\right)^2 (d_t - 3)\, \lambda_2^{l^{(t)}} \left(c_2^{(t-1)}(1)\right)^2 + \lambda_1^{l^{(t)}}\right] \\
&> 0,
\end{aligned}
\tag{106}
$$

as $d_t > 3$ (the case for $d_t = 2$ does not need any update) and $2\lambda_1^{l^{(t)}} < 1$. Hence the policy has positive gradient at relevant positions (Eq. (106)) and negative gradient at irrelevant positions (using Eq. (34)). Similar to the proof of Thm. 3.1, we can verify that

$$
s = 2, \forall t \in [T], l^{(t)} \in [d_t] : W_{N_{t-2}+p, N_{t-1}+l^{(t)}} = \begin{cases} 1 + 2\eta, & p \in \{i_1^{l^{(t)}}, i_2^{l^{(t)}}\}, \\ 1 - 2\eta, & \text{otherwise}. \end{cases}
\tag{107}
$$

Following the same argument, we can eventually show that after $S$-steps of the sign policy gradient update the model parameters are simply

$$
W_{N_{t-2}+p, N_{t-1}+l^{(t)}}(S) = \begin{cases} 1 + \eta S, & p \in \{i_1^{l^{(t)}}, i_2^{l^{(t)}}\}, \\ 1 - \eta S, & \text{otherwise}. \end{cases}
\tag{108}
$$

After the softmax we obtain

$$
\sigma_{N_{t-1}+l^{(t)}}^{N_{t-2}+p}(S) = \begin{cases} \frac{1}{2} \frac{1}{1 + \frac{d_t-2}{2} e^{-2\eta S}}, & p \in \{i_1^{l^{(t)}}, i_2^{l^{(t)}}\}, \\ \frac{1}{d_t - 2 + 2e^{2\eta S}}, & \text{otherwise}. \end{cases}
\tag{109}
$$

$\square$

*Proof of Lem. G.1.* We prove this lemma by induction. For $t = 0$, we directly have

$$
\forall l^{(0)} \in [d_0] : \mathbb{E}_{\mathbf{x}}\left[x_{l^{(0)}}\right] = 0.
\tag{110}
$$

Denote

$$
\chi_{l^{(t)}}(\mathbf{y}^{(t-1)}) = p_{\mathbf{W}}\left(y_{l^{(t)}}^{(t)} = 1 \middle| \mathbf{y}^{(t-1)}\right) = \left(\xi_{l^{(t)}}\right)^2.
\tag{111}
$$

We first verify the claim for $t = 1$, where the expectation can be written as

$$
\begin{aligned}
\mathbb{E}_{\mathbf{x}, \mathbf{y}^{(1)}}\left[y_{l^{(1)}}^{(1)}\right] &= \sum_{\mathbf{x}} p(\mathbf{x}) \left(2\chi_{l^{(1)}}(\mathbf{x}) - 1\right) \\
&\stackrel{(i)}{=} 2 \sum_{\mathbf{x}} p(\mathbf{x}) \left(\sum_{p,q} x_p x_q \sigma_{N_0 + l^{(1)}}^{N_{-1}+p} \sigma_{N_0 + l^{(1)}}^{N_{-1}+q}\right) - 1 \\
&\stackrel{(ii)}{=} 2 \sum_{p \neq q} \mathbb{E}_{\mathbf{x}}[x_p] \mathbb{E}_{\mathbf{x}}[x_q] \sigma_{N_0 + l^{(1)}}^{N_{-1}+p} \sigma_{N_0 + l^{(1)}}^{N_{-1}+q} + 2 \sum_{m} (\sigma_{N_0 + l^{(1)}}^{N_{-1}+m})^2 - 1 \\
&\stackrel{(iii)}{=} 2 \sum_{m} (\sigma_{N_0 + l^{(1)}}^{N_{-1}+m})^2 - 1 = 2c_1^{(t)} - 1,
\end{aligned}
\tag{112}
$$

where we use the definition of $\xi_{l^{(1)}}$ in $(i)$; we use that $x_j$ for different $j$ are independent w.r.t $p(\mathbf{x})$ in $(ii)$; we apply $\mathbb{E}_{\mathbf{x}}[x_p] = 0$ in $(iii)$. Therefore, with the condition Eq. (91), the statement is established for $t = 1$. Suppose that the statement is established for $(t - 1)$-th step

$$
\forall l^{(t-1)} \in [d^{(t-1)}] : \mathbb{E}_{\mathbf{x}, \mathbf{y}^{(:t-1)}}\left[y_{l^{(t-1)}}^{(t-1)}\right] = c_2^{(t-1)},
\tag{113}
$$

then

$$\mathbb{E}_{\mathbf{x},\mathbf{y}^{(:t)}}\left[y_{l^{(t)}}^{(t)}\right]$$

$$= \sum_{\mathbf{x}} p(\mathbf{x}) \sum_{\mathbf{y}^{(:t-1)}} p_{\boldsymbol{W}}(\mathbf{y}^{(:t-1)}|\mathbf{x}) \sum_{\mathbf{y}^{(t)}} p_{\boldsymbol{W}}(\mathbf{y}^{(t)}|\mathbf{y}^{(t-1)}) y_{l^{(t)}}^{(t)}$$

$$= 2 \sum_{\mathbf{x}} p(\mathbf{x}) \sum_{\mathbf{y}^{(:t-1)}} p_{\boldsymbol{W}}(\mathbf{y}^{(:t-1)}|\mathbf{x}) \chi_{l^{(t)}}(\mathbf{y}^{(t-1)}) - 1$$

$$= 2 \sum_{p^{(t-1)} \neq q^{(t-1)}} \mathbb{E}_{\mathbf{x},\mathbf{y}^{(:t-1)},}\left[y_{p^{(t-1)}}^{(t-1)}\right] \mathbb{E}_{\mathbf{x},\mathbf{y}^{(:t-1)},}\left[y_{q^{(t-1)}}^{(t-1)}\right] \sigma_{N_{t-1}+l^{(t)}}^{N_{t-2}+p^{(t-1)}} \sigma_{N_{t-1}+l^{(t)}}^{N_{t-2}+q^{(t-1)}}$$

$$+ 2 \sum_{m^{(t-1)}} \left(\sigma_{N_{t-1}+l^{(t)}}^{N_{t-2}+m^{(t-1)}}\right)^2 - 1$$

$$= 2 \left(c_2^{(t-1)}\right)^2 \sum_{p^{(t-1)} \neq q^{(t-1)}} \sigma_{N_{t-1}+l^{(t)}}^{N_{t-2}+p^{(t-1)}} \sigma_{N_{t-1}+l^{(t)}}^{N_{t-2}+q^{(t-1)}} + 2 \sum_{m^{(t-1)}} \left(\sigma_{N_{t-1}+l^{(t)}}^{N_{t-2}+m^{(t-1)}}\right)^2 - 1$$

$$= 2 \left(c_2^{(t-1)}\right)^2 \left(\sum_{p^{(t-1)}} \sigma_{N_{t-1}+l^{(t)}}^{N_{t-2}+p^{(t-1)}}\right)^2 + 2 \left[1 - \left(c_2^{(t-1)}\right)^2\right] \sum_{m^{(t-1)}} \left(\sigma_{N_{t-1}+l^{(t)}}^{N_{t-2}+m^{(t-1)}}\right)^2 - 1$$

$$= 2 \left(c_2^{(t-1)}\right)^2 + 2 \left[1 - \left(c_2^{(t-1)}\right)^2\right] \left(c_1^{(t-1)}\right)^2 - 1$$

does not depend on $l^{(t)}$. Thus we conclude the proof. □

### G.2.2. LEARNABILITY OF SFT: PROOF OF CLAIM. 4.1

*Proof.* To prove this claim, we only need to evaluate whether the separation of the critical gradient component is satisfied, which can be done by inspecting the forms of $\gamma_{l^{(t)}}^p$ similar to the case for RL. Specifically, according to Eq. (93) and recalling that $\tilde{\mathbf{y}}^{(t)}$ is the ground-truth of $\mathbf{y}^{(t)}$ given $\mathbf{x}$, we have

$$\mathbb{E}_{\mathbf{x}}\left[\gamma_{l^{(t)}}^p(\tilde{\mathbf{y}}^{(t-1)})\right] := \mathbb{E}_{\mathbf{x}}\left[\left(\sum_{j=1}^{d_{t-1}} \sigma_{N_{t-1}+l^{(t)}}^{N_{t-2}+j} \tilde{y}_j^{(t-1)}\right) \tilde{y}_{i_1^{l^{(t)}}}^{(t-1)} \tilde{y}_{i_2^{l^{(t)}}}^{(t-1)} \tilde{y}_p^{(t-1)}\right]. \tag{114}$$

As this is similar to the case for RL, we can easily evaluate it at the relevant positions and irrelevant ones, respectively, as shown below:

- if $p \in \{i_1^{l^{(t)}}, i_2^{l^{(t)}}\}$, say $p = i_2^{l^{(t)}}$, then

$$\mathbb{E}_{\mathbf{x}}\left[\gamma_{l^{(t)}}^p(\tilde{\mathbf{y}}^{(t-1)})\right]$$

$$= \sum_{j=1}^{d_{t-1}} \sigma_{N_{t-1}+l^{(t)}}^{N_{t-2}+j} \mathbb{E}_{\mathbf{x}}\left[\tilde{y}_j^{(t-1)} \tilde{y}_{i_1^{l^{(t)}}}^{(t-1)}\right]$$

$$\stackrel{(i)}{=} \sum_{j \neq i_1^{l^{(t)}}} \sigma_{N_{t-1}+l^{(t)}}^{N_{t-2}+j} \mathbb{E}_{\mathbf{x}}\left[\tilde{y}_j^{(t-1)}\right] \mathbb{E}_{\mathbf{x}}\left[\tilde{y}_{i_1^{l^{(t)}}}^{(t-1)}\right] + \sigma_{N_{t-1}+l^{(t)}}^{N_{t-2}+i_1^{l^{(t)}}}$$

$$\stackrel{(ii)}{=} \sigma_{N_{t-1}+l^{(t)}}^{N_{t-2}+i_1^{l^{(t)}}}, \tag{115}$$

where we use that $\tilde{y}_j^{(t)}$ and $\tilde{y}_{j'}^{(t)}$ are independent if $j \neq j'$ in $(i)$ and $\mathbb{E}_{\mathbf{x}}[\tilde{y}_j^{(t-1)}] = 0$ in $(ii)$. Similarly, if $p = i_1^{l^{(t)}}$, we can similarly obtain that

$$\mathbb{E}_{\mathbf{x}}\left[\gamma_{l^{(t)}}^p(\tilde{\mathbf{y}}^{(t-1)})\right] = \sigma_{N_{t-1}+l^{(t)}}^{N_{t-2}+i_2^{l^{(t)}}}. \tag{116}$$

- if $p \notin \{i_1^{l^{(t)}}, i_2^{l^{(t)}}\}$, then (similar to the case of $p \in \{i_1^{l^{(t)}}, i_2^{l^{(t)}}\}$)

$$
\begin{aligned}
&\mathbb{E}_{\mathbf{x}}\left[\gamma_{l^{(t)}}^p(\tilde{\mathbf{y}}^{(t-1)})\right]\\
&= \sum_j \sigma_{N_{t-1}+l^{(t)}}^{N_{t-2}+j}\,\mathbb{E}_{\mathbf{x}}\left[\tilde{y}_j^{(t-1)} y_p^{(t-1)} \tilde{y}_{i_1^{l^{(t)}}}^{(t-1)} \tilde{y}_{i_2^{l^{(t)}}}^{(t-1)}\right]\\
&= \sum_{j \neq i_1^{l^{(t)}}, i_2^{l^{(t)}}, p} \sigma_{N_{t-1}+l^{(t)}}^{N_{t-2}+j}\,\mathbb{E}_{\mathbf{x}}\left[\tilde{y}_j^{(t-1)} y_p^{(t-1)} \tilde{y}_{i_1^{l^{(t)}}}^{(t-1)} \tilde{y}_{i_2^{l^{(t)}}}^{(t-1)}\right] + \sigma_{N_{t-1}+l^{(t)}}^{N_{t-2}+p}\,\mathbb{E}_{\mathbf{x}}\left[\tilde{y}_{i_1^{l^{(t)}}}^{(t-1)} \tilde{y}_{i_2^{l^{(t)}}}^{(t-1)}\right]\\
&\quad + \sigma_{N_{t-1}+l^{(t)}}^{N_{t-2}+i_1^{l^{(t)}}}\,\mathbb{E}_{\mathbf{x}}\left[\tilde{y}_p^{(t-1)} \tilde{y}_{i_1^{l^{(t)}}}^{(t-1)}\right] + \sigma_{N_{t-1}+l^{(t)}}^{N_{t-2}+i_2^{l^{(t)}}}\,\mathbb{E}_{\mathbf{x}}\left[\tilde{y}_p^{(t-1)} \tilde{y}_{i_2^{l^{(t)}}}^{(t-1)}\right] = 0. 
\end{aligned}
\tag{117}
$$

Therefore, for any $t \in [T]$ and any $l^{(t)} \in [d_t]$,

$$
\forall p \in \{i_1^{l^{(t)}}, i_2^{l^{(t)}}\}, p' \in [d_{t-1}]\setminus\{i_1^{l^{(t)}}, i_2^{l^{(t)}}\}:\ \mathbb{E}_{\mathbf{x}}\left[\gamma_{l^{(t)}}^p(\tilde{\mathbf{y}}^{(t-1)}) - \gamma_{l^{(t)}}^{p'}(\tilde{\mathbf{y}}^{(t-1)})\right] > 0,
\tag{118}
$$

which is exactly the separation condition in Thm. 3.3, and hence the learnability of SFT is established for $k$-PARITY. $\square$

### G.3. Proofs of Sec. 4.2

We discuss the learnability of $k$-AND for transformers via RL or SFT in App. G.3.1, and discuss that of $k$-OR in App. G.3.2.

#### G.3.1. AND

As in the case for $k$-PARITY, we start with evaluating the formulation of the critical gradient component $\gamma_{l^{(t)}}^p(\mathbf{y}^{(t-1)})$, which can then be used to verify whether the separation condition in Thm. 3.1 or Thm. 3.3 is satisfied to indicate the learnability. Recall that for $k$-AND (Sec. 2.1),

$$
\psi'(\xi_{l^{(t)}}) = \mathbb{I}(\xi_{l^{(t)}} > 0),
\tag{119}
$$

$$
\phi\left(y_{i_1^{l^{(t)}}}^{(t-1)}, y_{i_2^{l^{(t)}}}^{(t-1)}\right) = \frac{y_{i_1^{l^{(t)}}}^{(t-1)} y_{i_2^{l^{(t)}}}^{(t-1)} + y_{i_1^{l^{(t)}}}^{(t-1)} + y_{i_2^{l^{(t)}}}^{(t-1)} - 1}{2},
\tag{120}
$$

where $\mathbb{I}(\cdot)$ is the indictor function. This gives us the expression of $\gamma$ (see Eq. (7) for its definition) as follows

$$
\begin{aligned}
&\gamma_{l^{(t)}}^p(\mathbf{y}^{(t-1)})\\
&= \mathbb{I}(\xi_{l^{(t)}} > 0)\left(y_{i_1^{l^{(t)}}}^{(t-1)} y_{i_2^{l^{(t)}}}^{(t-1)} y_p^{(t-1)} + y_{i_1^{l^{(t)}}}^{(t-1)} y_p^{(t-1)} + y_{i_2^{l^{(t)}}}^{(t-1)} y_p^{(t-1)} - y_p^{(t-1)}\right).
\end{aligned}
\tag{121}
$$

We now check the separation for the critical gradient component $\gamma_{l^{(t)}}^p$.

**RL.** Given $t \in [T]$, when $p \in \{i_1^{l^{(t)}}, i_2^{l^{(t)}}\}$, say $i_2^{l^{(t)}}$, according to Eq. (121), we have (the case for $i_1^{l^{(t)}}$ is similar)

$$
\mathbb{E}_{\mathbf{y}^{(:t-1)}}[\gamma_{l^{(t)}}^p(\mathbf{y}^{(t-1)})] = \frac{1}{2}\mathbb{E}_{\mathbf{y}^{(:t-1)}}\left[\mathbb{I}(\xi_{l^{(t)}} > 0) y_{i_1^{l^{(t)}}}^{(t-1)} y_{i_2^{l^{(t)}}}^{(t-1)}\right] + \frac{1}{2}\mathbb{E}_{\mathbf{y}^{(:t-1)}}\left[\mathbb{I}(\xi_{l^{(t)}} > 0)\right];
\tag{122}
$$

as a comparison, when $p \notin \{i_1^{l^{(t)}}, i_2^{l^{(t)}}\}$, we obtain that

$$
\begin{aligned}
&\mathbb{E}_{\mathbf{y}^{(:t-1)}}\left[\gamma_{l^{(t)}}^p(\mathbf{y}^{(t-1)})\right]\\
&= \frac{1}{2}\mathbb{E}_{\mathbf{y}^{(:t-1)}}\left[\mathbb{I}(\xi_{l^{(t)}} > 0)\left(y_{i_1^{l^{(t)}}}^{(t-1)} y_{i_2^{l^{(t)}}}^{(t-1)} y_p^{(t-1)} + y_{i_1^{l^{(t)}}}^{(t-1)} y_p^{(t-1)} + y_{i_2^{l^{(t)}}}^{(t-1)} y_p^{(t-1)} - y_p^{(t-1)}\right)\right].
\end{aligned}
\tag{123}
$$

It is now left for us to compare the difference between these critical gradient components. For convenience and ease of notation, we define $A := \{i_1^{l^{(t)}}, i_2^{l^{(t)}}, p\}$ and $\mathbf{y}_{\not{A}}^{(t-1)}$ as the sequence $\mathbf{y}^{(t-1)}$ without the tokens $y_{i_1^{l^{(t)}}}^{(t-1)}, y_{i_2^{l^{(t)}}}^{(t-1)}$, and $y_p^{(t-1)}$.

In addition, let $D_\lambda := \{\mathbf{y}_{\cancel{A}}^{(t-1)} | \sum_j y_{\cancel{A},j}^{(t-1)} \geq \lambda\}$, i.e., the sum of tokens of $\mathbf{y}_{\cancel{A}}^{(t-1)}$ is larger than or equal to $\lambda$. Finally, we simplify $p_{\mathbf{W}}$ as $p$ with $p_{\pm} = p(y_j^{(t-1)} = \pm 1 | \mathbf{y}^{(:t-2)})$. With these new symbols, we define four random variables depending on $\mathbf{y}^{(:t-2)}$ as follows:

$$a = \sum_{\mathbf{y}_{\cancel{A}}^{(t-1)} \in D_{-3}} p_+^3 p(\mathbf{y}_{\cancel{A}}^{(t-1)} | \mathbf{y}^{(:t-2)}), \tag{124}$$

$$b = \sum_{\mathbf{y}_{\cancel{A}}^{(t-1)} \in D_{-1}} p_+^2 p_- p(\mathbf{y}_{\cancel{A}}^{(t-1)} | \mathbf{y}^{(:t-2)}), \tag{125}$$

$$c = \sum_{\mathbf{y}_{\cancel{A}}^{(t-1)} \in D_{+1}} p_+ p_-^2 p(\mathbf{y}_{\cancel{A}}^{(t-1)} | \mathbf{y}^{(:t-2)}), \tag{126}$$

$$d = \sum_{\mathbf{y}_{\cancel{A}}^{(t-1)} \in D_{+3}} p_-^3 p(\mathbf{y}_{\cancel{A}}^{(t-1)} | \mathbf{y}^{(:t-2)}). \tag{127}$$

In Eq. (122) and Eq. (123), all $y_j$ can be treated equally as they are independent of each other. We omit all the subscripts and write the difference between $p \in \{i_1^{l^{(t)}}, i_2^{l^{(t)}}\}$ and $p' \in [d_{t-1}] \setminus \{i_1^{l^{(t)}}, i_2^{l^{(t)}}\}$ as

$$\mathbb{E}_{\mathbf{y}^{(:t-1)}}[\Delta] := \mathbb{E}_{\mathbf{y}^{(:t-1)}}[\gamma_{l^{(t)}}^p(\mathbf{y}^{(t-1)})] - \mathbb{E}_{\mathbf{y}^{(:t-1)}}[\gamma_{l^{(t)}}^{p'}(\mathbf{y}^{(t-1)})]$$
$$\propto \underbrace{\mathbb{E}_{\mathbf{y}^{(:t-1)}}[\mathbb{I}(\xi_{l^{(t)}} > 0)]}_{(i)} + \underbrace{\mathbb{E}_{\mathbf{y}^{(:t-1)}}[\mathbb{I}(\xi_{l^{(t)}} > 0)y]}_{(ii)} \tag{128}$$
$$- \underbrace{\mathbb{E}_{\mathbf{y}^{(:t-1)}}[\mathbb{I}(\xi_{l^{(t)}} > 0)yy]}_{(iii)} - \underbrace{\mathbb{E}_{\mathbf{y}^{(:t-1)}}[\mathbb{I}(\xi_{l^{(t)}} > 0)yyy]}_{(iv)}.$$

We below analyze each term of $\Delta$. The first term and the last term can be easily evaluated as follows:

$$(i) = \sum_{\mathbf{y}^{(:t-2)}} p(\mathbf{y}^{(:t-2)}) \left[ \sum_{\mathbf{y}_{\cancel{A}}^{(t-1)} \in D_{-3}} p_+^3 p(\mathbf{y}_{\cancel{A}}^{(t-1)} | \mathbf{y}^{(:t-2)}) \right.$$
$$+ 3 \sum_{\mathbf{y}_{\cancel{A}}^{(t-1)} \in D_{-1}} p_+^2 p_- p(\mathbf{y}_{\cancel{A}}^{(t-1)} | \mathbf{y}^{(:t-2)})$$
$$\left. + 3 \sum_{\mathbf{y}_{\cancel{A}}^{(t-1)} \in D_{+1}} p_+ p_-^2 p(\mathbf{y}_{\cancel{A}}^{(t-1)} | \mathbf{y}^{(:t-2)}) + \sum_{\mathbf{y}_{\cancel{A}}^{(t-1)} \in D_{+3}} p_-^3 p(\mathbf{y}_{\cancel{A}}^{(t-1)} | \mathbf{y}^{(:t-2)}) \right] \tag{129}$$
$$= \mathbb{E}_{\mathbf{y}^{(:t-2)}}[a + 3b + 3c + d],$$

and

$$(iv) = \sum_{\mathbf{y}^{(:t-2)}} p(\mathbf{y}^{(:t-2)}) \left[ \sum_{\mathbf{y}_{\cancel{A}}^{(t-1)} \in D_{-3}} p_+^3 p(\mathbf{y}_{\cancel{A}}^{(t-1)} | \mathbf{y}^{(:t-2)}) \right.$$
$$- 3 \sum_{\mathbf{y}_{\cancel{A}}^{(t-1)} \in D_{-1}} p_+^2 p_- p(\mathbf{y}_{\cancel{A}}^{(t-1)} | \mathbf{y}^{(:t-2)}) + 3 \sum_{\mathbf{y}_{\cancel{A}}^{(t-1)} \in D_{+1}} p_+ p_-^2 p(\mathbf{y}_{\cancel{A}}^{(t-1)} | \mathbf{y}^{(:t-2)})$$
$$\left. - \sum_{\mathbf{y}_{\cancel{A}}^{(t-1)} \in D_{+3}} p_-^3 p(\mathbf{y}_{\cancel{A}}^{(t-1)} | \mathbf{y}^{(:t-2)}) \right] \tag{130}$$
$$= \mathbb{E}_{\mathbf{y}^{(:t-2)}}[a - 3b + 3c - d].$$

For the rest terms, we express them with the four constants $a, b, c, d$ defined above such that we are able to evaluate $\Delta$. To this end, let $\mathbf{y}_{\cancel{1}}^{(t-1)}$ denote the sequence obtained by removing one token in $A$ from $\mathbf{y}^{(t-1)}$, $\mathbf{y}_{\cancel{2}}^{(t-1)}$ be obtained by removing

two tokens of $A$ from $\mathbf{y}^{(t-1)}$, and $D_\lambda$ be defined as earlier. This allows us to establish the following relation:

$$\sum_{\mathbf{y}^{(t-1)}_{\mathcal{V}} \in D_0} p(\mathbf{y}^{(t-1)}_{\mathcal{V}} | \mathbf{y}^{(:t-2)})$$
$$= \sum_{\mathbf{y}^{(t-1)}_{\mathcal{A}} \in D_{-1}} p_+ p(\mathbf{y}^{(t-1)}_{\mathcal{A}} | \mathbf{y}^{(:t-2)}) + \sum_{\mathbf{y}^{(t-1)}_{\mathcal{A}} \in D_{+1}} p_- p(\mathbf{y}^{(t-1)}_{\mathcal{A}} | \mathbf{y}^{(:t-2)}), \tag{131}$$

which can be further decomposed to be finally expressed by $a, b, c,$ and $d$. With this relation, we are able to derive

$$(ii) = \sum_{\mathbf{y}^{(:t-2)}} p(\mathbf{y}^{(:t-2)}) \Big[ \sum_{\mathbf{y}^{(t-1)}_{\mathcal{V}} \in D_{-1}} p_+ p(\mathbf{y}^{(t-1)}_{\mathcal{V}} | \mathbf{y}^{(:t-2)}) - \sum_{\mathbf{y}^{(t-1)}_{\mathcal{V}} \in D_{+1}} p_- p(\mathbf{y}^{(t-1)}_{\mathcal{V}} | \mathbf{y}^{(:t-2)}) \Big]$$
$$= \sum_{\mathbf{y}^{(:t-2)}} p(\mathbf{y}^{(:t-2)}) \Big[ \sum_{\mathbf{y}^{(t-1)}_{\mathcal{A}} \in D_{-3}} p_+^3 p(\mathbf{y}^{(t-1)}_{\mathcal{A}} | \mathbf{y}^{(:t-2)}) + \sum_{\mathbf{y}^{(t-1)}_{\mathcal{A}} \in D_{-1}} p_+^2 p_- p(\mathbf{y}^{(t-1)}_{\mathcal{A}} | \mathbf{y}^{(:t-2)})$$
$$- \sum_{\mathbf{y}^{(t-1)}_{\mathcal{A}} \in D_{+1}} p_+ p_-^2 p(\mathbf{y}^{(t-1)}_{\mathcal{A}} | \mathbf{y}^{(:t-2)}) - \sum_{\mathbf{y}^{(t-1)}_{\mathcal{A}} \in D_{+3}} p_-^3 p(\mathbf{y}^{(t-1)}_{\mathcal{A}} | \mathbf{y}^{(:t-2)}) \Big]$$
$$= \mathbb{E}_{\mathbf{y}^{(:t-2)}} [a + b - c - d]. \tag{132}$$

Similarly, for the second term we can write

$$(iii) = \sum_{\mathbf{y}^{(:t-2)}} p(\mathbf{y}^{(:t-2)}) \Big[ \sum_{\mathbf{y}^{(t-1)}_{\mathcal{A}} \in D_{-2}} p_+^2 p(\mathbf{y}^{(t-1)}_{\mathcal{A}} | \mathbf{y}^{(:t-2)})$$
$$- 2 \sum_{\mathbf{y}^{(t-1)}_{\mathcal{A}} \in D_0} p_+ p_- p(\mathbf{y}^{(t-1)}_{\mathcal{A}} | \mathbf{y}^{(:t-2)}) + \sum_{\mathbf{y}^{(t-1)}_{\mathcal{A}} \in D_{+1}} p_-^2 p(\mathbf{y}^{(t-1)}_{\mathcal{A}} | \mathbf{y}^{(:t-2)})$$
$$= \sum_{\mathbf{y}^{(:t-2)}} p(\mathbf{y}^{(:t-2)}) \Big[ \sum_{\mathbf{y}^{(t-1)}_{\mathcal{A}} \in D_{-3}} p_+^3 p(\mathbf{y}^{(t-1)}_{\mathcal{A}} | \mathbf{y}^{(:t-2)})$$
$$- \sum_{\mathbf{y}^{(t-1)}_{\mathcal{A}} \in D_{-1}} p_+^2 p_- p(\mathbf{y}^{(t-1)}_{\mathcal{A}} | \mathbf{y}^{(:t-2)}) - \sum_{\mathbf{y}^{(t-1)}_{\mathcal{A}} \in D_{+1}} p_+ p_-^2 p(\mathbf{y}^{(t-1)}_{\mathcal{A}} | \mathbf{y}^{(:t-2)})$$
$$+ \sum_{\mathbf{y}^{(t-1)}_{\mathcal{A}} \in D_{+3}} p_-^3 p(\mathbf{y}^{(t-1)}_{\mathcal{A}} | \mathbf{y}^{(:t-2)}) \Big]$$
$$= \mathbb{E}_{\mathbf{y}^{(:t-2)}} [a - b - c + d]. \tag{133}$$

Now combining these terms, we can easily show that

$$\mathbb{E}_{\mathbf{y}^{(:t-1)}} [\Delta] \propto 8b > 0; \tag{134}$$

hence the separation of the critical gradient component is satisfied, and the learnability is guaranteed for RL.

**SFT.** Compared to the case for fine-tuning via RL discussed above, the only difference brought by evaluating the critical gradient component for SFT is that we are now computing

$$\mathbb{E}_{\mathbf{x}} [\gamma^p_{l(t)} (\tilde{\mathbf{y}}^{(t-1)})], \tag{135}$$

where $\tilde{\mathbf{y}}^{(t-1)}$ is the ground-truth of $\mathbf{y}^{(t-1)}$. This is equivalent to changing $\mathbf{y}^{(t-1)}$ to $\tilde{\mathbf{y}}^{(t-1)}$ and all the dependence on $\mathbf{y}^{(:t-2)}$ to the dependence on $\mathbf{x}$. It can be easily seen that these do not change the conclusion that $\Delta \propto 8b > 0$ (as it is valid for any distribution $p(\mathbf{y}^{(t-1)} | \mathbf{y}^{(:t-2)})$ according to the calculation of Eq. (128), which includes $p_{\text{True}}(\mathbf{y}^{(t-1)} | \mathbf{x})$—the ground-truth distribution—as an example); thus the separation of the critical gradient component in Thm. 3.3 is also satisfied, which further guarantees the learnability via SFT.

G.3.2. OR

**RL.** The analysis for $k$-OR is highly similar to that for $k$-AND in App. G.3.1, except for that $\psi(\cdot)$ and $\phi_2(\cdot, \cdot)$ have different formulations. Specifically,

$$\psi'(\xi_{l^{(t)}}) = \mathbb{I}(\xi_{l^{(t)}} < 0), \tag{136}$$

$$\phi\left(y_{i_1^{l^{(t)}}}^{(t-1)}, y_{i_2^{l^{(t)}}}^{(t-1)}\right) = \frac{-y_{i_1^{l^{(t)}}}^{(t-1)}y_{i_2^{l^{(t)}}}^{(t-1)} + y_{i_1^{l^{(t)}}}^{(t-1)} + y_{i_2^{l^{(t)}}}^{(t-1)} + 1}{2}. \tag{137}$$

This gives us the expression for $\gamma$

$$
\begin{aligned}
&\mathbb{E}_{\mathbf{y}^{(:t-1)}}\left[\gamma_{l^{(t)}}^p\right]\\
&:= \frac{1}{2}\mathop{\mathbb{E}}_{\mathbf{y}^{(:t-1)}}\left[\mathbb{I}(\xi_{l^{(t)}} < 0)\left(-y_{i_1^{l^{(t)}}}^{(t-1)}y_{i_2^{l^{(t)}}}^{(t-1)}y_p^{(t-1)} + y_{i_1^{l^{(t)}}}^{(t-1)}y_p^{(t-1)} + y_{i_2^{l^{(t)}}}^{(t-1)}y_p^{(t-1)} + y_p^{(t-1)}\right)\right].
\end{aligned}
\tag{138}
$$

When $p \in \{i_1^{l^{(t)}}, i_2^{l^{(t)}}\}$, say $j_2$,

$$
\begin{aligned}
\mathbb{E}_{\mathbf{y}^{(:t-1)}}[\gamma_{l^{(t)}}^p] &:= \frac{1}{2}\mathop{\mathbb{E}}_{\mathbf{y}^{(:t-1)}}\left[\mathbb{I}(\xi_{l^{(t)}} < 0)\left(-y_{i_1^{l^{(t)}}}^{(t-1)} + y_{i_1^{l^{(t)}}}^{(t-1)}y_{i_2^{l^{(t)}}}^{(t-1)} + 1 + y_{i_2^{l^{(t)}}}^{(t-1)}\right)\right]\\
&= \frac{1}{2}\mathop{\mathbb{E}}_{\mathbf{y}^{(:t-1)}}\left[\mathbb{I}(\xi_{l^{(t)}} < 0)y_{i_1^{l^{(t)}}}^{(t-1)}y_{i_2^{l^{(t)}}}^{(t-1)}\right] + \frac{1}{2}\mathop{\mathbb{E}}_{\mathbf{y}^{(:t-1)}}\left[\mathbb{I}(\xi_{l^{(t)}} < 0)\right].
\end{aligned}
\tag{139}
$$

As a comparison, when $p \notin \{i_1^{l^{(t)}}, i_2^{l^{(t)}}\}$, $\gamma_{l^{(t)}}^p$ is simply Eq. (138). As in the case of $k$-AND (App. G.3.1), we also use the definition of $\mathbf{y}_{\mathcal{A}}^{(t-1)}$ and denote $\bar{D}_\lambda := \{\mathbf{y}_{\mathcal{A}}^{(t-1)}|\sum_j y_{\mathcal{A},j}^{(t-1)} \leq \lambda\}$. Furthermore, we also define four random variables depending on $\mathbf{y}^{(:t-2)}$, i.e.,

$$\bar{a} = \sum_{\mathbf{y}_{\mathcal{A}}^{(t-1)}\in\bar{D}_{-3}} p_+^3 p(\mathbf{y}_{\mathcal{A}}^{(t-1)}|\mathbf{y}^{(:t-2)}), \tag{140}$$

$$\bar{b} = \sum_{\mathbf{y}_{\mathcal{A}}^{(t-1)}\in\bar{D}_{-1}} p_+^2 p_- p(\mathbf{y}_{\mathcal{A}}^{(t-1)}|\mathbf{y}^{(:t-2)}), \tag{141}$$

$$\bar{c} = \sum_{\mathbf{y}_{\mathcal{A}}^{(t-1)}\in\bar{D}_{+1}} p_+ p_-^2 p(\mathbf{y}_{\mathcal{A}}^{(t-1)}|\mathbf{y}^{(:t-2)}), \tag{142}$$

$$\bar{d} = \sum_{\mathbf{y}_{\mathcal{A}}^{(t-1)}\in\bar{D}_{+3}} p_-^3 p(\mathbf{y}_{\mathcal{A}}^{(t-1)}|\mathbf{y}^{(:t-2)}). \tag{143}$$

In this case, the difference of the critical gradient component has the form of

$$
\mathbb{E}_{\mathbf{y}^{(:t-1)}}[\bar{\Delta}] \propto \underbrace{\mathop{\mathbb{E}}_{\mathbf{y}^{(:t-1)}}\left[\mathbb{I}(\xi_{l^{(t)}} < 0)\right]}_{(i)} - \underbrace{\mathop{\mathbb{E}}_{\mathbf{y}^{(:t-1)}}\left[\mathbb{I}(\xi_{l^{(t)}} < 0)y\right]}_{(ii)}\\
- \underbrace{\mathop{\mathbb{E}}_{\mathbf{y}^{(:t-1)}}\left[\mathbb{I}(\xi_{l^{(t)}} < 0)yy\right]}_{(iii)} + \underbrace{\mathop{\mathbb{E}}_{\mathbf{y}^{(:t-1)}}\left[\mathbb{I}(\xi_{l^{(t)}} < 0)yyy\right]}_{(iv)}.
\tag{144}
$$

Each term is computed as follows (similar to App. G.3.1):

$$
\begin{aligned}
(i) &= \sum_{\mathbf{y}^{(:t-2)}} p(\mathbf{y}^{(:t-2)})\Big[\sum_{\mathbf{y}_{\mathcal{A}}^{(t-1)}\in\bar{D}_{-3}} p_+^3 p(\mathbf{y}_{\mathcal{A}}^{(t-1)}|\mathbf{y}^{(:t-2)}) + 3\sum_{\mathbf{y}_{\mathcal{A}}^{(t-1)}\in\bar{D}_{-1}} p_+^2 p_- p(\mathbf{y}_{\mathcal{A}}^{(t-1)}|\mathbf{y}^{(:t-2)})\\
&\quad + 3\sum_{\mathbf{y}_{\mathcal{A}}^{(t-1)}\in\bar{D}_{+1}} p_+ p_-^2 p(\mathbf{y}_{\mathcal{A}}^{(t-1)}|\mathbf{y}^{(:t-2)}) + \sum_{\mathbf{y}_{\mathcal{A}}^{(t-1)}\in\bar{D}_{+3}} p_-^3 p(\mathbf{y}_{\mathcal{A}}^{(t-1)}|\mathbf{y}^{(:t-2)})\Big]\\
&= \mathbb{E}_{\mathbf{y}^{(:t-2)}}[\bar{a} + 3\bar{b} + 3\bar{c} + \bar{d}];
\end{aligned}
\tag{145}
$$

$$(ii) = \sum_{\mathbf{y}^{(:t-2)}} p(\mathbf{y}^{(:t-2)}) \Big[ \sum_{\mathbf{y}_{\mathcal{A}}^{(t-1)} \in \bar{D}_{-3}} p_+^3 p(\mathbf{y}_{\mathcal{A}}^{(t-1)} | \mathbf{y}^{(:t-2)}) + \sum_{\mathbf{y}_{\mathcal{A}}^{(t-1)} \in \bar{D}_{-1}} p_+^2 p_- p(\mathbf{y}_{\mathcal{A}}^{(t-1)} | \mathbf{y}^{(:t-2)})$$

$$- \sum_{\mathbf{y}_{\mathcal{A}}^{(t-1)} \in \bar{D}_{+1}} p_+ p_-^2 p(\mathbf{y}_{\mathcal{A}}^{(t-1)} | \mathbf{y}^{(:t-2)}) - \sum_{\mathbf{y}_{\mathcal{A}}^{(t-1)} \in \bar{D}_{+3}} p_-^3 p(\mathbf{y}_{\mathcal{A}}^{(t-1)} | \mathbf{y}^{(:t-2)}) \Big]$$

$$= \mathbb{E}_{\mathbf{y}^{(:t-2)}} [\bar{a} + \bar{b} - \bar{c} - \bar{d}]; \tag{146}$$

$$(iii) = \sum_{\mathbf{y}^{(:t-2)}} p(\mathbf{y}^{(:t-2)}) \Big[ \sum_{\mathbf{y}_{\mathcal{A}}^{(t-1)} \in \bar{D}_{-3}} p_+^3 p(\mathbf{y}_{\mathcal{A}}^{(t-1)} | \mathbf{y}^{(:t-2)})$$

$$- \sum_{\mathbf{y}_{\mathcal{A}}^{(t-1)} \in \bar{D}_{-1}} p_+^2 p_- p(\mathbf{y}_{\mathcal{A}}^{(t-1)} | \mathbf{y}^{(:t-2)}) - \sum_{\mathbf{y}_{\mathcal{A}}^{(t-1)} \in \bar{D}_{+1}} p_+ p_-^2 p(\mathbf{y}_{\mathcal{A}}^{(t-1)} | \mathbf{y}^{(:t-2)})$$

$$+ \sum_{\mathbf{y}_{\mathcal{A}}^{(t-1)} \in \bar{D}_{+3}} p_-^3 p(\mathbf{y}_{\mathcal{A}}^{(t-1)} | \mathbf{y}^{(:t-2)}) \Big]$$

$$= \mathbb{E}_{\mathbf{y}^{(:t-2)}} [\bar{a} - \bar{b} - \bar{c} + \bar{d}]; \tag{147}$$

and

$$(iv) = \sum_{\mathbf{y}^{(:t-2)}} p(\mathbf{y}^{(:t-2)}) \Big[ \sum_{\mathbf{y}_{\mathcal{A}}^{(t-1)} \in \bar{D}_{-3}} p_+^3 p(\mathbf{y}_{\mathcal{A}}^{(t-1)} | \mathbf{y}^{(:t-2)})$$

$$- 3 \sum_{\mathbf{y}_{\mathcal{A}}^{(t-1)} \in \bar{D}_{-1}} p_+^2 p_- p(\mathbf{y}_{\mathcal{A}}^{(t-1)} | \mathbf{y}^{(:t-2)}) + 3 \sum_{\mathbf{y}_{\mathcal{A}}^{(t-1)} \in \bar{D}_{+1}} p_+ p_-^2 p(\mathbf{y}_{\mathcal{A}}^{(t-1)} | \mathbf{y}^{(:t-2)})$$

$$- \sum_{\mathbf{y}_{\mathcal{A}}^{(t-1)} \in \bar{D}_{+3}} p_-^3 p(\mathbf{y}_{\mathcal{A}}^{(t-1)} | \mathbf{y}^{(:t-2)}) \Big]$$

$$= \mathbb{E}_{\mathbf{y}^{(:t-2)}} [\bar{a} - 3\bar{b} + 3\bar{c} - \bar{d}]. \tag{148}$$

Combining these expressions gives us the final result

$$\mathbb{E}_{\mathbf{y}^{(:t-1)}} [\bar{\Delta}] \propto 8c > 0; \tag{149}$$

hence the separation of the critical gradient in Thm. 3.1 is satisfied such that $k$-OR can also be learned perfectly via RL.

**SFT.** As discussed earlier in App. G.3.1, compared to the case for fine-tuning via RL, the only difference brought by evaluating the critical gradient component for SFT is that we are now computing

$$\mathbb{E}_{\mathbf{x}} [\gamma_{l(t)}^p (\tilde{\mathbf{y}}^{(t-1)})], \tag{150}$$

where $\tilde{\mathbf{y}}^{(t-1)}$ is the ground-truth of $\mathbf{y}^{(t-1)}$, which does not change the conclusion that $\bar{\Delta} \propto 8c > 0$; thus the separation of the critical gradient component in Thm. 3.3 is also satisfied, which further guarantees the learnability via SFT.

## H. Exact dynamics of RL CoT with SFT-like ground truth labels

In the **Intuitive analysis** paragraphs of the main text, it is argued that the difference in the learning curve of RL and SFT stems from how the ground truths are set: in RL the ground truth labels for the $t$-th step are computed from the output of the $(t-1)$-step, whereas in SFT all the ground truth labels are directly computed from the initial inputs. However, RL and SFT also differ in how the CoT outputs are generated—probabilistically for RL and deterministically for SFT. To eliminate this confounding factor, we consider a model where outputs are generated like RL and ground truths are computed like SFT, and demonstrate that the learning curve is still step-wise just like the SFT behavior discussed in the main text.

Let us consider RL generation probability for $k$-parity (table 1 and Eq. (3)):

$$p_{\boldsymbol{W}}(y_{l(t)}^{(t)} | \mathbf{y}^{(t-1)}) = \left( \sum_{i=1}^{d_{t-1}} y_i^{(t-1)} \sigma_{N_{t-1}+l(t)}^{N_{t-2}+i} \right)^2 \tag{151}$$

where $\sigma_j^i := \exp(W_{i,j}) / \sum_m \exp(W_{m,j})$.

Let us use a reward function:

$$R(\boldsymbol{W}) := \mathbb{E}_{\mathbf{x} \sim \text{uniform}} \mathbb{E}_{\mathbf{y} \sim p_{\boldsymbol{W}}(\cdot | \mathbf{x})} [r(\mathbf{x}, \mathbf{y})],$$

$$r(\mathbf{x}, \mathbf{y}) := \sum_{t=1}^{T} r_t \left( \mathbf{y}^{(t)}, \bar{\mathbf{y}}^{(t)} \right),$$

$$r_t \left( \mathbf{y}^{(t)}, \bar{\mathbf{y}}^{(t)} \right) := \frac{1}{k-1} \sum_{j=1}^{d_t} y_j^{(t)} \bar{y}_j^{(t)}, \tag{152}$$

which is almost identical to the RL reward Eq. (5), except here we use $\tilde{y}_j^{(t)}$, the correct label of $y_j^{(t)}$ computed from the initial input $\mathbf{x}$ as in SFT (Sec. 3.2), whereas in the RL reward we use $\bar{y}_j^{(t)}$, the correct label computed from $\mathbf{y}^{(:t-1)}$ (generated tokens at $(t-1)$th CoT step). We will compute the exact dynamics of 1-step and 2-step CoT in terms of simple differential equations. For more CoT steps, the same technique applies but the equations get progressively more complex. We are going to see how step-wise learning behaviour arises from an ODE perspective. We stress that there is no teacher forcing in this setup.

## H.1. One-step CoT

Consider a one-step chain, i.e., the bottom layer is the $d$ input tokens $\mathbf{x}$, $k$ of which are used to compute ground truth parities and the second layer consists of $k/2$ generated tokens. In this simple case,

$$p_{\boldsymbol{W}}(y_{l^{(1)}}^{(1)} = 1 | \mathbf{x}) = \left( \sum_{j=1}^{d} x_j \sigma_{d+l}^j \right)^2 \tag{153}$$

and the ground truth is

$$\tilde{y}_l^{(1)} = x_{i_1^l} x_{i_2^l}, \qquad l = 1, 2, \ldots, k/2, \tag{154}$$

where $i_1^l, i_2^l$ are the two positions of $\mathbf{x}$ tokens which $y_l$ truly depends on, and we have abbreviated $l^{(1)}$ simply as $l$. Therefore

$$\mathbb{E}_{\mathbf{x}, \mathbf{y}}[r(\mathbf{x}, \mathbf{y})] = \frac{1}{k-1} \mathbb{E}_x \left( \sum_{l=1}^{k/2} x_{i_1^l} x_{i_2^l} \left[ 2 \left( \sum_j x_j \sigma_{d+l}^j \right)^2 - 1 \right] \right). \tag{155}$$

Now use

$$\mathbb{E}_{\mathbf{x} \sim \text{uniform}}[x_{i_1} x_{i_2} x_{j_1} x_{j_2}] = \delta_{i_1 j_1} \delta_{i_2 j_2} + \delta_{i_2 j_1} \delta_{i_1 j_2} \qquad i_1 \neq i_2 \tag{156}$$

we obtain

$$\mathbb{E}_{\mathbf{x}, \mathbf{y}}[r(\mathbf{x}, \mathbf{y})] = \frac{4}{k-1} \sum_l \sigma_{d+l}^{i_1^l} \sigma_{d+l}^{i_2^l}. \tag{157}$$

Gradient decent with small learning rate easily follows:

$$\dot{W}_{p,l}(t) = \begin{cases} \frac{4\sigma_{d+l}^{j_1} \sigma_{d+l}^{j_2}}{k-1} \left[ 1 - 2\sigma_{d+l}^p \right] & p \in \{i_1^l, i_2^l\}, \\ -\frac{8\sigma_{d+l}^p \sigma_{d+l}^{i_1^l} \sigma_{d+l}^{i_2^l}}{k-1} & \text{otherwise.} \end{cases} \tag{158}$$

This set of ODE has a simple solution for equal weight initializations

$$e^{W_{p,l}(0)} = \frac{1}{z_0}, \tag{159}$$

which is

$$\sigma_{d+l}^p(t) = \begin{cases} \frac{(z(t)/z_0)^{\frac{d}{2}}}{(d-2) + 2(z(t)/z_0)^{\frac{d}{2}}}, & p \in \{i_1^l, i_2^l\} \\ \frac{1}{(d-2) + 2(z(t)/z_0)^{\frac{d}{2}}}, & p \notin \{i_1^l, i_2^l\}, \end{cases} \tag{160}$$

where $z(t)$ is solved as

$$\frac{t}{k-1} = \frac{(2-d)^3}{8d}\left[\left(\frac{z_0}{z}\right)^d - 1\right] - \frac{3(2-d)^2}{2d}\left[\left(\frac{z_0}{z}\right)^{\frac{d}{2}} - 1\right] + \frac{2}{d}\left[\left(\frac{z}{z_0}\right)^{\frac{d}{2}} - 1\right] + \frac{3}{2}(d-2)\log\frac{z}{z_0}. \tag{161}$$

Since $d \geq 2$, $t \to +\infty$ implies that $z \to +\infty$. The right-hand side is dominated by the third term and $z/z_0 \sim \left[\frac{d}{2(k-1)}t\right]^{2/d}$. Therefore as $t \to \infty$,

$$\sigma_{d+l}^{i_1^l} = \sigma_{d+l}^{i_2^l} \sim \frac{1}{2}\left(1 - \frac{d-2}{d}\frac{k-1}{t}\right), \sigma_l^p \sim \frac{k-1}{d}\frac{1}{t}. \tag{162}$$

The dynamics indeed converges to the desired outcome.

## H.2. Two-step CoT and the step-wise learning behavior

The two-step reward is

$$\mathbb{E}[r(\mathbf{x}, \mathbf{y})] = \mathbb{E}[r_{t=1}(\mathbf{x}, \mathbf{y})] + \mathbb{E}[r_{t=2}(\mathbf{x}, \mathbf{y})]. \tag{163}$$

For ease of notation we denote $l^{(1)} = l_1$ and $l^{(2)} = l_2$, and $y$. The first term is just what we computed in the one-step case:

$$\mathbb{E}[R_{t=1}(\mathbf{x}, \mathbf{y})] = \frac{4}{k-1}\sum_{l^{(1)}=1}^{k/2}\sigma_{d+l_1}^{i_1^{l_1}}\sigma_{d+l_1}^{i_2^{l_1}}. \tag{164}$$

We only need to compute $\mathbb{E}[R_{t=2}(\mathbf{x}, \mathbf{y})]$:

$$\mathbb{E}[r_{t=2}(x, y)] = \frac{1}{k-1}\mathbb{E}\left[\sum_{l_2=1}^{k/4}\tilde{y}_{l_2}^{(2)}y_{l_2}^{(2)}\right]$$

$$= \frac{2}{k-1}\sum_{l_2}\mathbb{E}\left[x_{i_1^{l_2}}x_{i_2^{l_2}}x_{i_3^{l_2}}x_{i_4^{l_2}}\left(\sum_{l_1=1}^{k/2}y_{l_1}^{(1)}\sigma_{d+k/2+l_2}^{l_1}\right)^2\right] \tag{165}$$

$$= \frac{2}{k-1}\sum_{l_2}\sum_{l_1, \tilde{l}_1}\sigma_{l_2}^{l_1}\sigma_{l_2}^{\tilde{l}_1}\mathbb{E}[x_{i_1^{l_2}}x_{i_2^{l_2}}x_{i_3^{l_2}}x_{i_4^{l_2}}y_{l_1}^{(1)}y_{\tilde{l}_1}^{(1)}]$$

where $x_{i_1^{l_2}}, x_{i_2^{l_2}}, x_{i_3^{l_2}}, x_{i_4^{l_2}}$ are the four bits at $t = 0$ that $y_{l_2}$ truly depends on (so $i_1^{l_2}, i_2^{l_2}, i_3^{l_2}, i_4^{l_2}$ are all different), and $\hat{y}_{\tilde{l}_1}$ are the sampled output of first step of the CoT. Note since $i_1, i_2, i_3, i_4$ are all different, only the quartic term in $x$ of the probablity of $y_{l_1}^{(1)}y_{\tilde{l}_1}^{(1)}$ can contribute a nonzero answer, that is, it must be that $l_1 \neq \tilde{l}_1$. So we can further write

$$\mathbb{E}[r_{t=2}(x, y)] = \frac{8}{k-1}\sum_{l_2=1}^{k/4}\sum_{l_1 \neq \tilde{l}_1}^{k/2}\sigma_{d+k/2+l_2}^{l_1}\sigma_{d+k/2+l_2}^{\tilde{l}_1}\mathbb{E}\left[x_{i_1^{l_2}}x_{i_2^{l_2}}x_{i_3^{l_2}}x_{i_4^{l_2}}\left(\sum_i^d x_i\sigma_{d+l_1}^i\right)^2\left(\sum_{\tilde{i}}^d x_{\tilde{i}}\sigma_{d+\tilde{l}_1}^{\tilde{i}}\right)^2\right]. \tag{166}$$

To compute this we can again use the pairing identity Eq. (156), and obtain

$$\mathbb{E}[r_{t=1}] + \mathbb{E}[r_{t=2}] = \frac{4}{k-1}\sum_{l_1=1}^{k/2}\sigma_{d+l_1}^{i_1^{l_1}}\sigma_{d+l_1}^{i_2^{l_1}} + \frac{8}{k-1}\sum_{l_2=1}^{k/4}\sum_{l_1 \neq \tilde{l}_1}^{k/2}\sigma_{d+k/2+l_2}^{l_1}\sigma_{d+k/2+l_2}^{\tilde{l}_1}\sigma_{d+l_1}^{\{i_1^{l_2}}\sigma_{d+l_1}^{i_2^{l_2}}\sigma_{d+\tilde{l}_1}^{i_3^{l_2}}\sigma_{d+\tilde{l}_1}^{i_4^{l_2}\}} \tag{167}$$

where $\sigma_{l_1}^{\{i_1}\sigma_{l_1}^{i_2}\sigma_{\tilde{l}_1}^{i_3}\sigma_{\tilde{l}_1}^{i_4\}}$ denotes the sum over all 24 permutations of the four superscripts.

We can see how step-wise learning emerges when $d \gg 1$: at random intitialization, each of $\sigma_{d+l_1}^i$ roughly has a size of $1/d$ and each of $\sigma_{d+k/2+l_2}^{l_1}$ has a size of $1/k$. Therefore the first term is of order $1/d^2$ and the second term is of order $1/d^4$, so there is a scale separation for large $d$. Because of this, at earlier time the dynamics is dominated by the first term, and only at

late time the second term becomes important. That is, the system will first learn the first layer of CoT until the first layer is very close to the final correct weights:

$$\sigma_{d+l_1}^i \approx \begin{cases} 1/2, & \text{if } i = i_1^{l_1}, i_2^{l_1} \\ 0, & \text{otherwise.} \end{cases} \tag{168}$$

At this point the gradient contribution of the first term is very close to zero, and the second term becomes non-negligible, and essentially becomes

$$\mathbb{E}[r_{t=2}] \propto \sum_{l_2=1}^{k/4} \sum_{l_1 \neq \tilde{l}_1}^{k/2} \sigma_{d+k/2+l_2}^{l_1} \sigma_{l_2}^{d+k/2+\tilde{l}_1} = \text{constant} \times \sum_{l_2=1}^{k/4} \sigma_{d+k/2+l_2}^{l_1} \sigma_{d+k/2+l_2}^{\tilde{l}_1}. \tag{169}$$

Note that the sum over $l_1, \tilde{l}_1$ disappears because the Eq. (168) determines $l_1, \tilde{l}_1$ as a function of $l_2$ (for nonzero $\sigma_{l_2}^{l_1}, \sigma_{l_2}^{\tilde{l}_1}$). This becomes essentially the same form of reward as $\mathbb{E}[r_{t=1}]$, so at late time the dynamics basically repeats the first step and the desired weights will be learned.

This scale separation behavior continues for more CoT steps, as each new step brings out a reward term with higher powers in $\sigma_{d+l_1}^i$. Therefore the system learns step by step.

## I. Extension to Serial CoT

In this section, we analyze the masked serial CoT extension (Fig. 3) beyond the complete binary-tree decomposition of the sparse Boolean function. For simplicity, we focus on $k$-PARITY. $k$-AND and OR should have similar formulations. Our goal is to show that, under uniform initialization and sign-gradient updates, the two true parent coordinates receive positive gradient updates, while all non-parent coordinates receive negative updates, which leads to the learnability results in Thm. 3.1 and 3.3. Overall, our theoretical conclusions will be

- For both RL and SFT with teacher forcing, the transformer can provably learn $k$-PARITY in one sign gradient update, exhibiting simultaneous learning as in Thm. 3.1;

- For SFT without teacher forcing, the direct learning fails due to noisy later-CoT-token update; however, with a filtering that suppresses the noisy later-CoT-token update when learning earlier tokens, the transformer exhibits step-wise learning that learns one CoT token per sign-gradient update.

**Setup.** Given $\mathbf{x} = (x_1, \ldots, x_d)$, where $x_1, \ldots, x_d \overset{\text{i.i.d.}}{\sim} \text{uniform}\{\pm 1\}$, and the relevant bits set $B = \{i_1, \ldots, i_k\} \subseteq [d]$ as in Sec. 2.1, let the $k$-sparse Boolean function be solved by following a CoT-style reasoning chain (Fig. 8a)

$$\text{Serial CoT decomposition: } y_1 = \phi_2(x_{i_1}, x_{i_2}), \ y_j = \phi_2(y_{j-1}, x_{i_{j+1}}), \ j = 2, \ldots, k-1, \tag{170}$$

where $\mathbf{x}$ is the input, $\mathbf{y}$ is a serial CoT, and we do not require $k = 2^T$ for some integer $T$. For the transformer, we still use the transformer defined in Sec. 2.2 but only replace the tree mask there with a more flexible and weaker serial mask (Fig. 8b) that, when generating $y_j$, allows the model to attend to all input bits of $\mathbf{x}$ but only to the most recent CoT step $y_{j-1}$, i.e., at step $t + 1$, the model predicts $y_{j+1}$ from the visible set $\{x_1, \ldots, x_d, y_j\}$. This matches the serial recursion and no longer strictly enforces the exact binary tree in Fig. 2a. Formally, given the activation function $\psi(\cdot)$ and the input sequence $\mathbf{x}$ along with the CoT reasoning chain $\mathbf{y}$ such that $\mathbf{z} = (\mathbf{x}, \mathbf{y})$, the transformer has the following output:

$$[f(\mathbf{z}; \boldsymbol{W})]_{d+j} = \psi(\xi_j), \quad \xi_j := y_{j-1}\sigma_{d+j}^{d+j-1} + \sum_{i=1}^d x_i \sigma_{d+j}^i \tag{171}$$

For convenience, define the sum of all bits of $\mathbf{x}$

$$S := \sum_{m=1}^d x_m, \qquad M := d + 1, \tag{172}$$

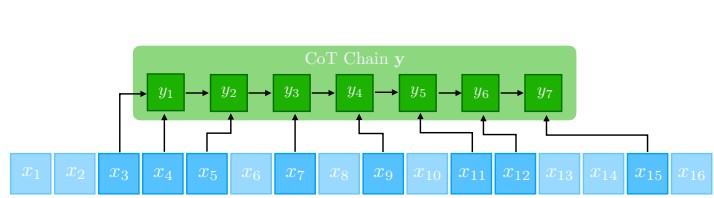 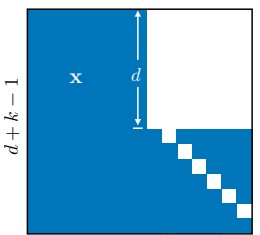

*(a)* Serial CoT-style decomposition of learning a $k$-sparse Boolean function $\Phi_k(\mathbf{x})$ with a random set $B \subseteq [d]$ (shaded boxes in $\mathbf{x}$) into solving sub-tasks by following a reasoning chain $\mathbf{y}$ (green boxes).

*(b)* "Pretrained" mask for the transformer in the serial CoT setting. Each CoT token can attend to all input bits and its last CoT step.

*Figure 8.* Learning $k$-sparse Boolean functions in a serial CoT manner.

then at uniform initialization, every visible source has attention weight $\sigma_j^i = 1/M$. Therefore the the transformer output Eq. (171) will be

$$\xi_j = \frac{S + y_j}{M}. \tag{173}$$

Below we let $\phi_2(a, b) = ab$ to specifically consider $k$-PARITY, so $\psi(a) = a^2$.

### I.1. RL Fine-tuning

We first introduce the problem setup for RL fine-tuning. Then, similar to Sec. 3.1, we still view each input $\mathbf{z} = (\mathbf{x}, \mathbf{y})$ as a state but now each output of the transformer is a distribution of the action $y_j$ (a reasoning step of the CoT chain) under the state $\mathbf{z}$, i.e., $p_{\mathbf{W}}(y_j = 1|\mathbf{z}) = [f(\mathbf{z}; \mathbf{W})]_{d+j}$. Thus, now the distribution over one entire CoT reasoning chain $\mathbf{y}$ conditioned on the initial state $\mathbf{x}$ is

$$p_{\mathbf{W}}(\mathbf{y}|\mathbf{x}) = \prod_{j=1}^{k-1} p_{\mathbf{W}}(y_j|\mathbf{x}, y_{j-1}). \tag{174}$$

Then, different from Sec. 3.1, starting from $\mathbf{z} = (\mathbf{x}, \mathbf{y})$ with $\mathbf{y}$ initialized as 0, the transformer solves $\Phi_k(\mathbf{x})$ by generating a reasoning sequence where each sampled action $\hat{y}_j$ updates the state from $\hat{\mathbf{z}} = (\mathbf{x}, \hat{\mathbf{y}}_{:j-1}, \mathbf{y}^{(t:)})$ to $(\mathbf{x}, \hat{\mathbf{y}}_{:j-1}, \hat{y}_j, \hat{\mathbf{y}}_{:j+1})$ in a CoT manner. And now RL still aims to maximize the expected reward but with a new $t$-th step process reward:

$$\begin{aligned} \max_{\mathbf{W}} \mathcal{R}(\mathbf{W}) &:= \max_{\mathbf{W}} \mathbb{E}_{\mathbf{x}} \left[ R(\mathbf{x}; \mathbf{W}) \right], \\ R(\mathbf{x}; \mathbf{W}) &:= \mathbb{E}_{\mathbf{y} \sim p_{\mathbf{W}}(\cdot|\mathbf{x})} [r(\mathbf{x}, \mathbf{y})], \\ r(\mathbf{x}, \mathbf{y}) &:= \sum_{t=1}^{k-1} r_t = \sum_{t=1}^{k-1} \frac{1}{k-1} y_t \bar{y}_t, \end{aligned} \tag{175}$$

where $r : \{-1, +1\}^{d+k-1} \to [-1, 1]$ is the reward function, $\bar{y}_t$ is the correct label of $y_t$ given $\mathbf{y}_{:t-1}$. We still consider RL optimized by policy gradient to approximate the optimal parameters. We summarize the learnability result for this setup as follows.

**Proposition I.1** (Learnability via RL for serial CoT). *Consider the serial CoT decomposition of $k$-PARITY and RL with process reward. Let $W(0) = \mathbf{1}$ be the uniform initialization, let the parity activation function be $\psi(z) = z^2$, and let $\mathbf{W}^\star$ be the optimal serial attention parameters. Then the separation condition for the gradient is satisfied in the sense that, for the current token $y_j$, the gradient always has positive value at relevant tokens ($x_{i_j}$ and $y_{j-1}$) while having negative values at irrelevant tokens (all other $x_i$'s for $i \neq i_j$). As a result, one sign-policy-gradient update*

$$\mathbf{W}(1) = \mathbf{W}(0) + \eta \, \text{sign}(\nabla_{\mathbf{W}} \mathcal{R}(\mathbf{W}(0)))$$

*with $\eta = \Omega(\log(d/\epsilon))$ satisfies*

$$\|\text{softmax}(\mathbf{W}(1)) - \text{softmax}(\mathbf{W}^\star)\|_1 \leq \epsilon.$$

*Consequently, RL with process reward learns all serial CoT steps simultaneously in one sign-gradient update.*

*Proof.* We start with the learning dynamics. In this setup, according to Eq. (173), the model samples $y_{j+1} = 1$ with the probability (recall that $S = \sum_i x_i$)

$$p_{\boldsymbol{W}}(y_{j+1} = 1 \mid \mathbf{x}, y_j) = \psi(\xi_j) = \left(\frac{S + y_j}{M}\right)^2. \tag{176}$$

Since $y_{j+1} \in \{\pm 1\}$, we can derive the expectation of $y_{j+1}$ given $(\mathbf{x}, y_j)$ under the model $p_{\boldsymbol{W}}(\cdot \mid \mathbf{x}, y_j)$:

$$\mathbb{E}[y_{j+1} \mid \mathbf{x}, y_j] = 2\left(\frac{S + y_j}{M}\right)^2 - 1,$$

which only depends on $\mathbf{x}$ via the sum $S$, and thus we simplify the notation as $\mathbb{E}[y_{j+1} \mid S, y_j]$ such that

$$
\begin{aligned}
\mathbb{E}[y_{j+1} \mid S = s, y_j] &= 2\left(\frac{s + y_j}{M}\right)^2 - 1 \\
&= \left(\frac{2(s^2 + 1)}{M^2} - 1\right) + \frac{4s}{M^2} y_{j-1} \\
&= A(s) + B(s) y_{j-1},
\end{aligned}
\tag{177}
$$

where we define

$$A(s) := \frac{2(s^2 + 1)}{M^2} - 1, \qquad B(s) := \frac{4s}{M^2}. \tag{178}$$

As a result, we can calculate the expectation of $y_j$ given $\mathbf{x}$ with $S = s$ using the following recursive relation:

$$m_j(s) := \mathbb{E}[y_j \mid S = s] = \mathbb{E}[A(s) + B(s) y_{j-1} \mid S = s] = A(s) + B(s) m_{j-1}(s). \tag{179}$$

Now consider the prediction of $y_{j+1}$. For ease of notation, let the current relevant input parent be

$$x := x_{i_j}.$$

The local parity target is then $\bar{y}_{j+1} = y_j x$. At uniform initialization, the centered softmax-column gradient for a visible token $z$ has the form (computed from Eq. (176))

$$\textbf{Gradient direction: } \quad G_z \propto \mathbb{E}[2\xi_j \, \bar{y}_{j+1} (z - \xi_j)]. \tag{180}$$

Therefore

$$G_x \propto \mathbb{E}[2\xi_j \, y_j x \, (x - \xi_j)], \tag{181}$$
$$G_y \propto \mathbb{E}[2\xi_j \, y_j x \, (y_j - \xi_j)], \tag{182}$$

where $G_y$ denotes the gradient on the previous CoT parent $y_j$ (we are focusing on $y_{j+1}$ so we ignore $j$). To prove the separation condition of the gradient, we need to show that

$$G_x > 0, \quad G_y > 0, \quad \text{and } g_n < 0 \text{ for all other tokens} .$$

We prove these three conditions in the following.

**Positivity of the previous CoT token gradient:** $g_y > 0$. We first analyze $G_y$. Conditional on $S = s$, by exchangeability of the input coordinates,

$$\mathbb{E}[x \mid S = s] = \frac{s}{d}. \tag{183}$$

Using the definition $\xi_j = \frac{s + y_j}{M}$, and averaging over $y_j$ conditioning on $S = s$, one obtains (by direct expansion of Eq. (182))

$$G_y = \frac{C}{dM^2} \mathbb{E}\left[(d - 1)S^2 + S(d - S^2) m_j(S)\right],$$

for some constant $C > 0$ independent of the sign analysis. Fix $s$ and $j$, we abbreviate the notation as

$$m := m_j(s) \in [-1, 1], \tag{184}$$

$$F_y(s, m) := (d-1)s^2 + s(d - s^2)m. \tag{185}$$

To claim $G_y > 0$, we need to show

$$F_y(s, m) \geq 0 \quad \text{for all } s \in [-d, d], \ m \in [-1, 1].$$

Specifically, let $a := |s|$. Since $|m| \leq 1$,

$$F_y(s, m) \geq (d-1)a^2 - a|d - a^2|.$$

If $a^2 \leq d$, then

$$(d-1)a^2 - a(d - a^2) = a(a-1)(a+d) \geq 0.$$

If $a^2 \geq d$, then

$$(d-1)a^2 - a(a^2 - d) = a(d-a)(a+1) \geq 0.$$

Therefore

$$F_y(s, m) \geq 0$$

for every admissible $s, m$. Hence

$$G_y \geq 0.$$

Moreover, for $d \geq 2$, the inequality is strict on a set of positive probability. Thus

$$G_y > 0. \tag{186}$$

**Positivity of $G_x$.** We now analyze $G_x$. Starting from Eq. (181) and again conditioning on $S = s$, using Eq. (183), we obtain

$$G_x = \frac{C}{dM^2} \mathbb{E}\left[d(d+1) - 2S^2 + S(d^2 + d - 1 - S^2)m_j(S)\right], \tag{187}$$

for some constant $C > 0$. Thus the sign of $G_x$ is the sign of

$$\mathbb{E}\left[d(d+1) - 2S^2 + S(d^2 + d - 1 - S^2)m_j(S)\right]. \tag{188}$$

We prove that this expectation is strictly positive for every $j \geq 0$, which guarantees $G_x > 0$. For $a > 0$, define

$$\alpha(a) := d(d+1) - 2a^2, \tag{189}$$

$$\beta(a) := a(d^2 + d - 1 - a^2). \tag{190}$$

The contribution to the expectation in Eq. (188) at $S = a$ are

$$\begin{cases} \alpha(a) + \beta(a)m_j(a), & S = a \\ \alpha(a) - \beta(a)m_j(-a), & S = -a. \end{cases} \tag{191}$$

We lower-bound their sum uniformly over $j$. From the recursion Eq. (177) and the fact that $m_{j-1}(s) \in [-1, 1]$, for $a > 0$,

$$\begin{cases} m_j(a) \geq A(a) - B(a) = \frac{2(a-1)^2}{M^2} - 1, \\ m_j(-a) \leq A(a) + B(a) = \frac{2(a+1)^2}{M^2} - 1. \end{cases} \tag{192}$$

Therefore,

$$[\alpha(a) + \beta(a)m_j(a)] + [\alpha(a) - \beta(a)m_j(-a)] \geq 2\alpha(a) + \beta(a)\left(\frac{2(a-1)^2}{M^2} - 1\right) - \beta(a)\left(\frac{2(a+1)^2}{M^2} - 1\right)$$

$$= 2\alpha(a) + \frac{2\beta(a)}{M^2}\left((a-1)^2 - (a+1)^2\right) \tag{193}$$

$$= \frac{2}{M^2}\left[4a^4 - (6d^2 + 8d - 2)a^2 + d^4 + 3d^3 + 3d^2 + d\right].$$

This bound is uniform in $j$. Now average over the symmetric law of $S$. Since $S$ and $-S$ have the same probability, Eq. (193) implies

$$\mathbb{E}\left[d(d+1) - 2S^2 + S(d^2 + d - 1 - S^2)m_r(S)\right] \geq \frac{1}{M^2}\mathbb{E}\left[4S^4 - (6d^2 + 8d - 2)S^2 + d^4 + 3d^3 + 3d^2 + d\right]. \quad (194)$$

Because $S = \sum_{m=1}^{d} x_m$, with i.i.d. Rademacher $x_m$, we have

$$\mathbb{E}[S^2] = d, \qquad \mathbb{E}[S^4] = 3d^2 - 2d. \quad (195)$$

Substituting Eq. (195) into (194), we obtain

$$\begin{aligned}
&\mathbb{E}\left[d(d+1) - 2S^2 + S(d^2 + d - 1 - S^2)m_j(S)\right] \\
&\geq \frac{1}{M^2}\left[4(3d^2 - 2d) - (6d^2 + 8d - 2)d + d^4 + 3d^3 + 3d^2 + d\right] \\
&= \frac{1}{M^2}\left[d^4 - 3d^3 + 7d^2 - 5d\right] \\
&= \frac{d(d-1)(d^2 - 2d + 5)}{M^2}.
\end{aligned}$$

Since $d \geq 2$,

$$d(d-1)(d^2 - 2d + 5) > 0.$$

Therefore the expectation in Eq. (188) is strictly positive and therefore

$$G_x > 0 \qquad \text{for every } j \geq 0. \quad (196)$$

**Negativity of irrelevant tokens gradients**   Let $G_n$ denote the gradient on a non-parent raw input coordinate $x_n$, $n \neq i_j$. By exchangeability of the non-parent coordinates, all such gradients are equal. Moreover, the softmax column gradient sums to zero as in Eq. (29):

$$G_x + G_y + \sum_{n \neq i_j} G_n = 0.$$

There are $d - 1$ non-parent raw-input coordinates, so

$$G_x + G_y + (d-1)G_n = 0.$$

By Eq. (186) and Eq. (196),

$$G_x > 0, \qquad G_y > 0.$$

Therefore

$$(d-1)G_n = -(G_x + G_y) < 0 \implies G_n < 0. \quad (197)$$

**Conclusion**   For the masked serial parity model at uniform initialization, for every serial step $j \geq 0$,

$$G_x > 0, \qquad G_y > 0, \qquad G_n < 0. \quad (198)$$

Thus the two true parent coordinates receive positive sign-gradient updates, while every non-parent raw-input coordinate receives a negative sign-gradient update. As a result, one step of sign gradient update enables the transformer to learn all the CoT steps simultaneously by directly following the proof of Thm. 3.1 (**Step (ii)** of App. E.3). $\qquad\square$

## I.2. SFT

In this section, we analyze SFT without teacher forcing, and we stress that the results for SFT with teacher forcing can be easily generalized. We show that unfiltered SFT suffers from noisy gradients on later CoT tokens, whereas a filtered SFT procedure learns the serial CoT step by step. More precisely, after the previous CoT token has been learned, the filtered SFT gradient for the next token has the correct sign pattern: the two true parent coordinates receive positive sign-gradient updates, and all non-parent coordinates receive negative sign-gradient updates.

Consider the serial CoT decomposition of $k$-parity. Define the ground truth label of the CoT chain given $\mathbf{x}$ for $s = 1, \ldots, k-1$

$$\tilde{y}_s = \prod_{r=1}^{s+1} x_{i_r}.$$

For the "pretrained" transformer Eq. (171), when predicting $y_1$, the visible set is $\{x_1, \ldots, x_d\}$, whereas when predicting $y_s$ for $s \geq 2$, the visible set is

$$\{x_1, \ldots, x_d, \hat{y}_{s-1}\}.$$

Similar to Sec. 3.2, for a generated token $y_s$, write the model score as

$$\hat{q}_s = 2[f(\hat{z}; W)]_{d+s} - 1, \quad \hat{y}_s = \text{sign}(\hat{q}_s).$$

Using the hinge loss $\ell_s(\boldsymbol{W}) = \max\{0, 1 - \hat{q}_s \tilde{y}_s\}$, we write the token-level SFT objective as

$$L_s(\boldsymbol{W}) = \mathbb{E}_x[\ell_s(\boldsymbol{W})],$$

which gives us the objective for SFT without teacher forcing:

$$L(\boldsymbol{W}) = \sum_{s=1}^{k-1} L_s(\boldsymbol{W}),$$

where each $\hat{q}_s$ is evaluated on the model-generated prefix. In contrast, filtered SFT uses a phase-wise objective: at phase $s$, we optimize only $L_s(\boldsymbol{W})$ and suppress all later-token losses

$$L_{s+1}(\boldsymbol{W}), \ldots, L_{k-1}(\boldsymbol{W}).$$

We analyze sign-gradient descent, and we will study three cases:

1. SFT without teacher forcing and without filtering, where the learning is hard.

2. SFT without teacher forcing but with filtering that suppresses noisy later token updates, where the CoT is learned step by step.

3. SFT with teacher forcing, where the CoT is learned simultaneously.

### I.2.1. SFT WITHOUT TEACHER FORCING HAS NOISY LATER-TOKEN UPDATES

For $s \geq 2$, the model predicts $y_s$ from the generated previous token $\hat{y}_{s-1}$. However, the supervised target is

$$\begin{aligned} \tilde{y}_s &= \tilde{y}_{s-1} x_{i_{s+1}} \\ &= \Delta_{s-1} \hat{y}_{s-1} x_{i_{s+1}}, \end{aligned} \tag{199}$$

where we define the prefix-correctness sign

$$\Delta_{s-1} := \hat{y}_{s-1} \tilde{y}_{s-1} \in \{\pm 1\}.$$

Therefore, relative to the clean local target

$$\hat{y}_{s-1} x_{i_{s+1}}$$

given the generated $\hat{y}_{s-1}$, the SFT label is multiplied by the noise sign $\Delta_{s-1}$. Consequently, the later-token SFT update has the form

$$\text{SFT direction for token } s = \Delta_{s-1} \cdot \text{clean local direction.}$$

If $\hat{y}_{s-1}$ has not yet been learned, then $\Delta_{s-1}$ can be weakly correlated with the clean direction or even point in the wrong direction. Thus SFT without teacher forcing and without filtering tries to train later CoT tokens before their conditioning prefixes are reliable, producing noisy later-token gradients. Therefore, the learning is hard.

### I.2.2. SFT WITHOUT TEACHER FORCING BUT WITH THE CLEAN LOCAL SIGNAL UNDER FILTERING

Filtered SFT avoids the noisy later token update issue by suppressing token $s$ until token $s - 1$ has been learned. At phase $s$, suppose inductively that the previous token has already been learned in the sense that

$$\hat{y}_{s-1} = \tilde{y}_{s-1} \implies \tilde{y}_s = \tilde{y}_{s-1} x_{i_{s+1}} = \hat{y}_{s-1} x_{i_{s+1}}.$$

Thus, at phase $s$, the ground truth label $\tilde{y}_s$ agrees exactly with the clean local target $\hat{y}_{s-1} x_{i_{s+1}}$. To prove the learnability, we need to show that the $s$-step SFT gradient has positive value at these two positions $\hat{y}_{s-1}$ and $x_{i_{s+1}}$ (i.e., the relevant bits) and has negative value at all other positions. Then applying the same analysis as in Thm. 3.3, we will be able to confirm that CoT chain is learned step by step. Define the descent direction

$$g_{p,s} := -\frac{\partial L_s(\boldsymbol{W})}{\partial W_{p,s}} \tag{200}$$

Thus $g_{p,s} > 0$ means that sign-gradient descent increases the attention weight (and thus the logit) $W_{p,s}$. We summarize our conclusion in the following theorem. The proof is deferred to App. I.2.4.

**Proposition I.2** (SFT without teacher forcing but with filtering learns $k$-PARITY step by step)**.** *Consider the serial CoT parity decomposition Eq. (170). At phase $s$, suppose that all later-token losses are suppressed by an appropriate filtering. If the previous CoT token has been learned $\hat{y}_{s-1} = \tilde{y}_{s-1}$ for $s \geq 2$, then the sign-gradient direction satisfies the following.*

- *For $s = 1$, $g_{i_1,1} > 0$, $g_{i_2,1} > 0$, $g_{p,1} < 0$   for all $p \notin \{i_1, i_2\}$.*

- *For $s \geq 2$, $g_{i_{s+1},s} > 0$, $g_{d+s-1,s} > 0$, $g_{p,s} < 0$   for every irrelevant input token $x_p$. Moreover, $g_{i_{s+1},s} = g_{d+s-1,s}$.*

*Consequently, one sign-gradient update increases exactly the two relevant positions and decreases all the other irrelevant positions, and thus the CoT chain is learned step by step.*

### I.2.3. SFT WITH TEACHER FORCING

The condition of learning the $s$-th token in Prop. I.2 is that its previous token $\hat{y}_{s-1}$ is correctly learned, which is automatically satisfied by SFT with teacher forcing across the whole CoT chain. This is because SFT with teacher forcing only learns $y_s$ based on the correct previous token $\tilde{y}_{s-1}$. Therefore, all steps are learned simultaneously.

### I.2.4. PROOF OF THEOREM I.2

*Poof:* The proof consists of two parts: the learning of the first CoT token $y_1$ and that of later CoT tokens with $s \geq 2$, which are studied as follows.

**Learning of the first CoT token $y_1$.**   We first study how the first CoT token $\tilde{y}_1 = x_{i_1} x_{i_2}$ is learned, where the visible set is $\{x_1, \ldots, x_d\}$. Recall that $S := \sum_{m=1}^{d} x_m$. At uniform initialization,

$$\xi_1 = \frac{S}{d}.$$

For a visible bit $x_p$, the SFT descent direction is, up to a positive constant,

$$g_{p,1} \propto \mathbb{E}\left[2\xi_1 \tilde{y}_1 (x_p - \xi_1)\right] = \mathbb{E}\left[2\xi_1 x_{i_1} x_{i_2} (x_p - \xi_1)\right]. \tag{201}$$

First consider $p = i_1$:

$$g_{i_1,1} \propto \mathbb{E}[\xi_1 x_{i_2}] - \mathbb{E}[\xi_1^2 x_{i_1} x_{i_2}]$$
$$= \frac{1}{d} - \frac{\mathbb{E}[S^2 x_{i_1} x_{i_2}]}{d^2}. \tag{202}$$

The only terms in $S^2$ that survive multiplication by $x_{i_1} x_{i_2}$ and expectation are the two cross terms $x_{i_1} x_{i_2}$ and $x_{i_2} x_{i_1}$. Hence $\mathbb{E}[S^2 x_{i_1} x_{i_2}] = 2$, and therefore

$$g_{i_1,1} \propto \frac{1}{d} - \frac{2}{d^2} = \frac{d-2}{d^2} > 0 \text{ for } d > 2. \tag{203}$$

By symmetry, we can easily conclude that

$$g_{i_2,1} > 0.$$

Now take $p \notin \{a, b\}$. Then, by parity orthogonality,

$$\mathbb{E}[\xi_1 x_{i_1} x_{i_2} x_p] = 0 \implies g_{p,1} \propto -\frac{2}{d^2} < 0.$$

Thus,

$$g_{i_1,1} > 0, \quad g_{i_2,1} > 0, \quad g_{p,1} < 0 \quad \text{for all } p \notin \{i_1, i_2\}. \tag{204}$$

Therefore one sign-gradient step increases the two relevant logits and decreases every irrelevant logit for the first serial CoT token.

**Learning of the later CoT tokens $y_s$ for $s \geq 2$.** Assume inductively that the $(s-1)$-th step CoT token is perfectly learned in the sense that

$$\hat{y}_{s-1} = \tilde{y}_{s-1} = \prod_{r=1}^{s} x_{i_r}.$$

The target token for $y_s$ is thus

$$\tilde{y}_s = \hat{y}_{s-1} x_{i_{s+1}} = \tilde{y}_{s-1} x_{i_{s+1}},$$

and the visible set is $\{x_1, \ldots, x_d, \tilde{y}_{s-1}\}$. In the following we use $\tilde{y}_{s-1}$ and $\hat{y}_{s-1}$ interchangeably. Recall Eq. (172). At uniform initialization,

$$\xi_s = \frac{S + \tilde{y}_{s-1}}{M}.$$

For a visible $z$, again, according to Eq. (200), the SFT descent direction is, up to a positive constant,

$$G_z \propto \mathbb{E}\left[2\xi_s \tilde{y}_{s-1} x_{i_{s+1}} (z - \xi_s)\right], \tag{205}$$

where we directly use $G_z$ to denote the gradient of the position of $z$ for simplicity (i.e., if $z$ is the $j$-th token of $(\mathbf{x}, \mathbf{y})$ when generating the $s$-th token, then $G_z$ represents $g_{j,s}$). The second term is not related to any specific $z$:

$$
\begin{aligned}
\mathbb{E}[\xi_s^2 \tilde{y}_{s-1} x_{i_{s+1}}] &= \frac{1}{M^2} \mathbb{E}[(S + \tilde{y}_{s-1})^2 \tilde{y}_{s-1} x_{i_{s+1}}] \\
&= \frac{1}{M^2} \left(\mathbb{E}[S^2 \tilde{y}_{s-1} x_{i_{s+1}}] + 2\mathbb{E}[S x_{i_{s+1}}] + \mathbb{E}[\tilde{y}_{s-1} x_{i_{s+1}}]\right).
\end{aligned}
\tag{206}
$$

Noting that $\mathbb{E}[S x_{i_{s+1}}] = 1$, $\mathbb{E}[\tilde{y}_{s-1} x_{i_{s+1}}] = 0$, and

$$\mathbb{E}[S^2 \tilde{y}_{s-1} x_{i_{s+1}}] = 0$$

as $S^2$ contains only monomials of degree 0 and 2, whereas $\tilde{y}_{s-1} x_{i_{s+1}}$ has degree at least 3, and thus no term in $S^2 \tilde{y}_{s-1} x_{i_{s+1}}$ reduces to the constant monomial. Therefore,

$$\mathbb{E}[\xi_s^2 \tilde{y}_{s-1} x_{i_{s+1}}] = \frac{2}{M^2}.$$

In fact, the first term

$$\mathbb{E}\left[2\xi_s \tilde{y}_{s-1} x_{i_{s+1}} z\right] \tag{207}$$

is closely related to the critical gradient component in Sec. 3.1 and 3.2, so we in fact only need to examine whether it satisfies the separation condition to show the step by step learning dynamics.

In the following, we discuss three cases: $z$ is the relevant input token $x_{i_{s+1}}$; $z$ is the previous token $y_{s-1}$; and $z$ is any irrelevant token from $\mathbf{x}$.

1. **Relevant input token $x_{i_s+1}$.** For the relevant input token $z = x_{i_s+1}$,

$$G_{x_{i_s+1}} \propto \mathbb{E}\left[2\xi_s\tilde{y}_{s-1}x_{i_s+1}(x - \xi_s)\right]$$
$$\propto \mathbb{E}[\xi_s\tilde{y}_{s-1}] - \mathbb{E}[\xi_s^2\tilde{y}_{s-1}x_{i_s+1}]. \tag{208}$$

For the first term,

$$\mathbb{E}[\xi_s\tilde{y}_{s-1}] = \frac{1}{M}\mathbb{E}[(S + \tilde{y}_{s-1})\tilde{y}_{s-1}] = \frac{1}{M}$$

because $\mathbb{E}[S\tilde{y}_{s-1}] = 0$. Combined with Eq. (208), we obtain

$$G_{x_{i_s+1}} \propto \frac{1}{M} - \frac{2}{M^2} = \frac{d-1}{(d+1)^2} > 0 \tag{209}$$

as desired.

2. **Previous CoT token $y_{s-1}$.** For the previous CoT token $z = \tilde{y}_{s-1}$,

$$G_{\tilde{y}_{s-1}} \propto \mathbb{E}\left[2\xi_s\tilde{y}_{s-1}x_{i_s+1}(\tilde{y}_{s-1} - \xi_s)\right]$$
$$\propto \mathbb{E}[\xi_s x_{i_s+1}] - \frac{2}{M^2} \tag{210}$$

up to a positive constant. Each term has already been calculated in last case. Therefore

$$G_{\tilde{y}_{s-1}} \propto \frac{1}{M} - \frac{2}{M^2} = \frac{d-1}{(d+1)^2} > 0. \tag{211}$$

More precisely, we in fact have

$$G_{x_{i_s+1}} = G_{\tilde{y}_{s-1}}.$$

3. **Irrelevant tokens from the input x.** According to the form of $G_{p,s}$ in Eq. (205), the zero-sum softmax gradient relation easily follows

$$\sum_{z \in \{x_1, \dots, x_d, \tilde{y}_{s-1}\}} G_z = 0. \tag{212}$$

Now let $x_n$ be an input token with $n \neq i_s+1$. By symmetry, all $G_{x_n}$ have the same value. As $G_{x_{i_s+1}} = G_{\tilde{y}_{s-1}}$, we must conclude that

$$G_{x_n} < 0 \quad \text{for all} \quad n \neq i_s+1.$$

**Conclusion.** Now we have the separation of the gradient (also the critical gradient component), following the analysis in Thm. 3.3, the stepwise learning dynamics of SFT without teacher forcing but with filtering under the serial CoT emerges. Specifically, the base case $s = 1$ follows directly from the Eq. (204), where the two relevant positions $i_1, i_2$ have positive directions, while every irrelevant input position has negative direction. Now let $s \geq 2$, and assume inductively that

$$\hat{y}_{s-1} = \tilde{y}_{s-1}.$$

Then

$$g_{i_s+1,s} = G_{d+s-1,s} > 0$$

and all other $g_{p,s} < 0$ for $p \notin \{i_s+1, d+s-1\}$. Thus, under sign-gradient descent from symmetric initialization, the two relevant logits remain tied while all non-parent logits are decreased. This completes the induction following the argument in App. F.1. $\qquad\square$

