# OpenReview forum: "Transformers with RL or SFT Provably Learn Sparse Boolean Functions, But Differently"
_ICML.cc/2026/Conference — ICML 2026 regular_

### Official Review · Reviewer_bCJ8 · 2026-03-10

**Soundness:** 2
**Presentation:** 3
**Significance:** 2
**Originality:** 3
**Overall Recommendation:** 3
**Confidence:** 4

**Summary:**

This paper studies the ability of a simple idealized transformer model to learn a specific subclass of sparse boolean functions, under either RL or SFT. Results indicate that both paradigms can succeed, but with differences.

**Compliance With Llm Reviewing Policy:**

Affirmed.

**Final Justification:**

The rebuttal addressed my questions. I don't have concerns about the soundness of the work.

Concerns about the significance due to restrictive setup remain, as indicated in the exchange with the authors.

**Key Questions For Authors:**

- 1. line 164 "can be expressed by a complete binary tree": this seems to depend on the function? In line 157, the relevant class is defined in terms of the existence of a recursive decomposition using a fixed 2-sparse function. Whether this decomposition can be done using a binary tree seems to depend on the function? This relates to Weakness 2 above.

- 2. More generally, how large is the class of sparse functions that can be expressed in terms of applying a fixed two-sparse Boolean function along a complete binary tree?

**Limitations:**

yes

**Strengths And Weaknesses:**

Strengths:

- 1. The paper studies a very timely question: whereas by now quite a lot of research has studied supervised learning of transformers, understanding how RL for reasoning changes the picture is a very timely question.

- 2. The paper provides rigorous theorems on a challenging question, even if in a highly simplified setup.


Weaknesses:


- 1. The title makes a very general claim that sounds exciting but is an overclaim: (i) the paper only studies a specific subclass of sparse functions; (ii) both RL and SFT are idealized in various simplifications, (iii) the Transformer model is highly simplified and in particular the tree structure and the binary combination function are both hard-coded into the model.

- 2. Considering k-sparse Boolean functions that can be decomposed into recursive application of a fixed 2-sparse Boolean function seems very restrictive. See also questions.

- 3. Eq (6) makes a key simplification in disregarding rewards from future steps.

- 4. The comparison between RL and SFT is limited by the fact that SFT uses hinge loss, RL uses a discrete reward. Both choices are presumably helpful in making the theoretical analysis tractable, but hinge loss is already an idealization not matching LLMs. How do we know this choice isn't a confounding factor in the comparison?

- 5. The binary tree structure is hardcoded into the formal transformer model (line 218). This seems like a big weakness as practical transformers learn without such hard-coding of input structure. Given that learning such structure might not be easy for transformers, this assumption risks missing the most difficult and thus interesting part of the learning model.

---

> ### Author Rebuttal · Authors · 2026-03-30
>
> We thank the reviewer for the constructive comments, and for appreciating that the paper studies a timely and challenging question in a rigorous way.
> 1. **Re to W1:** We agree that the current scope reads too general. In revision, we will revise Abstract and Introduction to state the scope more explicitly: we study a specific recursively decomposable subclass of k-sparse Boolean functions under a simplified one-layer transformer. In addition, we believe our paper makes a meaningful contribution by giving a timely, unified framework that compares RL and SFT on transformers.
>
>     For sub-point **(i)**, please see **Re to W2**. For **(ii)**, please see **Re to W4**. For **(iii)**, please see **Re to W5**.
> 2. **Re to W2:**  We agree that the function class is restrictive. We would also like to highlight that our goal is not to solve arbitrary Boolean functions at once, which remains as an interesting open question, but to **obtain a timely tractable analysis for RL and its comparison to SFT**. Compared to prior works that focused on k-PARITY and SFT, our paper already develops **a more unified framework** that covers both RL and SFT and a broader class of Boolean functions.
> 3. **Re to W3:** Eq. (6) is based on the intuition that rewards from future steps are not very informative for the current step, as the model can still achieve a high reward on the current step when its future steps are wrong.
>
>     To see whether this affects the main conclusion, during the rebuttal, we also ran the experiments in Appendix B (Fig.3a, page 13) but included rewards from future steps, and observed **the same qualitative learning dynamics**. In the revision, we will clarify it and add the corresponding experiments.
> 4. **Re to W4:** We thank the reviewer for this question. The qualitative SFT result in our paper is **not specific to hinge loss**, as the step-wise learning dynamics intuitively comes from the autoregressive nature rather than the exact form of the loss, i.e., later CoT steps only become reliably learnable once earlier steps are correct.
>
>     This intuition extends beyond hinge loss. For a loss function $\ell(\hat q,a)=h(a\hat q),$ where $h$ is nonincreasing such as logistic loss and exponential loss $h(z)=e^{-z}$, then $$\nabla_W\ell(\hat q,a)=h'a\nabla_W\hat q.$$ Thus, compared to hinge loss where $\nabla_W\ell=-a\nabla_W\hat q$ given $\hat qa\leq1$, the gradient keeps the same direction and is only rescaled by a factor $-h'(a\hat q)>0$, i.e., the loss changes the magnitude of the update while preserving its direction.
>
>     As a result, the same proof strategy still applies if we replace the original critical gradient component $\gamma$ by a scaled $\Gamma=-h'(a\hat q)\gamma$. In this way, the comparison between relevant and irrelevant nodes keeps the same sign, and the qualitative step-wise conclusion is unchanged. Thus, hinge loss indeed gives a cleaner result, but it is **not a confounding factor**.
> 5. **Re to W5:** We would like to highlight that our main contribution is the identification of a general framework that characterizes how RL and SFT learn over CoT-style recursive structure.
>
>     In particular, as also suggested by Reviewer **yzEa and 1fvS**, during the rebuttal we worked out that the same framework can also be extended to a more common **serial CoT** setting with a weaker structural prior where $$y_1=\phi_2(x_{i_1},x_{i_2}),\ y_{j}=\phi_2(y_{j-1},x_{i_{j+1}}),\ j\geq2,$$ with $x$ being the input and $y$ a serial CoT. Thus, the analysis is not restricted to complete binary tree. Due to space limitations, please also see the **first point of our response to Reviewer yzEa** for a more detailed discussion.
>
>     In addition, we agree that learning such structure from scratch is an important challenge for transformers. But we believe the current paper still addresses a central and nontrivial part of the problem: even when a structural prior is available, it remains far from obvious whether RL and SFT can learn, and how their learning dynamics differ. And our paper gives, to our knowledge, one of the first unified theoretical characterizations of the learning dynamics.
> ---
> 1. **Re to Q1:** Yes, this depends on the function. Not every $k$-sparse Boolean function admits a recursive decomposition via a fixed $\phi_2$, and our paper only studies the subclass that can. We will make this scope more explicit.
> 2. **Re to Q2:** Fixing $\phi_2$ induces a single Boolean function on relevant bits; the class then comes from varying $\phi_2$, and thus has up to 16 possible forms (as $\phi_2:\\{-1,+1\\}^2\to\\{\pm1\\}$ has two bits, so there are 4 possible inputs; for each possible input, the function can output $\pm1$, thus total $2^4=16$ forms).
>
>     Our goal here is to study learnability for this class, rather than to fully classify all induced functions. We thus focus on a general analysis and on more representative examples PARITY, AND, and OR, and leave a complete combinatorial characterization to future work.

---

> > ### Author Rebuttal · Reviewer_bCJ8 · 2026-04-03
> >
> > Thanks to the authors for their insightful and constructive rebuttal.
> >
> > I do believe that the strong simplifications of the model (W1) substantially limit the paper's contributions. The extension to serial CoT sketched in the rebuttal is a step in a good direction, but still restrictive because it still involves a hard-coded, apparently ad-hoc, mask. If the paper is accepted, the authors must make this explicit in abstract and introduction, as they promise in the rebuttal.
> >
> > I thus am reluctant to increase my score, but am not opposed to the paper being accepted, provided that these limitations are clearly advertised and overclaims about generality are avoided.
> >
> > Overall, if the paper is accepted, I strongly ask the authors to both (i) rephrase the paper where appropriate (e.g., to address W1, W2) and (ii) add appropriate discussion of each of these points to the paper (e.g., in a dedicated "FAQ" appendix section), for full transparency and benefit to the paper's readers who want to build their future research on it. I'm sure they'll appreciate those insights and nuances.

---

> > > ### Author Response · Authors · 2026-04-03
> > >
> > > We thank the reviewer for the follow-up and the constructive suggestions. In the revision, we will do exactly that: we will **revise the Abstract and Introduction** to clarify the scope more explicitly, avoid overclaiming, and **add a dedicated FAQ** section in the Appendix, as suggested.
> > >
> > > At the same time, we would like to emphasize that we view the paper as a substantive first theoretical step on a **timely and challenging problem**: it gives, to our knowledge, one of the first unified analyses of **how RL and SFT differ in their learning dynamics for transformers with CoT reasoning**, and it already goes beyond prior theory that focused essentially on parity and SFT only. Furthermore, the serial CoT extension we worked out during the rebuttal further broadens the scope beyond the current theory setup and assumptions.
> > >
> > > Therefore, while the paper makes assumptions that limit the scope, we believe it still makes a significant and nontrivial contribution by characterizing the learning dynamics that was not previously understood at this level.
> > >
> > > Regarding the structural prior, we agree that the hard-coded mask is a restrictive assumption, and that learning such structure from scratch is an important open problem. We will make that explicit. At the same time, we do not view the structural mask as merely ad hoc: it is a theoretically meaningful structural prior that can be understood as an abstraction of one important role of pretraining for studying recursive reasoning under the transformer model analyzed here.
> > >
> > > More generally, we appreciate the reviewer’s suggestion on transparency, and we will incorporate these revisions so that the final paper presents both its scope and its contribution as clearly as possible.

---

### Official Review · Reviewer_1fvS · 2026-03-11

**Soundness:** 3
**Presentation:** 3
**Significance:** 2
**Originality:** 2
**Overall Recommendation:** 4
**Confidence:** 2

**Summary:**

The paper investigates the theoretical learning dynamics of transformers fine-tuned with RL versus SFT to acquire CoT reasoning capabilities. The main theoretical contribution is establishing the provable learnability of both approaches under specific sufficient conditions, while revealing a fundamental dichotomy in their learning mechanisms: RL learns the entire CoT chain simultaneously due to global feedback mechanisms, whereas SFT learns the chain step-by-step sequentially.

**Compliance With Llm Reviewing Policy:**

Affirmed.

**Final Justification:**

I recommend the paper as borderline accept.

**Key Questions For Authors:**

1. The framework seems to tailor the feedforward activation function $\psi$ to the specific target Boolean function (e.g., using $z^2$ for PARITY and $\max(z,0)$ for AND). Does this imply that the target logic is partially baked into the architecture prior to training? It would be helpful to clarify if this specific architectural choice provides a strong inductive bias that fundamentally simplifies the task.
2. How would the theoretical separation conditions behave if the $k$-sparse Boolean function could not be perfectly decomposed into fixed 2-sparse functions, but instead had a more entangled or overlapping graph structure?

**Limitations:**

yes

**Strengths And Weaknesses:**

Strengths:
The framework improves upon prior theoretical formulations by sampling intermediate tokens, which better captures the autoregressive nature of Chain-of-Thought generation.
By successfully identifying a "critical gradient component", the paper rigorously explains why RL learns reasoning steps simultaneously while SFT naturally exhibits a step-by-step learning behavior.

Weaknesses:
1. The analysis is strictly limited to a one-layer Transformer. However, in multi-layer architectures, CoT reasoning steps might be distributed across layers, which could alter the strictly "simultaneous" nature of the RL learning dynamics observed in a single layer.
2. The RL setup relies on perfect intermediate, step-wise supervision, whereas practical RLHF typically uses sparse, outcome-only terminal rewards. The paper would benefit from explicitly contextualizing this distinction early on to prevent misleading interpretations of the RL capabilities.
3. The theoretical setup relies on a hard-coded "pretrained" attention mask that strictly enforces the exact binary tree topology of the recursive decomposition. This artificial constraint limits the realism of the findings, as it remains unclear if the established learning dynamics would hold under more flexible or unconstrained attention patterns.

---

> ### Author Rebuttal · Authors · 2026-03-30
>
> We thank the reviewer for the valuable and constructive comments.
> 1. **Response to W1:** We agree that our current analysis is limited to a one-layer transformer, and that, in a deeper model, different parts of the reasoning process may be distributed across layers. Extending the analysis to the multi-layer case is thus an important open problem.
>
>    Our aim in this paper is to identify the learning dynamics in a setting where the analysis is tractable. Thus the one-layer model is a first step. In addition, we would like to highlight that our paper still gives one of the first theoretical results on this timely and nontrivial question. We will make this scope clearer in the revision and highlight the multi-layer case as an important future direction.
> 2. **Response to W2:** In revision, we will emphasize in Abstract that the analysis is for RL with process-reward and clarify the distinction early on in Introduction to prevent misleading.
>
>    Meanwhile, we believe RL with process reward is also practically relevant when intermediate reasoning steps are verifiable, e.g., [1, 2] suggest that process-based reward can improve reasoning-trajectory quality or model performance. Overall, we will revise our abstract and introduction as suggested by the reviewer to clarify our scope.
> 3. **Response to W3:** We thank the reviewer for this insightful question. During the rebuttal, as also suggested by **Reviewer yzEa**, we worked out that our current critical gradient component framework can be extended beyond the binary-tree topology to a more common **serial CoT** setting.
>
>     In this serial CoT setting, the problem has the structure $$y_1=\phi_2(x_{i_1},x_{i_2}),\ y_{j}=\phi_2(y_{j-1},x_{i_{j+1}}),\ j\geq2,$$ where $x$ is the input and $y$ is a serial CoT. For the transformer, we can replace the current tree mask with a more flexible and weaker serial mask that, when generating $y_j$, allows the model to attend to all input bits of $x$ but only to the most recent CoT step $y_{j-1}$. This matches the serial recursion and removes the exact binary tree in Fig.1a (page 3). In this setting, the same proof strategy can be adapted, and the same qualitative learning dynamics can be recovered. We will add this extension in the revision. Please also refer to **the first point of our response to Reviewer yzEa** for details.
>
>     In addition, we also agree with the reviewer that, for even more complex setup, it is still unclear if the established learning dynamics holds, which should be an interesting open question. Hence, we will also clarify this scope in revision.
> ---
> 1. **Response to Q1:** We do not intend to bake the target logic into the model. $\psi$ is chosen only to **ensure that the model has enough expressivity** to represent the target $\phi_2$. In this sense, $\psi$ is chosen so that $\psi((z_1+z_2)/2)$ is large when $\phi_2(z_1,z_2)=1$, which makes the correct model output interpretable as a valid generation probability. Without a suitable $\psi$, learning is impossible because the model cannot express the target function regardless of the optimization method.
>
>     This gives a partial inductive bias, but it does not fundamentally solve the task, because the model still has to identify the correct relevant positions, which is the central part. Thus, $\psi$ specifies the right function class, but learning still requires recovering the relevant positions via learning the right attention pattern. We will clarify this point and that a mismatch between the task and $\psi$ leads to failure of learning in revision.
> 2. **Response to Q2:** We believe the key part of our theory is not the binary-tree structure, but that **gradient signal can distinguish relevant nodes from irrelevant ones**, which can be captured through the separation of the critical gradient component---the most informative part of gradient signal. Thus we were also able to extend the same framework to a more common serial CoT setting during the rebuttal (please also see our response to W3), where the topology changes but the relevant nodes of each step are still distinguishable from irrelevant ones through the gradient signal.
>
>     In this sense, the separation condition is the formal way of expressing that the informative part of the gradient is stronger on the relevant nodes than on the irrelevant ones. Thus, for a more entangled graph, we think the separation condition would be:
>
>     - if each relevant node still has larger gradient signal than irrelevant ones, then similar separation condition should still apply;
>
>     - if relevance is no longer visible from gradient signal at the level of individual nodes, then the current clean separation condition would fail in its present form; and needs to be generalized to a group-level version.
> ---
> Reference
>
> [1] Uesato et al. Solving math word problems with process and outcome-based feedback. 2022.
>
> [2] Shao et al. Deepseekmath: Pushing the limits of mathematical reasoning in open language models. 2024

---

> > ### Author Rebuttal · Reviewer_1fvS · 2026-04-03
> >
> > Thanks for the responses. I will keep my score.

---

### Official Review · Reviewer_YhNh · 2026-03-11

**Soundness:** 4
**Presentation:** 3
**Significance:** 4
**Originality:** 4
**Overall Recommendation:** 5
**Confidence:** 2

**Summary:**

This study examines two learning strategies of Chain of Thought (CoT) in Transformer: reinforcement learning (RL) and superfined-tuning (SFT). With a single-layer transformer, theoretical analysis shows that both strategies allow Transformers to learn CoT but in different ways. RL enables simultaneous learning of CoT, whereas SFT enables step-by-step learning. The core of the theory is the critical gradient component, which relates to the separation conditions for successful learning. The case studies show that the separation condition holds for several Boolean functions, including k-PARITY.

**Compliance With Llm Reviewing Policy:**

Affirmed.

**Final Justification:**

The rebuttal clarifies the claims and positioning of this work. This work presents an interesting contrast between SFT and RL, while the setup might be a little restrictive.

**Key Questions For Authors:**

Q1. Are the assumptions on the problem/model setup generally the same as those in (Kim & Suzuki, 2025)? Or are there any further assumptions?

Q2. Does this work present any advantage of RL over SFT (or vice versa)? The results show that RL is simultaneously learning CoT. Does it lead to more efficient learning, or does it make the learning challenging? Empirically, RL seems to be more unstable than SFT. Do the theoretical results align with this intuition?

**Limitations:**

No limitations are presented. The authors are encouraged to discuss the aforementioned weaknesses and questions if these are not resolved through the rebuttal.

**Strengths And Weaknesses:**

**Strength**

I found the following strength in this work.

First, the work shows an interesting contrast between RL and SFT in learning CoT by Transformers. This result is shown by a solid theoretical analysis, and the proposed concept, the critical gradient component, provides separation conditions for successful learning. Importantly, the separation conditions are shown to hold for several Boolean functions, including K-PARITY.

In my understanding, the results for SFT are more general than the prior work by (Kim & Suzuki, 2025), as no teacher forcing and data augmentation are required, addressing the gap between training and inference.

The takeaway and presentation are also clear.

---
**Weakness**

Although this work has the aforementioned strengths, there are also several weaknesses. Addressing them will strengthen the value of this work.

First, the assumptions on the problem setup seem to be strong: one-layer "pretrained" Transformer. Although I understand this based on the prior work (Kim & Suzuki, 2025), these appear to be strong and far from the realistic case.

Second, the theoretical claims are not empirically validated. As I mentioned above, the problem setup is based on a strong assumption, and thus it is worth examining if this empirically generalizes to more general cases.

Third, although several boolean functions are shown to be learnable, there is no discussion on what non-learnable cases, which would present the boundary of feasibility/infeasibility. The separation conditions should also be examined empirically.

Overall, this work relies on a strong assumption and lacks empirical validation. I appreciate the current results, but addressing them would strengthen this work.

---

> ### Author Rebuttal · Authors · 2026-03-30
>
> We thank the reviewer for appreciating our work and the constructive comments. We agree that clarifying the scope, empirical evidence, and the feasibility/infeasibility boundary will strengthen the paper.
> 1. **Response to Weakness 1:** We agree that our setting is a stylized one. Our goal is to compare RL and SFT for transformers within a unified and tractable framework. At the same time, we would also like to highlight that this paper provides, to the best of our knowledge, one of the first theoretical characterizations of this challenging and timely question.
> 2. **Response to Weakness 2:** We agree that empirical validation would strengthen our work, and our current numerical experiments are for $k$-PARITY (Appendix B, page 13) and main contributions are mostly theoretical.
>
>     Meanwhile, we note that our core intuition is **qualitatively consistent with prior empirical observations on process supervision**. Specifically, our theory shows that process reward can provide informative learning signal across the reasoning chain, and thus RL does not need to wait until earlier steps are fully learned before improving later ones. Empirically, [1] studies process- and outcome-based reward for mathematical reasoning and shows that low reasoning-trajectory error requires process-based feedback. [2] explains this advantage through precise credit assignment: process supervision specifies the exact location of any errors that occur. These observations do not directly prove our theorem, but they are qualitatively consistent with the same underlying intuition, i.e., process reward provides a step-level learning signal over multiple parts of the chain.
>
>     In the revision we will better position empirical validation on more realistic and general tasks as an important future direction, while clarifying that the present contribution is a theoretical characterization of learning dynamics in a tractable setting and how the core intuition can be consistent with prior empirical observations.
> 3. **Response to Weakness 3:** We thank the reviewer for this insightful suggestion and agree that the paper will be stronger if we clarify the boundary.
>
>    Our current theory mainly provides sufficient conditions for learnability by identifying the separation condition, which guarantees sufficient signal for parameter update. In this sense, at a high level, failure should occur when this separation collapses, i.e., the training signal is not informative enough to consistently favor the relevant positions over irrelevant ones. One concrete example provided in the paper is RL with only final reward, where the gradient signal becomes exponentially small relative to noise for $k$-PARITY, making the learning hard.
>
>    We will make the discussion explicit in revision. In addition, during the rebuttal, we additionally conducted experiments for $k$-PARITY, AND, and OR, and confirm that the **separation condition is satisfied during training** for RL with process reward and SFT. We will add these experiments in the revision.
> ---
> 1. **Response to Question 1:** Our model setup is generally similar to Kim&Suzuki, (2025). Key differences are: a). we **allow tokens to be generated via sampling** as we additionally study RL, better capturing autoregressive CoT, while Kim&Suzuki, (2025) directly let the model output be the generated token; b) our model can cover more Boolean functions, while they study parity.
> 2. **Response to Question 2:**
>    - *"Advantage of RL over SFT":* We believe that the simultaneous learning of RL with process reward indicates a potential learning-efficiency advantage in multi-step reasoning, and it will make the learning more efficient.
>    - *"Instability of RL:"* We thank the reviewer for this thoughtful question. Our results do not analyze optimization stability directly and are more for the **learning dynamics**.
>
>       More broadly, as our results show that learning dynamics depends on reward type of RL and on whether SFT uses teacher forcing, we believe the comparison on stability may also depend on these factors. For example, it is possible that RL with process reward is closer in stability to SFT than broad empirical comparisons sometimes suggest. We believe this is an interesting direction for future work when comparing RL and SFT for multi-step reasoning.
> ---
> **On Limitation:** We thank the reviewer for the suggestion. In the revision, we will add a new Limitation section that explicitly discusses: (1). Our setting relies on assumptions that make theoretical analysis tractable. We will also explicitly state what these assumptions are; (2). The empirical validation does not cover more general tasks, which should be an important next step; (3). Our current results do not directly analyze stability, which should also be an important future direction.
>
> ---
> Reference
>
> [1] Uesato et al. Solving math word problems with process and outcome-based feedback. 2022.
>
> [2] Lightman et al. Let’s Verify Step by Step. 2023

---

> > ### Author Rebuttal · Reviewer_YhNh · 2026-04-02
> >
> > I appreciate the authors' clarification. The rebuttal clarifies the claims and positioning of this work. Given that the planned update of the manuscript, I'll keep my score.

---

### Official Review · Reviewer_yzEa · 2026-03-12

**Soundness:** 4
**Presentation:** 3
**Significance:** 2
**Originality:** 2
**Overall Recommendation:** 4
**Confidence:** 4

**Summary:**

This paper considers learning sparse Boolean functions which have a compositional/recursive structure (in particular AND, OR, PARITY) using Transformers and using a tree-like chain-of-thought (CoT). More specifically, the paper compares the learning dynamics of RL training with dense process rewards and SFT without teacher-forcing. In a simplified theoretical setting, the paper shows that RL with dense rewards can learn the task with only a single GD update, while SFT (without teacher-forcing) learns the CoT in a step-wise fashion and requires $k$ iterations where $k$ is the length of the CoT.

**Compliance With Llm Reviewing Policy:**

Affirmed.

**Final Justification:**

I keep my original weak accept score. The paper is well-written and is easy to read. The main downside to me is the not using teacher-forcing for SFT and the use of process reward for RL. Both these choice are rather uncommon for LLMs. Moreover, the paper is currently written with the tone of comparing SFT and RL as two training frameworks. However, I think these paper does not explore the underlying differences between RL and SFT, and the differences shown in this paper are primarily stemming from different supervision at RL (process reward and dense) and SFT.

**Key Questions For Authors:**

- Q1. Currently the paper uses a tree-like recursive computation for the CoT. What would happen if we had a more common serial CoT, e.g., cumulative subset parities where the $j$-th token of the CoT is the parity of the first $j$ bits?

- Q2. Comparing outcome supervision and process supervision for RL, authors correctly state that functions like parities are not efficiently learnable via outcome supervision alone. Of course this would depend on the exact task, but I'm curious how the authors expect learning to happen for outcome supervision RL when tasks are learnable (take AND/OR function or real-world math tasks).

- Q3. SFT with teacher-forcing is the most common of supervised fine-tuning used for the post training of LLMs. What would be the learning dynamics of SFT with teacher-forcing? Are there any downsides to the use of teacher-forcing?

- Q4. Considering the points above does the paper provide some insight for the observation that RL (with outcome rewards) generalizes while SFT (with teacher forcing) memorizes (as cited in the paper) or similar observations? Can we identify the benefits of RL itself compared to SFT and isolate the benefits of dense supervision used? What's the more general message for practitioners?

- Minor remark: It would be nice to state the final form of the self attention after equation 2 and the simplifications.

**Limitations:**

The main limitation is the stylized assumptions common in theory works. However, it would be nice to be more upfront about them.

**Strengths And Weaknesses:**

The paper is generally well-written and is easy to read. The paper focuses on an important problem and the modeling is quite interesting.

Like the most theoretical papers on Transformers, the paper imposes lots of simplification on the architecture and training procedure. These include simplifying the weights of a single-layer Transformer, using sign GD, and imposing some simplifying structure on the attention matrices through masking. Further RL with dense process rewards and SFT without teacher-forcing are relatively less common in practice. It is also assumed that the process reward of RL has a nice structure that computes the reward of step $t$ assuming that what the model has done during step $t-1$ is correct (it doesn't check step $t$ correctness per-se).

Please see the questions below.

---

> ### Author Rebuttal · Authors · 2026-03-30
>
> We thank the reviewer for the constructive and valuable comments. We answer your questions below.
> 1. **Response to Q1:** We thank the reviewer very much for this insightful question. Our current framework can be extended to a **serial CoT**. We will add a new section in revision for this extension. Below we use k-PARITY as an example to briefly indicate it.
>
>      Let $z=(x,y)$ be the input sequence, where $x$ is the input and $y$ a more common serial CoT: $$y_1=\phi_2(x_{i_1},x_{i_2}),\ y_{j}=\phi_2(y_{j-1},x_{i_{j+1}}),\ j\geq2.$$For parity, we have $\phi_2(a,b)=ab$, and the $j$-th CoT token is the parity of the first $j + 1$ relevant bits. Under this definition, information of earlier CoT tokens is already summarized by $y_{j-1}$. Accordingly, for the transformer, we add a weaker serial mask as the prior under which, when generating $y_j$, the model can attend to all bits of $x$ but only to the most recent CoT step $y_{j-1}$, matching the serial recursion.
>
>       In this setup, our theoretical conclusions will be (supported by numerical experiments):
>      - For both **RL** and **SFT with teacher forcing**, the transformer can provably learn $k$-PARITY in one sign gradient update, exhibiting **simultaneous learning** as before;
>      - For **SFT without teacher forcing**, the direct learning fails due to noisy later-CoT-token update; however, with a filtering that suppresses the noisy later-CoT-token update when learning earlier tokens, the transformer exhibits **step-wise learning** that learns one CoT token per sign-gradient update.
>
>       The proof strategy uses the same **critical gradient component** framework as in the current paper: for each CoT step $y_j$, the relevant positions are $y_{j-1}$ and $x_{i_{j+1}}$, and one studies whether their critical gradient components dominate over those of others (separation condition). We show that these two parent positions indeed have larger critical gradient components than others. This yields the same qualitative mechanisms as in our current analysis: simultaneous learning for RL and SFT with teacher-forcing, but stepwise learning for SFT without teacher-forcing.
> 2. **Response to Q2:** Outcome RL does not provide an exact learning signal for each intermediate step. Instead, each step is updated only through its effect on the final reward. Therefore, when a task is learnable from outcome reward, learning can occur but the signal is weaker and depends on how strongly intermediate progress is correlated with final reward.
>
>     Accordingly, we expect outcome-only RL to be generally **a weaker form of simultaneous learning**, since outcome reward can offer gradient signal to multiple intermediate steps within a single update, but, in contrast to process supervision RL, this signal is not a direct one. Thus the effectiveness of the form of simultaneous learning for outcome-only RL should depend on how strongly intermediate steps are reflected in the outcome reward.
> 3. **Response to Q3:** The learning dynamics of SFT with teacher forcing will be **simultaneous learning**. The reason is that, under teacher forcing, each CoT step is trained with the correct previous steps. Therefore, the same separation condition in Theorem 3.3 is satisfied for all CoT steps simultaneously, so all steps can receive useful gradient in the same update.
>
>     The downside is the potential train-inference mismatch, i.e., at training the model only has correct prior steps, while at inference it must condition on its own generated steps.
> 4. **Response to Q4:**
>    - **Insights on comparison between RL and SFT:** We provide a partial but useful insight. The comparison is not a pure RL-vs.-SFT because we show that changing teacher forcing affects learning dynamics of SFT, and changing supervision reward affects learning dynamics of RL. Thus, the empirical comparisons between RL and SFT might conflate optimization method with these additional factors (e.g., reward design, teacher forcing), rather than isolating RL vs. SFT alone.
>    - **Benefits of RL itself:** The benefit of dense supervision is stronger intermediate credit assignment, while the benefit of RL itself compared to SFT with teacher-forcing is that it updates the model on its own sampled trajectories. Thus, here RL better matches the inference-time distribution, whereas SFT with teacher forcing optimizes all steps simultaneously but under a train–inference mismatch.
>    - **Message for practitioners:** a). Dense reward can be useful when intermediate reasoning steps are verifiable, as also illustrated by [1]; b) One should control for whether supervision is outcome- or process-based and whether training uses perfect prefixes or model-generated prefixes when comparing RL and SFT.
> 5. **Minor remark:** We will add them in the revision.
> 6. **Limitations**: We will add a Limitation section in revision to summarize them.
> ---
> [1] Shao et al. Deepseekmath: Pushing the limits of mathematical reasoning in open language models. 2024

---

> > ### Author Rebuttal · Reviewer_yzEa · 2026-04-04
> >
> > Thank you for your response and clarification.
> >
> > I still have two questions on two points:
> > - Regarding SFT with teacher-forcing, it is true that there's a potential mismatch at inference and training, but I'm not convinced this is an issue with current LLMs. In particular auto-regressive Transformers sample a token at a time, and this discretization (instead of relying on continuous variables or distributions) helps models with avoiding the typical train-inference mismatches. This is also validated by the common practice. So is there really any downside to SFT with teacher-forcing in modern LLMs?
> >
> > - Getting back to the discussed paper "SFT Memorizes, RL Generalizes:", this paper uses outcome supervision vs. teacher-forcing SFT and shows that (in their setting) RL has better generalization behaviors compared to SFT. Does your work support/connect to this work?

---

> > > ### Author Response · Authors · 2026-04-05
> > >
> > > We thank the reviewer for the follow-up questions.
> > >
> > > - We would like to clarify that our point is not to claim a strong practical downside of teacher forcing in modern LLMs, where it is indeed standard and often works well. Our point in this paper is a narrower one: teacher forcing and autoregressive training correspond to **different conditioning regimes**.
> > >
> > >     Under teacher forcing, each CoT step is always trained under the correct previous steps, and this is why, according to our analysis, SFT with teacher forcing learns the whole chain simultaneously while SFT without teacher forcing learns step-by-step. Therefore, the relevance here is this theoretical difference, rather than a broader practical criticism of teacher forcing.
> > >
> > > - Regarding *SFT Memorizes, RL Generalizes*, our theory does not directly address that comparison, because that paper studies outcome supervision, whereas our main analysis is for process reward, which is also practically relevant as illustrated by [1, 2, 3], especially when intermediate steps are verifiable.
> > >
> > >     The connection is partial, but still useful: our results show that both **conditioning regime and reward design** can substantially change the learning dynamics. In particular, teacher forcing already changes the SFT dynamics, and process reward changes the RL dynamics. Thus, our work suggests that comparisons between RL and SFT should be interpreted carefully, since they may conflate the optimization method with these additional factors, rather than isolating the RL vs. SFT distinction alone.
> > >
> > > ---
> > >
> > > Reference
> > >
> > > [1] Shao et al. Deepseekmath: Pushing the limits of mathematical reasoning in open language models. 2024
> > >
> > > [2] Uesato et al. Solving math word problems with process and outcome-based feedback. 2022.
> > >
> > > [3] Lightman et al. Let’s Verify Step by Step. 2023

---

### Decision · Program_Chairs · 2026-04-30

**Decision:**

Accept (regular)

**Comment:**

This submission compares the learning dynamics of RL and SFT in learning sparse Boolean functions with compositional structure, where a single-layer transformer is trained to solve the problem using chain-of-thought. The reviewers find the distinct learning behaviors of RL and SFT interesting, but they also raise concerns regarding the simplified and somewhat unconventional theoretical setting. The authors partly address these concerns by proposing a revision that clarifies the scope.